# A drug-free cardiovascular stent functionalized with tailored collagen supports in-situ healing of vascular tissues

Haoshuang Wu[1,5], Li Yang[1,5], Rifang Luo[1], Li Li[2], Tiantian Zheng[1], Kaiyang Huang[1], Yumei Qin[1], Xia Yang[3], Xingdong Zhang[1] & Yunbing Wang [1,4] ✉

Drug-eluting stent implantation suppresses the excessive proliferation of smooth muscle cells to reduce in-stent restenosis. However, the efficacy of drug-eluting stents remains limited due to delayed reendothelialization, impaired intimal remodeling, and potentially increased late restenosis. Here, we show that a drug-free coating formulation functionalized with tailored recombinant humanized type III collagen exerts one-produces-multi effects in response to injured tissue following stent implantation. We demonstrate that the one-produces-multi coating possesses anticoagulation, anti-inflammatory, and intimal hyperplasia suppression properties. We perform transcriptome analysis to indicate that the drug-free coating favors the endothelialization process and induces the conversion of smooth muscle cells to a contractile phenotype. We find that compared to drug-eluting stents, our drug-free stent reduces in-stent restenosis in rabbit and porcine models and improves vascular neointimal healing in a rabbit model. Collectively, the one-produces-multi drug-free system represents a promising strategy for the next-generation of stents.

Recently, cardiovascular disease has surpassed cancer as the number one threat to human health worldwide with the aging of the world's population, and by 2030, it is expected that 23.3 million people across the world will die from it[1–4]. Stent implantation has gradually replaced traditional surgical thoracotomy and become the most effective method for treating coronary artery disease (CAD)[5–7]. After implantation, the exposure of the stent inevitably damages arterial tissue, triggering a series of pathological processes, including thrombus formation and acute inflammatory response. These processes are interrelated and responsible for the excessive deposition of extracellular matrix (ECM) and proliferation of smooth muscle cells (SMCs), which ultimately causes intimal hyperplasia (the key reason for in-stent restenosis (ISR))[8]. Drug-eluting stent (DES) can effectively reduce the incidence of ISR via the local release of antihyperplasia drugs (e.g., rapamycin, sirolimus, tranilast, and everolimus)[9,10]. However, thus far, delayed healing of endothelial monolayers has been considered the main risk factor for late stent thrombosis (LST), limiting the long-term efficacy of DES[11]. Accordingly, searching the possibilities to develop drug-free stent that could both avoid LST and promote vascular neointimal healing against the antihyperplasia drugs is still of significance[12,13]. Considering the complex material/blood interactions, the development of stents (whether degradable or non-degradable) with a multifunctional surface that endows the vascular stent with anticoagulant, anti-inflammatory, and selectively enhanced endothelialization functions is a promising approach,

[1]National Engineering Research Center for Biomaterials and College of Biomedical Engineering, Sichuan University, Chengdu 610065, China. [2]Institute of Clinical Pathology, West China Hospital of Sichuan University, Chengdu 610041, China. [3]Shanxi Key Laboratory of Functional Proteins, Shanxi Jinbo Bio-Pharmaceutical Co., Ltd., Taiyuan 030032 Shanxi, China. [4]Tianfu Jincheng Laboratory (Frontier Medical Center), Chengdu 610213, China. [5]These authors contributed equally: Haoshuang Wu, Li Yang. ✉e-mail: yunbing.wang@scu.edu.cn

regulating vascular neointimal remodeling and promoting the regeneration of injured tissues[14,15].

Recombinant humanized collagen type III (rhCol III) was meticulously tailored by advanced structural biology and genetic engineering technologies[7]. rhCol III possessed enhanced cell adhesion activity and anticoagulant properties, which are attributed to the retention of highly adhesive fragments (Gly-Glu-Arg (GER) and Gly-Glu-Lys (GEK)) of humanized collagen type III and the bypass of the hydroxyproline (O) sequence that might induce platelet adhesion and activation. In addition, the properties of rhCol III in terms of suitable water solubility, low inflammatory response, and the promotion of vascular remodeling have been reported[7]. Multifunctional rhCol III has demonstrated promise in treating cardiovascular diseases such as atherosclerosis, myocardial infarction, and valvular heart disease[3,16,17]. Over the past two decades, it has been confirmed that tissue-inducing biomaterials, a class of optimized biomaterials, possess the ability to positively mediate tissue repair or regeneration without growth factor or drug stimulation[18,19]. Inspired by tissue-induced biomaterials, we propose a hypothesis that the stent coating with only rhCol III coating (drug-free) can exert multiple effects in response to the complex microenvironment after stent implantation and that the drug-free coating on

cardiovascular stents can reduce the LST risk, inhibit ISR, and promote vascular neointimal healing.

In this article, we report the design of a one-produces-multi drug-free coating formulation using tailored recombinant humanized collagen type III (rhCol III) to modulate vascular remodeling, with the prospect of addressing challenges such as ISR and incomplete endothelialization from drug-eluting stents (Fig. 1). Tailored rhCol III was efficiently linked on an amine-rich surface prepared by the copolymerization of dopamine and polyethyleneimine. The stability of the one-produces-multi coating was investigated using a simulated blood flow system. In vitro and in vivo assays suggested that the single component rhCol III coating can suppress the inflammatory response by mediating the transition of macrophages to the M2 phenotype, accelerating the endothelialization process, and inhibiting the transition of SMCs from a synthetic to a contractile phenotype, thereby reducing ISR and facilitating long-term stent patency in both small animal and large animal models. Combining all of the above results, this one-produces-multi strategy (drug-free coating) demonstrated immense potential in vascular neointimal healing and is expected to be the direction for next-generation vascular stent.

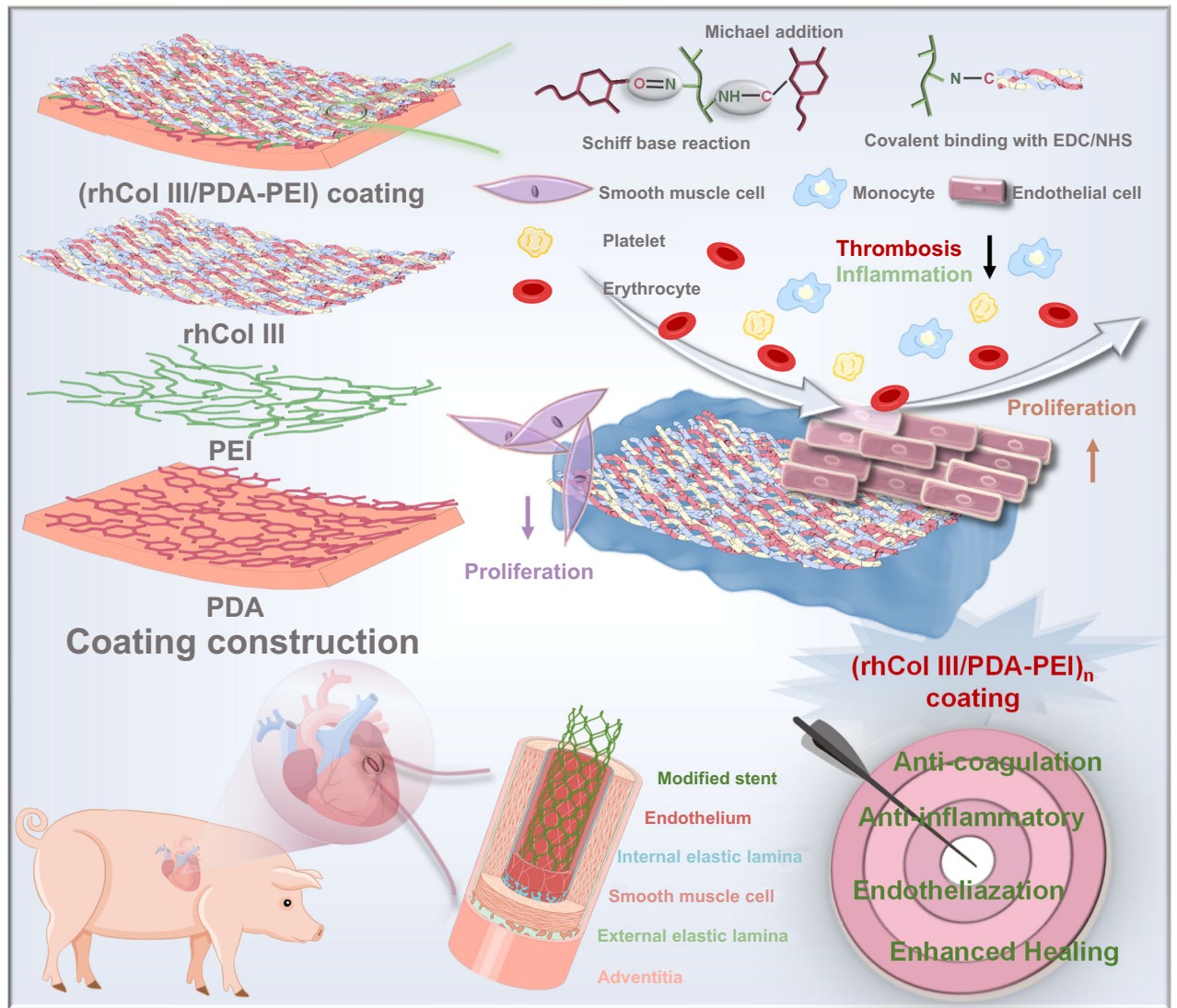

**Fig. 1 | The functional model, possible existing interactions, and process of preparing the one-produces-multi coating with anticoagulant, anti-inflammatory, and enhanced endothelialization properties for cardiovascular** implants/interventional devices. PDA polydopamine, PEI polyethyleneimine, rhCol III recombinant humanized collagen type III.

## Results and discussion

### Construction and characterization of (rhCol III/PDA-PEI)$_n$ coatings

The drug-free formulation of (rhCol III/PDA-PEI)$_n$ was shown in Fig. 1. In brief, poly (l-lactic acid) (PLA) substrates, including sheets and stents, were first treated with dopamine and polyethyleneimine (PEI) to generate a classic mussel-mimicking amine-rich surface (PEI-PDA)

for further modification[20–22]. Then, rhCol III was robustly immobilized on the PEI-PDA surface by covalent bonding and electrostatic interactions in the presence of 1-ethyl-3(3-dimethyl aminopropyl) carbodiimide (EDC) and N-hydroxysuccinimide (NHS), resulting in the formation of (rhCol III/PDA-PEI)$_n$. The immobilization of rhCol III on the PDA-PEI coating could be visualized directly via the fluorescence signal of FITC-labeled rhCol III. As shown in Fig. 2a and Supplementary

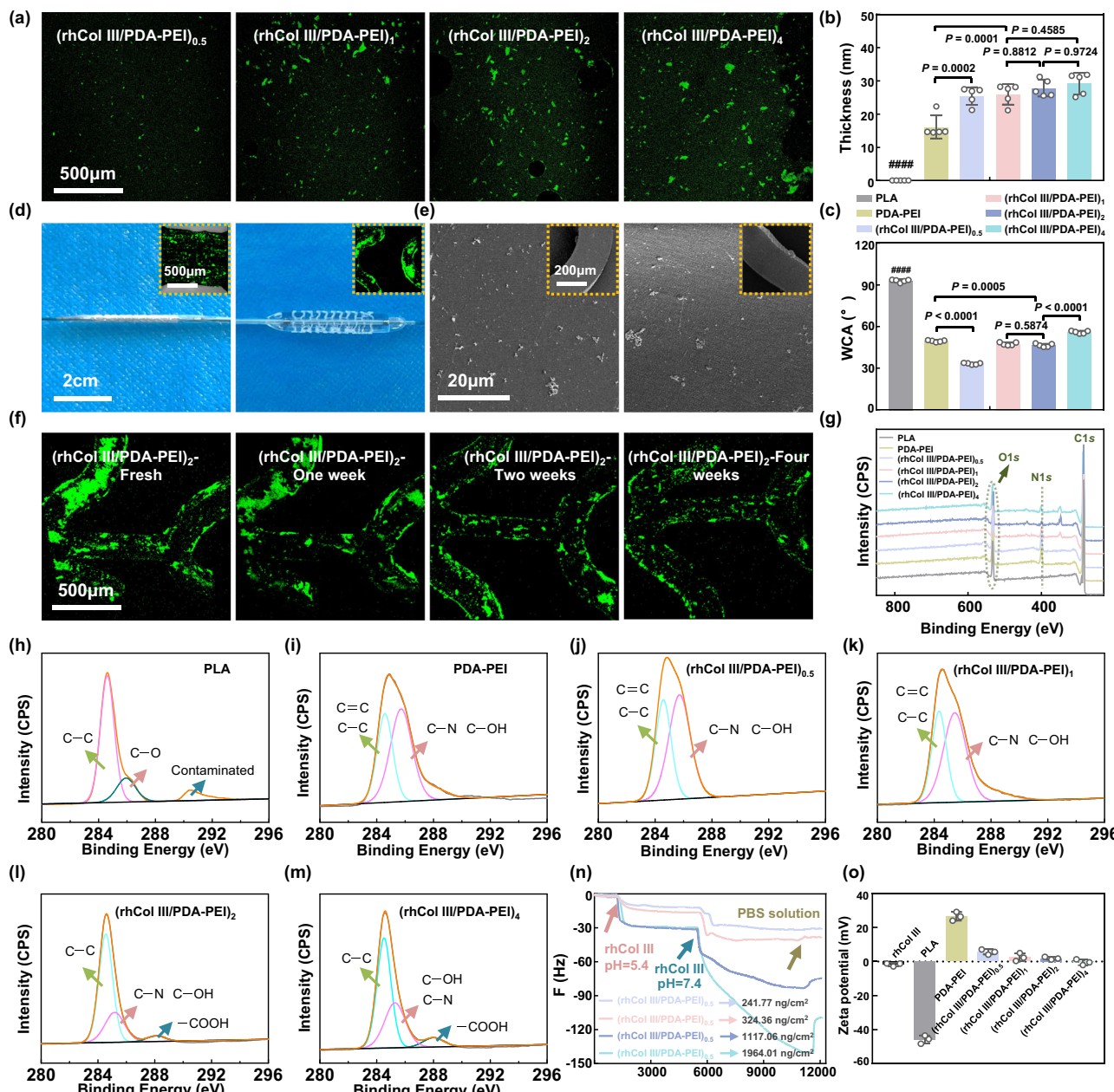

**Fig. 2 | Development of (rhCol III/PDA-PEI)$_n$ drug-free coatings for vascular stents. a** Representative FITC fluorescence signals (green) of (rhCol III/PDA-PEI)$_n$-coated ($n$ = 0.5, 1, 2, 4) PLA sheets strengthened as the concentration of rhCol III increased. Scale bars, 500 μm. Quantification of (**b**) Thickness and (**c**) Water contact angle of the (rhCol III/PDA-PEI)$_n$ coatings ($n$ = 5 independent samples). **d** Representative photographs and fluorescence images of PLA stents coated with (rhCol III/PDA-PEI)$_2$ before and after balloon dilation in PBS at 37 °C. Scale bars, 2 cm and 500 μm. **e** SEM images of (rhCol III/PDA-PEI)$_2$-coated PLA stents before and after balloon dilation in PBS at 37 °C. Scale bars, 20 μm and 200 μm. Five samples were repeated independently with similar results. **f** Under the flowing system with bovine blood serum at 37 °C, the representative fluorescence signal (green) of the (rhCol III/PDA-PEI)$_2$ coating after balloon dilation, measured at 1, 2,

and 4 weeks. Fresh (rhCol III/PDA-PEI)$_2$ coating was used as a control. Scale bars, 500 μm. Five samples were repeated independently with similar results. **g** Representative wide scan XPS spectra showing the surface signal of O, C, and N from PDA-PEI- and (rhCol III/PDA-PEI)$_n$-coated ($n$ = 0.5, 1, 2, 4) PLA sheets. **h**–**m** Representative high resolution of the C1s spectrum of different samples. **n** Representative real-time QCM-D monitoring of immobilization of different rhCol III concentrations on PDA-PEI-modified gold slides. **o** Surface zeta potential of rhCol III and (rhCol III/PDA-PEI)$_n$ coatings in PBS ($n$ = 3 independent samples). One-way ANOVA with Tukey's multiple comparisons was used for the comparisons in (**b**), (**c**), and (**o**). The data are presented as the mean ± SD ($p$ values < 0.05 were considered statistically significant, #### indicated $p$ values < 0.0001 compared with other groups).

Fig. 1, no fluorescence signals were observed in the control bare PLA and PDA-PEI groups, whereas the fluorescence signals of the (rhCol III/PDA-PEI)$_n$ ($n = 0.5, 1, 2,$ and 4) coatings prepared on PLA sheets strengthened as the concentration of rhCol III increased. The results indicated that rhCol III was successfully immobilized onto the PDA-PEI coating. Then, the surface micromorphology of the (rhCol III/PDA-PEI)$_n$ coatings ($n = 0.5, 1, 2, 4$) prepared on PLA substrates including sheets and stents was observed by SEM, and the results were shown in Supplementary Figs. 2 and 3. The PLA surface was relatively smooth, while the PDA-PEI coating presented a rougher surface with few aggregated nanoparticles that were formed by the self-polymerization of dopamine (DA) into PDA and a simultaneous reaction with PEI via Michael-addition and Schiff-base reactions[23]. As the concentration of rhCol III increased, a gradually increased number of particles on (rhCol III/PDA-PEI)$_n$ were observed, and the particles appeared more uniform and equally distributed. The thickness of the (rhCol III/PDA-PEI)$_n$ coating increased from 25.1 to 29.2 nm with increasing concentrations of rhCol III from 0.5 to 4 mg/ml (Fig. 2b). The difference between the (rhCol III/PDA-PEI)$_n$ groups was insignificant, indicating that the coatings did not grow persistently in a vertical orientation but instead were distributed more equally, consistent with the SEM results. After being modified with the PDA-PEI coating, the PLA substrate exhibited significantly improved hydrophilicity due to the introduction of abundance amine groups, with the water contact angle (WCA) decreasing from 92.3° to 24.1°. The hydrophilicity of (rhCol III/PDA-PEI)$_n$ gradually decreased (the WCA ranged from 33.3° to 55.9°) with increasing rhCol III concentration (Fig. 2c). The WCA results further supported the above conclusions. The results above confirmed that higher concentrations of rhCol III induced a higher coverage ratio of the (rhCol III/PDA-PEI)$_n$ coatings.

To further assess the mechanical stability of the coatings modified onto PLA stents, we initially performed a dilatation test with an angioplasty balloon in phosphate-buffered saline (PBS, pH 7.4) solution at 37 °C. No obvious attenuation of the fluorescent signal was observed in the (rhCol III/PDA-PEI)$_2$ coating (prepared with FITC-labeled rhCol III) after stent expansion (Fig. 2d). As expected, the (rhCol III/PDA-PEI)$_n$ coatings remained intact (Fig. 2e and Supplementary Fig. 4), and the visible particles showed no clear differences after dilation. Then, the dilated (rhCol III/PDA-PEI)$_n$-modified stents ($n = 0.5, 1, 2,$ and 4) were placed in a flowing system with bovine blood serum at 37 °C, simulating a complex and dynamic physiological environment. After 4 weeks of circulation, relatively few particles were shed compared to the corresponding fresh coatings (Supplementary Fig. 5). Among them, both (rhCol III/PDA-PEI)$_2$ and (rhCol III/PDA-PEI)$_4$ maintained uniform and continuous rhCol III coverage. Consequently, we selected (rhCol III/PDA-PEI)$_2$ as the initial attempted sample to evaluate the stability of the coatings. Compared to that of the fresh (rhCol III/PDA-PEI)$_2$ coating (for control, fluorescent signal with 100%), the fluorescence signal of the (rhCol III/PDA-PEI)$_2$ coating correspondingly diminished to 89.13%, 75.64%, and 60.15% of the initial fluorescence signal intensity after circulation for 1, 2, and 4 weeks, respectively (Fig. 2f and Supplementary Fig. 6), implying that the coating was relatively stable after implantation.

The chemical structure and composition on the surface of PLA sheets modified with PDA-PEI and (rhCol III/PDA-PEI)$_n$ ($n = 0.5, 1, 2,$ and 4) coatings were investigated by XPS. As shown in Fig. 2g, an obvious peak ascribed to the N1$s$ appeared on the PEI-PDA and (rhCol III/PDA-PEI)$_n$ coatings in contrast to the PLA substrate, providing strong evidence that the PDA-PEI coating was grafted. The successful fabrication of the (rhCol III/PDA-PEI)$_n$ coatings was further confirmed by the C1$s$ high-resolution spectra (Fig. 2h–m). When rhCol III was modified on the PDA-PEI coating, a new peak appeared at 288.3 eV, which was attributed to carboxyl Group C=O bonding[24]. The signal of the COOH peak (the fitting peak at 288.3 eV) was not visualized on the (rhCol III/PDA-PEI)$_n$ ($n = 0.5, 1$) coatings, possibly due to the lower

amount and uneven distribution of rhCol III on the surface of PLA substrates[7]. However, the signal for the COOH peak was displayed on the surface of (rhCol III/PDA-PEI)$_n$ ($n = 2, 4$) coatings with increased rhCol III loading, indicating that rhCol III was successfully assembled via the formation of an EDC/NHS active layer.

The QCM-D experiment was applied to monitor the construction process of the (rhCol III/PDA-PEI)$_n$ ($n = 0.5, 1, 2,$ and 4) coatings and investigate the stability of the coatings with PBS flow. As described in previous work, a decrease in frequency implied that the molecules were absorbed on the wafer, while an increase in frequency represented the desorption of the molecules[25,26]. The F-time curve of the (rhCol III/PDA-PEI)$_n$ coatings was shown in Fig. 2n. The ΔF gradually increased as the concentration of rhCol III increased. Notably, the rhCol III adsorption amount of the (rhCol III/PDA-PEI)$_n$ coatings ($n = 2$ and 4) increased significantly compared with that of the (rhCol III/PDA-PEI)$_n$ coatings ($n = 0.5$ and 1) when the mixed solution pH was adjusted from 5.4 to 7.4, confirming that protonation of the amino group was prevented and that the reaction of the carboxyl group of rhCol III with the surface amino group was promoted. After 1.5 h, the adsorption of rhCol III reached a steady state (loading rhCol III amount: 241.77 ng/cm$^2$ for (rhCol III/PDA-PEI)$_{0.5}$, 324.36 ng/cm$^2$ for (rhCol III/PDA-PEI)$_1$, 1117.06 ng/cm$^2$ for (rhCol III/PDA-PEI)$_2$, and 1964.06 ng/cm$^2$ for (rhCol III/PDA-PEI)$_4$). The value of F slightly increased when PBS was added, suggesting that rhCol III was strongly bonded to the wafer surface after PBS rinsing. The surface zeta potential shift of different groups at pH 7.4 was shown in Fig. 2o. The surface potentials of PLA and PDA-PEI were −45.89 mV and 26.52 mV, respectively. After rhCol III was grafted, the surface potentials of (rhCol III/PDA-PEI)$_n$ decreased gradually, from 5.60 ± 1.52 mV (for the (rhCol III/PDA-PEI)$_{0.5}$ group) to −0.46 ± 1.64 mV (for the (rhCol III/PDA-PEI)$_4$ group). The surface potential of (rhCol III/PDA-PEI)$_4$ was almost identical to that of pure rhCol III (−1.72 ± 1.15 mV), indicating that rhCol III covered the PLA substrate uniformly and continuously. In summary, the above results confirmed that the (rhCol III/PDA-PEI)$_n$ coatings were successfully constructed.

## Drug-free (rhCol III/PDA-PEI)$_n$ coatings improve blood compatibility

As biomedical devices contact blood after implantation, it is vital to evaluate their antithrombotic properties in vitro and in vivo. The aggregation and activation of platelets on the surface of samples is closely associated with in-stent restenosis and LST[27]. The anticoagulant ability of uncoated, PDA-PEI-coated, and (rhCol III/PDA-PEI)$_n$-coated PLA was evaluated in vitro by the platelet adhesion test (Fig. 3a). After 2 h of incubation, severe platelets adhered to the surface of bare PLA and the PDA-PEI coating with a full spread shape or a typical pseudo-podia shape, indicating that the platelets were highly activated. However, platelet adhesion on the surface of the (rhCol III/PDA-PEI)$_n$ coatings reduced significantly and showed a spherical shape with excellent anticoagulant effect; this effect was related to rhCol III bypassing the hydroxyproline (O) sequence, which may induce platelet adhesion and activation (Fig. 3b). Quantitative analysis also confirmed that the (rhCol III/PDA-PEI)$_n$ groups (with concentrations of 0.5, 1, 2, and 4) could significantly inhibit the adhesion and activation of platelets (Fig. 3c, d). Notably, the area coverage and the number of adherent platelets on the surface of the (rhCol III/PDA-PEI)$_n$ groups decreased sharply at concentrations greater than 2, indicating the importance of a sufficient rhCol III content for platelet inhibition.

The antithrombogenic property was investigated by an ex vivo arteriovenous shunt assay after positive in vitro results. The PDA-PEI and (rhCol III/PDA-PEI)$_n$ coatings ($n = 0.5, 1, 2,$ and 4) were prepared on the PVC tubes, which were then carefully connected with the right carotid artery and left external jugular vein of the rabbits (Fig. 3e). After 2 h of ex vivo circulation, there was severe thrombus formed on the surface of the bare PLA and PDA-PEI groups with the network of fibers linked with red blood cells and activated platelets, whereas only

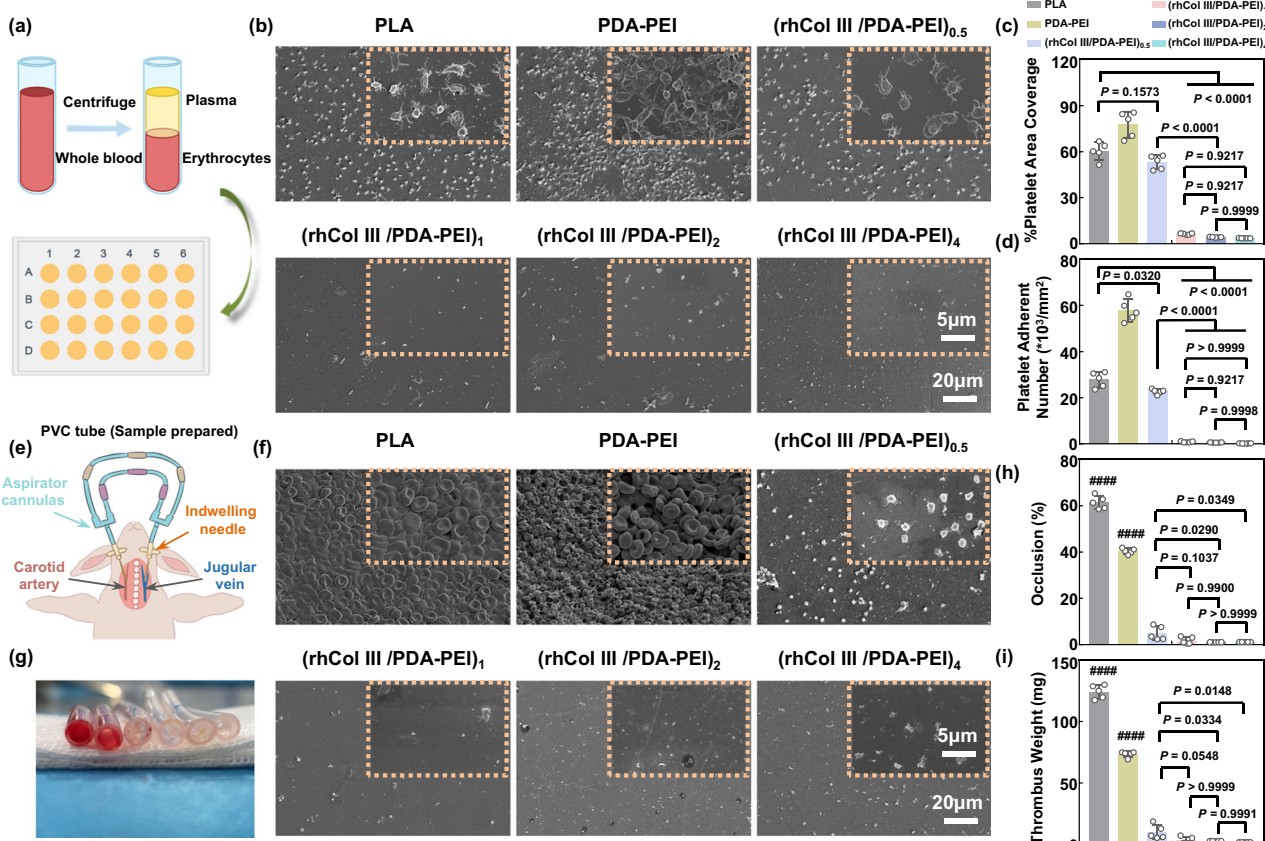

**Fig. 3 | (rhCol III/PDA-PEI)$_n$ coatings ($n$ = 0.5, 1, 2, and 4) improve blood compatibility. a** Schematic of the platelet adhesion test in vitro. **b** Representative SEM morphology of adhered platelets on the surfaces of uncoated, PDA-PEI-, and (rhCol III/PDA-PEI)$_n$-coated PLA sheets. Scale bars, 5 µm and 20 µm. Quantitative analysis results of (**c**) platelet area coverage and (**d**) platelet adherent number on the surfaces of different samples ($n$ = 5 independent samples). **e** Ex vivo schematic diagram of the arteriovenous shunt model in New Zealand white rabbits. **f** Representative SEM images of thrombi formed on different samples after 2 h of blood circulation. Scale bars, 5 µm and 20 µm. **g** Cross-sectional photographs of thrombi deposited on uncoated, PDA-PEI- and (rhCol III/PDA-PEI)$_n$-coated PVC tubes (Thrombus was red in color). Quantification of (**h**) occlusion rates of various samples ($n$ = 5 independent samples in independent animals). **i** Weight of thrombi formed on the surfaces of different samples ($n$ = 5 independent samples in independent animals). One-way ANOVA with Tukey's multiple comparisons was used for the comparisons in (**c**), (**d**), (**h**), and (**i**). The data are presented as the mean ± SD ($p$ values < 0.05 were considered statistically significant, #### indicated $p$ values < 0.0001 compared with other groups).

a few platelets with spherical shapes adhered to the surface of the (rhCol III/PDA-PEI)$_n$ coatings ($n$ = 1, 2, 4), implying that these platelets exhibited a resting, nonactivated state (Fig. 3f). Furthermore, we quantitatively evaluated the occlusion rates and thrombus weights of different groups in the circuit. The cross-sectional areas of the PLA and PDA-PEI group circuits were reduced by 61.43% and 40.43%, respectively. This phenomenon was consistent with other studies, in which the tight red blood cell-platelet contact and the dense network of thin fibrin fibers formed on the surface of PLA facilitated occlusion of the vessel lumen compared with PDA-PEI[28]. However, the cross-sectional area of the circuit coated with (rhCol III/PDA-PEI)$_n$ ($n$ = 1, 2, 4) coatings was only reduced by 6.49%, 1.5%, and 0.98%, respectively (Fig. 3g, h). The thrombus weight adherent on the surface of the (rhCol III/PDA-PEI)$_n$ ($n$ = 1, 2, 4) groups was significantly decreased compared to that of the bare PLA and PDA-PEI groups, which further supported the above observations (Fig. 3i). In conclusion, the amplified functionalization of the rhCol III coating reduced platelet adhesion and aggregation, exhibiting excellent anticoagulant properties and showing the potential to improve the hemocompatibility of blood-contacting biomaterials.

From the above results, it was concluded that PDA-PEI could induce severe thrombosis, and the lower rhCol III loading of the (rhCol III/PDA-PEI)$_{0.5}$ coating might attenuate the subsequent biological functions. Therefore, (rhCol III/PDA-PEI)$_n$ ($n$ = 1, 2, 4) coatings were utilized to study other subsequent biological characterizations, with the exclusion of the PDA-PEI and (rhCol III/PDA-PEI)$_{0.5}$ groups.

## Drug-free (rhCol III/PDA-PEI)$_n$ coatings regulate the growth behavior of HUVECs

The blood vessel wall is inevitably damaged during vascular stent implantation. This behavior triggers a series of pathological reactions, including thrombogenesis, inflammatory reactions, and excessive smooth muscle cell (SMC) migration and proliferation, which may lead to in-stent restenosis (ISR). The dense and healthy endothelial layer could effectively contribute to the excellent thromboresistance of the endothelium and its positive functions in vascular wall remodeling[29]. Compared with the bare PLA substrate, the (rhCol III/PDA-PEI)$_n$ ($n$ = 1, 2, 4) coatings loaded with different amounts of rhCol III all showed better coverage of HUVECs according to the results of rhodamine-conjugated phalloidin (TRITC-Phalloidin), 4′,6-diamidino-2-phenylindole (DAPI) staining, and fluorescein diacetate (FDA) staining after 1, 3, 5, and 7 days of culture, indicating good HUVEC compatibility (Fig. 4a and Supplementary Fig. 7). Of note, HUVECs adherent to the (rhCol III/PDA-PEI)$_n$ ($n$ = 1, 2, 4) surface formed a confluent monolayer after 7 days of culture, but those adherent to the PLA surface did not. The above phenomenon occurred because rhCol III retained the highly adhesive fragments GER and GEK of humanized collagen type III. As shown in Fig. 4b, c, the number of HUVECs on the

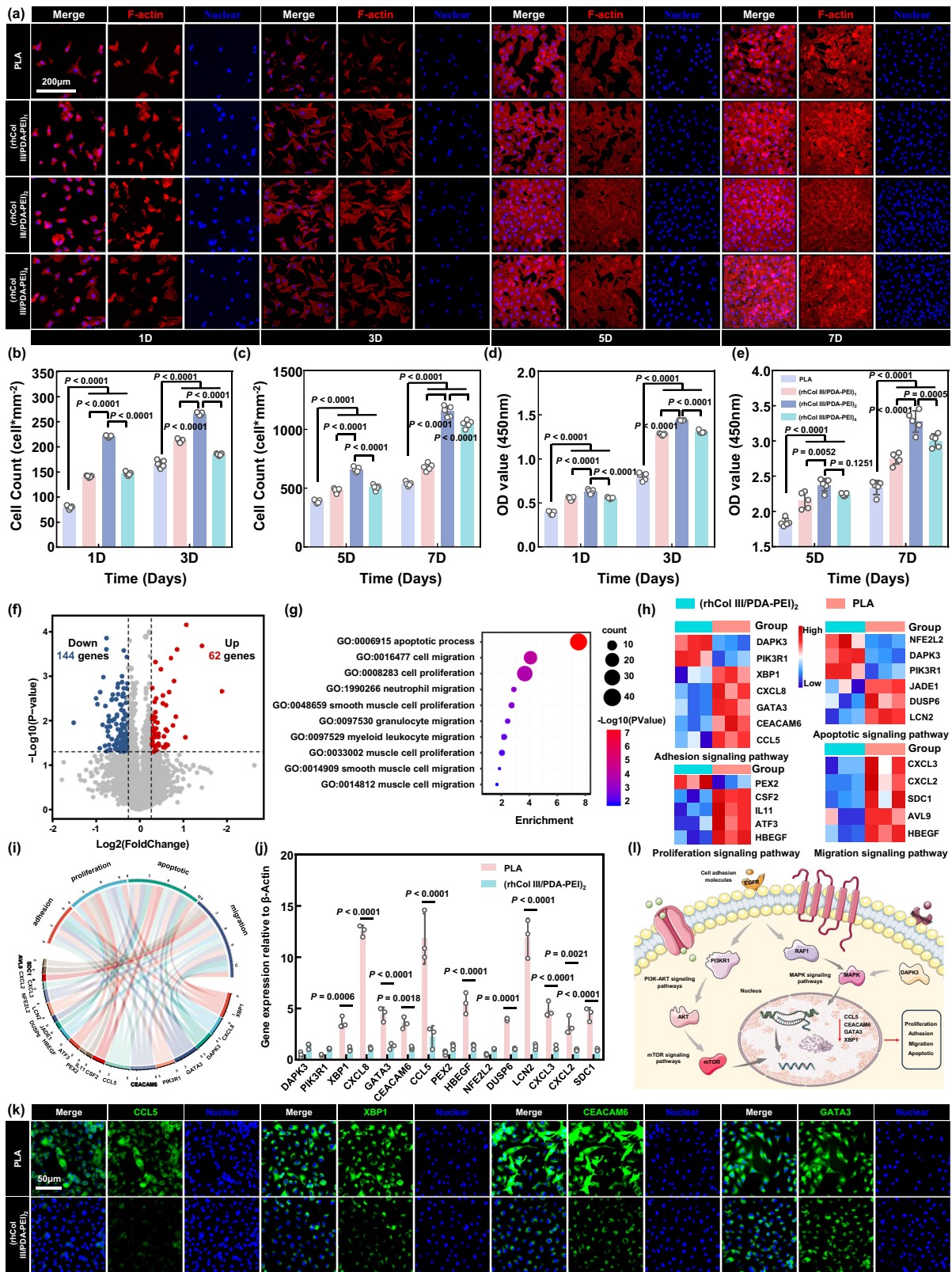

surface of (rhCol III/PDA-PEI)$_n$ coatings was higher than that on the surface of the control group at the designated time points. As detailed, the HUVECs in the PLA group merely exhibited a higher growth rate in the early stage (80 cells/mm² after 1 day and 166 cells/mm² after 3 days), whereas the (rhCol III/PDA-PEI)$_n$ groups consistently demonstrated a dominant advantage in promoting cell growth (e.g., (rhCol III/PDA-PEI)$_2$, 659 cells/mm² after 5 days and 1165 cells/mm² after 7 days).

The CCK-8 assay was used to determine cell viability, and the results were shown in Fig. 4d, e. Most impressively, compared with the rest of the samples, the (rhCol III/PDA-PEI)$_2$ sample showed the best cell viability at the designated time points; however, the cell viability of the (rhCol III/PDA-PEI)$_4$ group did not consistently increase as the rhCol III concentration increased from 2 to 4 mg/ml. This dose-independent pattern might be because the surface potential of the (rhCol III/PDA-

**Fig. 4 | (rhCol III/PDA-PEI)ₙ coatings (n = 0.5, 1, 2, and 4) regulate HUVEC behaviors. a** Representative rhodamine-conjugated phalloidin (TRITC-Phalloidin, red) and 4′,6-diamidino-2-phenylindole (DAPI, blue) fluorescence images of HUVECs showing the (rhCol III/PDA-PEI)ₙ coatings loaded with different amounts of rhCol III all encouraged HUVECs compared with control PLA group. Scale bars, 200 μm. Quantification of (**b, c**) cell number and (**d, e**) cell viability of HUVECs cultured on uncoated and (rhCol III/PDA-PEI)ₙ-coated PLA (n = 1, 2, and 4) sheets after 1, 3, 5, and 7 days of culture (n = 5 independent samples). **f** Volcano plot showing differentially expressed genes in the (rhCol III/PDA-PEI)₂ group compared to the PLA control group. Downregulated and upregulated genes are colored blue and red, respectively, at significantly differentially expressed thresholds │log₂FC│ > 1.2 and p < 0.05. **g** Gene Ontology (GO) analysis of differentially expressed genes in PLA versus (rhCol III/PDA-PEI)₂. **h** Heatmap of the differentially expressed genes in the PLA and (rhCol III/PDA-PEI)₂ groups. **i** Circular visualization of the results of gene-annotation enrichment analysis. **j** Quantification of the expression of representative genes in PLA versus (rhCol III/PDA-PEI)₂, validated by qRT-PCR arrays. The value in the PLA group was normalized to that in the (rhCol III/PDA-PEI)₂ group (n = 3 independent samples). **k** Representative immuno-fluorescence staining of CCL5 (green), GATA3 (green), XBP1 (green), and CEACAM6 (green) of HUVECs on 3 days of culture. Scale bar, 50 μm. **l** Schematic diagram of the potential three signaling pathways involved in the regulation of HUVEC behavior induced by the (rhCol III/PDA-PEI)₂ coating, including PI3K/AKT, mTOR, and MAPK. Two-way ANOVA with Tukey's multiple comparisons was used for the comparisons in (**b**)–(**e**) and (**j**). Two-sided Student's t test with multiple testing corrections was used in (**f**) and (**g**). The data are presented as the mean ± SD (p values < 0.05 were considered statistically significant).

PEI)₄ coating was approximated and tended to approach 0 mV with increasing rhCol III loading, which was similar to zwitterion surface behavior, thus discouraging cell adhesion.

Based on the above results, the (rhCol III/PDA-PEI)₂ coating was selected to further clarify the changes in gene expression after coculture with HUVECs by global gene expression profiling. Compared with the PLA control group, 62 upregulated genes and 144 downregulated genes were identified in the (rhCol III/PDA-PEI)₂ group, as shown in the volcano plot (abbreviations and full names of some genes labeled in the volcano plot were provided in Supplementary Table 1) (Fig. 4f). The Gene Ontology (GO) database was used for gene set enrichment analysis to identify related pathways. The results showed that compared with the control group, the (rhCol III/PDA-PEI)₂ group influenced the HUVEC behavior primarily by regulating cell adhesion, proliferation, migration, and apoptotic pathways (Fig. 4g–i). In detail, (rhCol III/PDA-PEI)₂ treatment favored the normal development of HUVEC adhesion, migration, and proliferation, as indicated by the downregulated expression of XBP1[30], CCL5[31], GATA3[32], and CEACAM6[33], which were confirmed to be associated with adverse cell growth behavior. Compared with the PLA group, the (rhCol III/PDA-PEI)₂ group further upregulated the expression of cancer cell apoptosis-related genes such as DAPK3[34] and PIK3R1[35], which contributed to maintaining homeostasis. Moreover, the upregulation of the CXCL2, CXCL3[36], and HBEGF[37] genes in the PLA group promoted the development of atherosclerosis. Among them, CCL5, GATA3, XBP1, and CEACAM6 were involved in all the abovementioned signaling pathways, suggesting that they might regulate the fate of HUVECs. Quantitative real-time polymerase chain reaction (RT-PCR) array further confirmed that the four genes (CCL5, GATA3, XBP1, and CEACAM6) were significantly downregulated in the (rhCol III/PDA-PEI)₂ group compared with the control group (Fig. 4j). We also performed immunofluorescence staining (Fig. 4k and Supplementary Fig. 8), and the results showed significant downregulation of the above-listed four genes in the (rhCol III/PDA-PEI)₂ group compared with the PLA group, consistent with the abovementioned results.

Combining the results of transcriptomic sequencing, PCR assay, and immunofluorescence staining mentioned above, we hypothesized that the potential underlying mechanism of (rhCol III/PDA-PEI)₂ treatment on the regulation of HUVEC behavior and physiological functions may be associated with the following signaling pathways (Fig. 4l). It is well known that the phosphatidylinositol 3-kinase (PI3K)/AKT/mammalian target of rapamycin (mTOR) signaling pathway plays key roles in cellular functions including adhesion, proliferation, migration, metabolism, and apoptosis, as well as angiogenesis. In particular, PI3KR1 is activated by epidermal growth factor receptor (EGFR), which subsequently binds to PI3K to activate serine/threonine protein kinase B (AKT) protein (PI3K/AKT signaling pathway)[38]. The phosphorylation of mTOR with activated AKT can activate signals such as growth factors, energy levels, cell stress, and amino acids to phosphorylate substrates and enhance cell metabolism, thus promoting HUVEC growth and proliferation (mTOR signaling pathway)[39].

Moreover, EGFR could be activated by extracellular growth factors and provide binding sites for adapter proteins. Various genes reportedly involved in the activation of MAPK, such as RAF1[40] and DAPK3[34], regulate HUVEC adhesion and prevent apoptosis (MAPK signaling pathway). In summary, the above signaling pathways influence each other and synergistically regulate the cell development microenvironment.

## Drug-free (rhCol III/PDA-PEI)ₙ coatings induce the conversion of HUASMCs to contractile phenotype

Inhibiting excessive proliferation of HUASMCs early in implantation could effectively prevent subsequent late thrombosis, late restenosis, and material failure. The attachment and morphology of HUASMCs were investigated by TRITC-phalloidin and DAPI fluorescence images. Compared with the PLA group, the (rhCol III/PDA-PEI)ₙ (n = 1, 2, and 4) coatings all suppressed the adhesion and proliferation of HUASMCs to varying degrees after 1, 3, 5, and 7 days of culturing (Fig. 5a and Supplementary Fig. 9a). Obviously, the number of HUASMCs adhering to the surface of all groups increased, especially with the highest growth rate in the PLA group with the extension of culturing time from 1 to 5 days (e.g., 360–1123 cells/mm² for the PLA, 313–434 cells/mm² for the (rhCol III/PDA-PEI)₁, 233–380 cells/mm² for the (rhCol III/PDA-PEI)₂, and 220–436 cells/mm² for the (rhCol III/PDA-PEI)₄). However, the counts of HUASMCs adhered to the surface of all groups did not show a continued increase with the prolongation of the incubation time to 7 days but rather gradually decreased, showing signs of apoptosis (Fig. 5b and Supplementary Fig. 9b). The results of the CCK-8 assay also confirmed that cell proliferation in the PLA group was significantly enhanced compared with that in the (rhCol III/PDA-PEI)ₙ groups at the designated time points (Fig. 5c and Supplementary Fig. 9c). In particular, the extent of inhibition of HUASMC proliferation remained almost unchanged between the (rhCol III/PDA-PEI)ₙ (n = 1, 2, and 4) coatings. This phenomenon may be attributed to the surface potentials of the coatings gradually approaching 0 mV with increasing rhCol III loading, as observed in HUVECs. The diverse growth trend of (rhCol III/PDA-PEI)ₙ (n = 1, 2, and 4) coatings on HUVECs versus HUASMCs might be due to the different sensitivities and responsiveness that interact between the material surface/interface and cells[7].

As mentioned in HUVECs, transcriptome analysis was performed to further uncover the suppressive effects of the drug-free coating on HUASMCs. Similarly, PLA and (rhCol III/PDA-PEI)₂ were selected as the control and experimental groups, respectively. After 3 days of incubation with HUASMCs, 182 and 197 genes were significantly upregulated and downregulated, respectively between the (rhCol III/PDA-PEI)₂ and control groups (Fig. 5d) under the threshold treatments of │log₂FC│ > 1.5 and p < 0.05. In particular, the abbreviations and full names of some genes labeled in the volcano plot were provided in Supplementary Table 2. Moreover, genomic enrichment analysis using the GO database revealed there were mainly seven signaling pathways enriched in the (rhCol III/PDA-PEI)₂ group that influenced the growth behavior of HUASMC (Fig. 5e), including pathways associated with cell

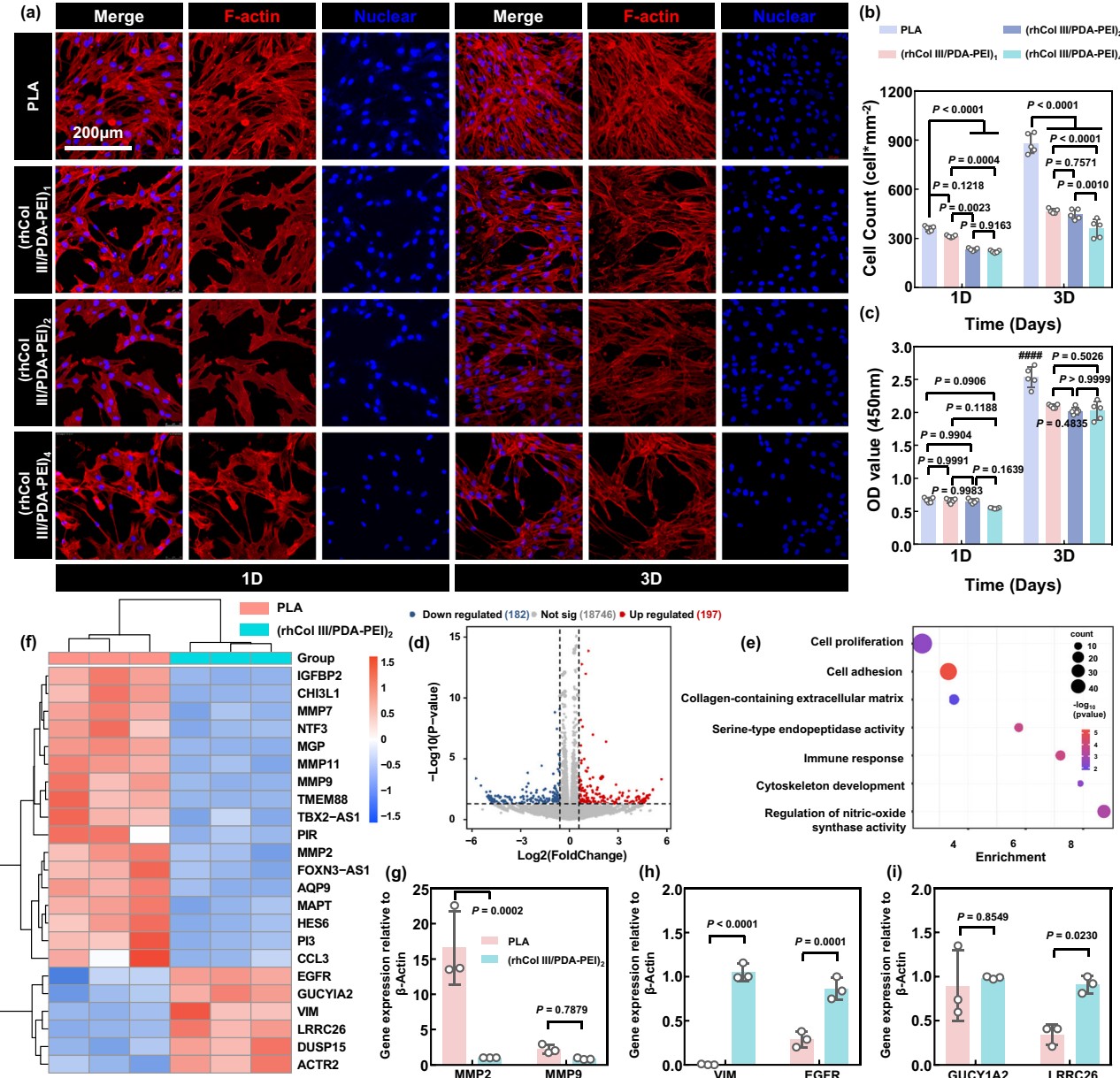

**Fig. 5 | (rhCol III/PDA-PEI)ₙ coatings (n = 0.5, 1, 2, and 4) inhibit the adhesion and proliferation of HUASMCs by mediating the phenotypic switch.**
**a** Representative fluorescence images of rhodamine- (red) and DAPI- (blue) stained HUASMCs showing that the (rhCol III/PDA-PEI)ₙ coatings loaded with different amounts of rhCol III all suppressed HUASMCs compared with the control PLA group. Scale bars, 200 μm. Quantification of (**b**) cell number and (**c**) cell viability of HUVSMCs cultured on uncoated- and (rhCol III/PDA-PEI)ₙ-coated PLA sheets (n = 1, 2, and 4) after 1 day and 3 days of culture (n = 5 independent samples). **d** Volcano plot showing differentially expressed genes in the (rhCol III/PDA-PEI)₂ group compared to the PLA control group. Downregulated and upregulated genes were colored blue and red, respectively, at significantly differentially expressed

thresholds │log₂FC│ > 1.5 and p < 0.05. **e** GO analysis of differentially expressed genes in HUASMCs. **f** Heatmap of the differentially expressed genes in the PLA and (rhCol III/PDA-PEI)₂ groups. PCR arrays displaying the expression of genes related to the (**g**) synthetic phenotype, (**h**) contractile phenotype, and (**i**) relaxation of HUASMCs in the PLA and (rhCol III/PDA-PEI)₂ groups. The value in the (rhCol III/PDA-PEI)₂ group was normalized to that in the PLA group (n = 3 independent samples). Two-way ANOVA with Tukey's multiple comparisons was used for the comparisons in (**b**), (**c**), and (**g**)–(**i**). Two-sided Student's t test with multiple testing corrections was used in (**d**) and (**e**). The data are presented as the mean ± SD (p values < 0.05 were considered statistically significant, #### indicated p values < 0.0001 compared with other groups).

proliferation, cell adhesion, collagen-containing extracellular matrix, serine-type endopeptidase activity, immune response, cytoskeleton development, and regulation of nitric-oxide synthase activity. A heatmap of differentially expressed genes (Fig. 5f) showed that the upregulation of *MAPT*[41], *MGP*[42], *MMP7*[43], and *CHI3L1*[44] in the PLA control group promoted the development of atherosclerosis. Furthermore, the PLA group further upregulated the expression of pro-proliferative genes such as *PI3*[45], *IGFBP2*[46], and *MMP7* and downregulated the

expression of the apoptosis gene *DUSP15*[47], which contributed to the proliferation of HUASMCs. Encouragingly, (rhCol III/PDA-PEI)₂ treatment induced the downregulation of the synthetic phenotypic HUASMC markers *MMP2* and *MMP9*[48], and the upregulation of the contractile phenotype HUASMC markers *VIM*[49] and *EGFR*[50], as compared to the control PLA group. The (rhCol III/PDA-PEI)₂ group also facilitated the relaxation of smooth muscle cells and thus prevented the abnormal proliferation of stented arteries, as indicated by the

upregulation of *GUCY1A2* [51] and *LRRC26* [52]. Furthermore, analysis of HUASMC phenotype-related PCR arrays (Fig. 5g–i) also supported that (rhCol III/PDA-PEI)$_2$ treatment downregulated the expression of the synthetic markers *MMP2* and *MMP9* and upregulated the contractile markers *VIM* and *EGFR*.

In summary, PLA group induced the proliferation and adverse development of HUASMCs. However, (rhCol III/PDA-PEI)$_2$ group enhanced anti-proliferative and pro-apoptotic functions, which was beneficial for maintaining the blood vessel patency rate and demonstrated the ability of the blood-contacting devices to promote neointimal healing.

### Drug-free (rhCol III/PDA-PEI)$_n$ coatings suppress the inflammatory response in vitro

After stent implantation, the tissue is damaged, and macrophages accumulate and adhere to the stent surface and distinctively respond to slight alterations in the microenvironment by polarizing into M1 and M2 phenotypes[53,54]. Classical M1 polarization is associated with the proinflammatory response. Conversely, M2 polarization is associated with anti-inflammatory properties[55]. The inflammatory response and cell damage can induce subsequent impaired endothelialization, excessive SMC proliferation, and intimal hyperplasia, leading to implantation failure[24–26]. To investigate the anti-inflammatory function of tailored rhCol III, TRITC-phalloidin and DAPI fluorescein staining were first performed. As shown in Supplementary Fig. 10, primary mouse bone marrow-derived mononuclear cells (MBMMCs) adhered to the surfaces of PLA and (rhCol III/PDA-PEI)$_1$ and presented elongated and stretched shapes. Nevertheless, more rounded MBMMCs were observed in the (rhCol III/PDA-PEI)$_2$ and (rhCol III/PDA-PEI)$_4$ groups as the rhCol III loading increased. The morphology and polarization of macrophages are closely correlated, as previously reported[51]. Thus, we further investigated the polarization of MBMMCs via immunofluorescence staining, with markers F4/80, CD86, and CD206 characterizing M1 and M2 phenotypes of macrophages, respectively. In the (rhCol III/PDA-PEI)$_n$ ($n$ = 2, and 4) groups, the expression of CD86 was significantly downregulated, while the expression of CD206 was markedly upregulated compared to the PLA and (rhCol III/PDA-PEI)$_1$ groups (Fig. 6a). In particular, both degradation products of PLA and exposed amines on the surface of (rhCol III/PDA-PEI)$_1$ (incomplete coverage by rhCol III) can lead to a strong inflammatory response[56,57]. Correspondingly, quantitative statistical results also revealed that the proportion of M1 macrophages decreased and the proportion of M2 macrophages of increased in the (rhCol III/PDA-PEI)$_n$ ($n$ = 2, and 4) groups compared to the control PLA and (rhCol III/PDA-PEI)$_1$ groups (Fig. 6b, c). At the same time, the gene expression of a list of representative cytokines (i.e., anti-inflammatory cytokines *CD206*, *Arginase-1*, *IL-10*, and *TGF-β* and proinflammatory cytokines *CD86*, *iNOS*, *TNF-α*, *IL-6*, and *IL-1β*) was determined by PCR arrays to prove that differences in macrophage phenotypes were mediated by the (rhCol III/PDA-PEI)$_n$ coatings. As expected, the expression of such factors was nicely correlated with CD86/CD206 immunofluorescence results, as indicated by significantly downregulated expression of proinflammatory genes and significantly increased expression of anti-inflammatory genes in the (rhCol III/PDA-PEI)$_n$ ($n$ = 2, and 4) groups (Fig. 6d).

Based on the results of all anti-inflammatory assays in vitro described above, (rhCol III/PDA-PEI)$_n$ ($n$ = 2 and 4) coatings were demonstrated to remarkably suppress the inflammatory response by promoting the polarization of macrophages toward the M2 phenotype and modulating the expression of inflammation-related proteins. Combining the above results, it could be concluded that both the (rhCol III/PDA-PEI)$_2$ and (rhCol III/PDA-PEI)$_4$ groups exhibited great anti-inflammatory activity with no significant difference, which was attributed to the loading of sufficient amounts of customized rhCol III, as investigated above.

### Tissue compatibility of drug-free (rhCol III/PDA-PEI)$_n$ coatings in a rat model

Subcutaneous implantation is a universal model for investigating inflammatory responses around the implant/tissue interface[58]. The in vivo anti-inflammatory ability of the (rhCol III/PDA-PEI)$_n$ ($n$ = 1, 2, and 4) coatings was further investigated after they were subcutaneously implanted in male Sprague–Dawley rats (Supplementary Fig. 11). The fibrous capsule surrounding the samples was harvested 15 and 30 days after implantation. The inflammatory response of the implants was reflected by the thickness of the fibrous encapsulation and the number of infiltrating inflammatory cells (primarily including macrophages and lymphocytes). Generally, inflammatory cell infiltration increases, and a thicker fibrous encapsulation indicates a more severe tissue response[59]. From the results of HE and Masson's trichrome staining, we discovered that the fibrous capsule formed around PLA sheets was markedly thicker than that of the (rhCol III/PDA-PEI)$_n$ ($n$ = 1, 2, 4) groups at the designated time points (Fig. 7a, b and Supplementary Fig. 12). Furthermore, we also performed CD86/CD206 immunofluorescence staining. The results revealed a decrease in the number of macrophages and an increase in the proportion of M2-phenotype macrophages (CD206/CD68 ratio) in the (rhCol III/PDA-PEI)$_n$-modified groups compared with the PLA group on both days 15 and 30, particularly in the (rhCol III/PDA-PEI)$_2$ and (rhCol III/PDA-PEI)$_4$ groups (Fig. 7c, d). Consistent with the observation in macrophages, the PLA group exhibited the largest number of infiltrated lymphocytes, and a very small amount of lymphocyte infiltration was observed in the (rhCol III/PDA-PEI)$_n$ groups (Supplementary Fig. 13a, b). In marked contrast to the massive macrophage infiltration, lymphocytes were observed in relatively small numbers, implying that the inflammatory response triggered by the implants was predominantly mediated by macrophages. In total, the in vivo anti-inflammatory results, which were consistent with the in vitro results, further substantiated that the (rhCol III/PDA-PEI)$_n$ ($n$ = 2 and 4) coatings could effectively reduce the in vivo inflammatory response of PLA substrates, displaying good tissue compatibility.

The healing of the injured endothelial layer involves complex interactions between inflammatory cells (including macrophages and lymphocytes), ECs, and SMCs, and the dysregulation of these responses could result in adverse vascular remodeling, neointimal proliferation, and restenosis[60]. Combined with the results of in vitro anticoagulation and vascular cell growth assays, it can be concluded that the (rhCol III/PDA-PEI)$_2$ coating, in addition to exhibiting superior anticoagulant and anti-inflammatory potential, is also effective for both rapid endothelialization and suppressing the excessive proliferation of SMCs. The combination of these functions is of great importance for the normal and healthy repair of the damaged endothelium. Therefore, the (rhCol III/PDA-PEI)$_2$ coating was utilized to further evaluate endothelialization in vivo.

### Stent implantation in rabbit and porcine models

To further evaluate whether our developed (rhCol III/PDA-PEI)$_n$ coatings could effectively promote in-situ endothelialization while inhibiting intimal hyperplasia, long-term stenting tests in rabbit and porcine models were conducted (Fig. 8a). After 3 months of implantation in the abdominal aorta of rabbits, the stents made of bare PLA, modified with rapamycin-eluting (RAPA) and (rhCol III/PDA-PEI)$_2$ PLA surrounded by the neointima were harvested. As shown in the SEM images (Fig. 8b), the PLA stents were completely covered by the neointima, and the cell morphology on the inner surface of the neointima was randomly distributed. Both the RAPA- and (rhCol III/PDA-PEI)$_2$-modified PLA stents were completely covered by a layer of regularly arranged cells, with more cells on the surface of the (rhCol III/PDA-PEI)$_2$-coated PLA stents compared to the RAPA group. The cells presented an irregular state on the surface of the PLA, suggesting that

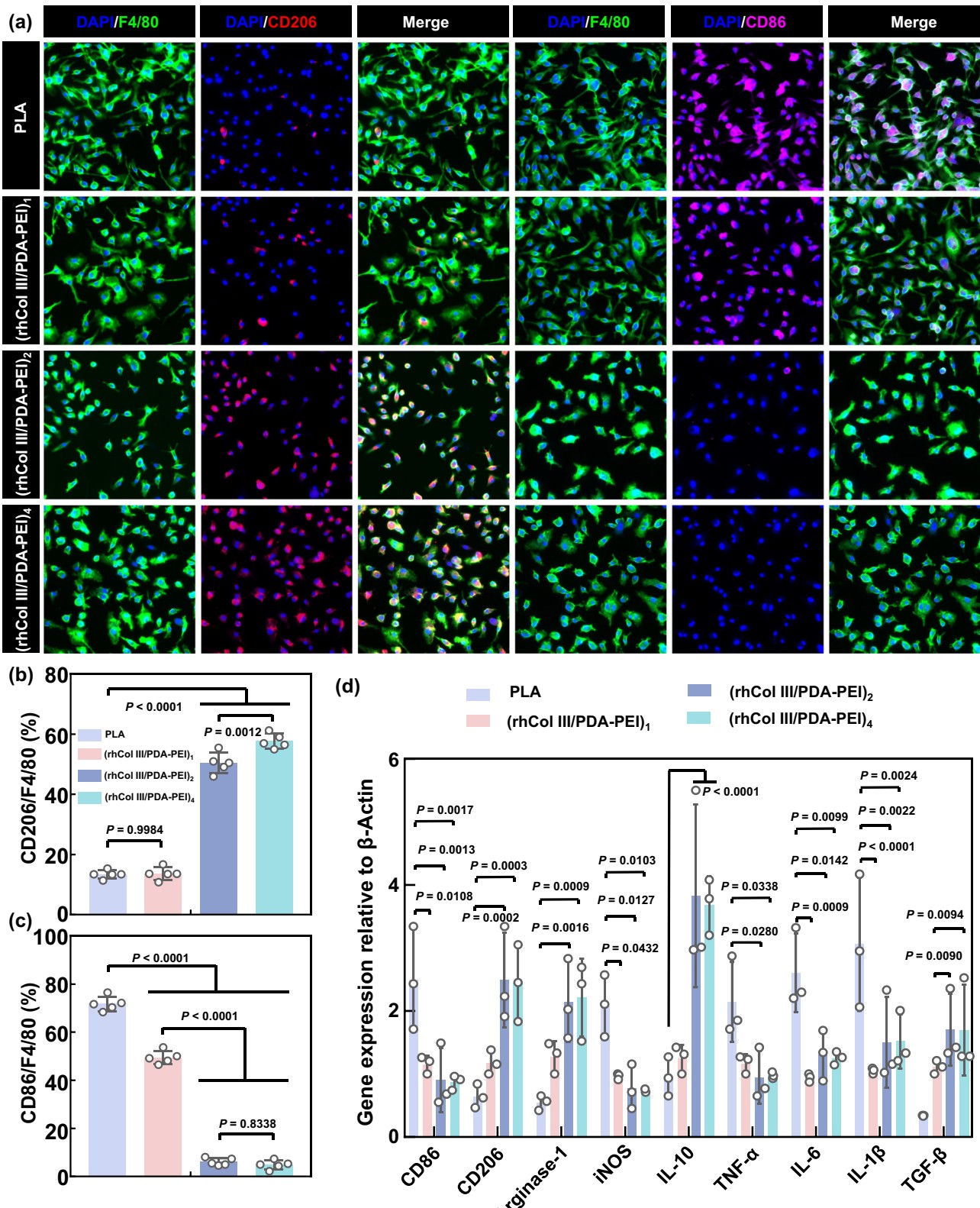

**Fig. 6 | (rhCol III/PDA-PEI)ₙ coatings (*n* = 0.5, 1, 2, and 4) reduce the inflammatory response and modulate the phenotype of MBMMCs. a** Representative immunofluorescence images of MBMMCs stained for F4/80 (All, green), CD206 (M2, red) and CD68 (M1, rose) on the surface of uncoated and (rhCol III/PDA-PEI)ₙ-coated PLA sheets (*n* = 1, 2, and 4) after 3 days of incubation at 37 °C. Scale bars, 100 μm. Quantification of the expression ratio of (**b**) the proinflammatory marker CD86/F4/80 and (**c**) the anti-inflammatory marker CD206/F4/80 after 3 days of incubation (*n* = 5 independent samples). **d** Quantification of the expression of representative cytokines (i.e., anti-inflammatory cytokines *CD206*, *Arginase-1*, *IL-10*,

and *TGF-β* and proinflammatory cytokines *CD86*, *iNOS*, *TNF-α*, *IL-6*, and *IL-1β*), validated by qRT-PCR arrays. Values in the PLA, (rhCol III/PDA-PEI)₂, and (rhCol III/PDA-PEI)₄ groups were normalized to that in the (rhCol III/PDA-PEI)₁ group (*n* = 3 independent samples). *p* values < 0.05 were considered statistically significant. One-way ANOVA with Tukey's multiple comparisons was used for the comparisons in (**b**) and (**c**). Two-way ANOVA with Tukey's multiple comparisons was used for the comparisons in (**d**). The data are presented as the mean ± SD (*p* values < 0.05 were considered statistically significant).

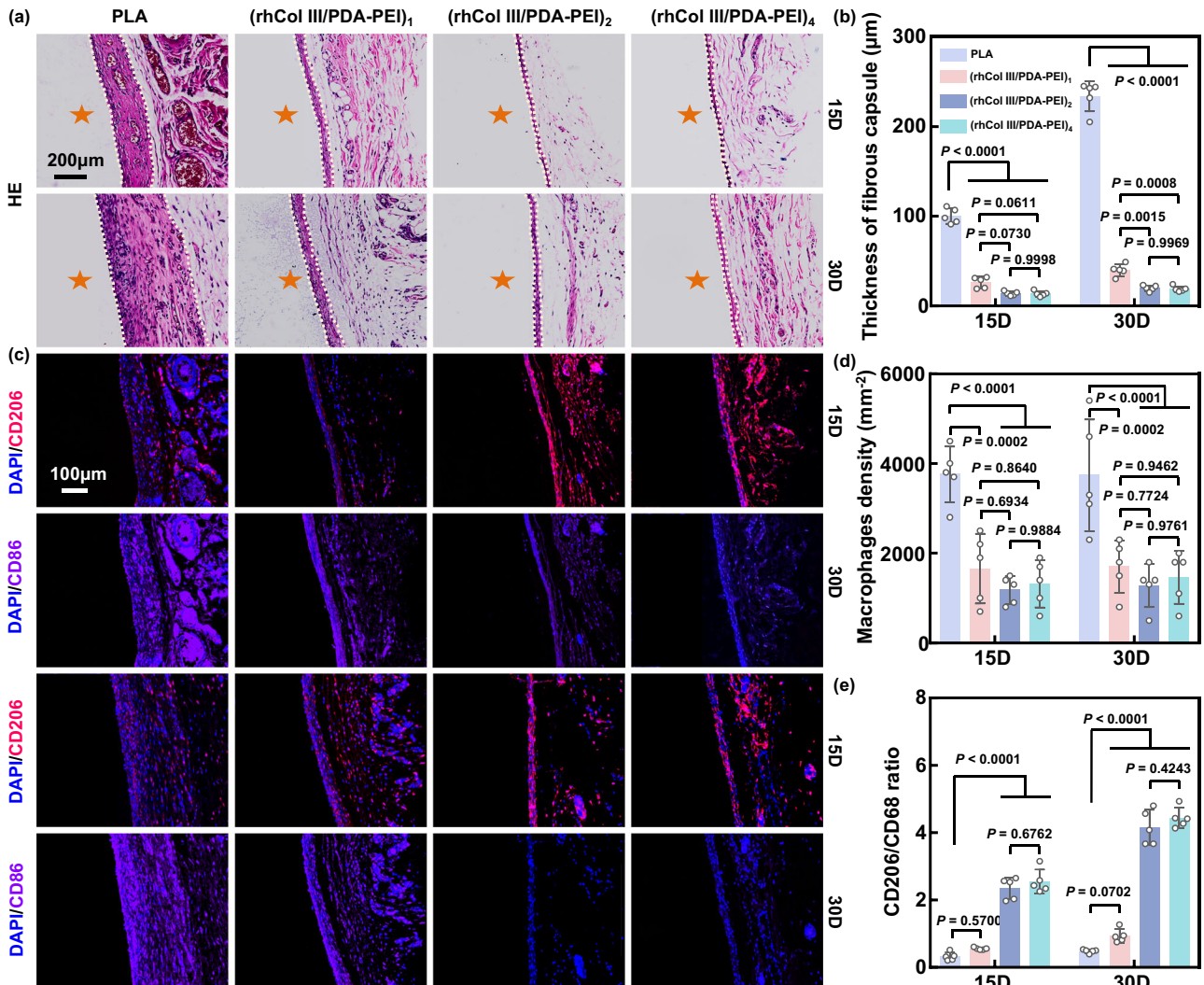

**Fig. 7 | In vivo (rhCol III/PDA-PEI)ₙ coatings (*n* = 0.5, 1, 2, and 4) reduce the inflammatory response in a rat model. a** Representative HE staining showing a thicker fibrous capsule in the PLA group than in the (rhCol III/PDA-PEI)ₙ groups (*n* = 1, 2, and 4) after subcutaneous implantation for 15 and 30 days. Scale bars, 200 μm. The fibrous capsule was labeled with pink dotted lines, and the PLA implants were marked by orange stars. **b** Quantification of the thickness of the fibrous capsules of the different samples depending on the H&E staining images ($n = 5$ independent samples in independent animals). **c** Representative immuno-fluorescence images of macrophages stained for CD206 (M2, red) and CD68 (M1, purple) from different groups. Scale bars, 100 μm. **d** Corresponding quantification of the numbers of macrophages and (**e**) CD206 cell/CD68 cell ratio ($n = 5$ independent samples in independent animals). Two-way ANOVA with Tukey's multiple comparisons was used for the comparisons in (**b**), (**d**), and (**e**). The data are presented as the mean ± SD (*p* values < 0.05 were considered statistically significant).

the surface of the neointima was composed of different cells and secreted extracellular matrix, while the cells on the surface of RAPA- and (rhCol III/PDA-PEI)₂-coated PLA stents were cobblestone-like, typical of endothelial cells[61]. Of note, the cells adhering to the (rhCol III/PDA-PEI)₂ group were denser and accompanied by a more mature phenotype with elongated morphology and a high degree of orientation compared to the RAPA group.

HE staining and immunohistochemical staining were conducted to verify the luminal stenosis rate, the inflammatory response, and the SMC phenotype of the neointima (Fig. 8c–e). The neointima formed around the bare PLA stents were thicker than those around the RAPA and (rhCol III/PDA-PEI)₂ stents (Fig. 8c). In addition, quantitative analysis (Supplementary Fig. 14) indicated that the diameters of PLA-based stents were almost identical and matched well with the reference value (2.75 mm). The bare PLA stents exhibited the most severe intimal hyperplasia in all groups, as evidenced by both a higher lumen stenosis rate (34.4 ± 0.51%) and area of the neointima (1.73 ± 0.041 mm²). Compared with the RAPA group, the restenosis rate and the area of the neointima of (rhCol III/PDA-PEI)₂ decreased from 31.5 ± 0.9% to

20.8 ± 0.56% and from 1.52 ± 0.040 mm² to 1.12 ± 0.037 mm², respectively, indicating stronger suppression performance of intimal hyperplasia (Fig. 8f, g).

The inflammation, proinflammatory, and anti-inflammatory degrees were scored semiquantitatively by CD68, CD86, and CD206 staining, respectively, to evaluate the inflammatory response. As shown in Fig. 8d, the inflammatory scores of RAPA-modified stents as determined by CD68 immunohistochemical images sharply decreased compared with those of the PLA stents, which was attributed to the anti-inflammatory properties of RAPA. In addition, low inflammation scores were also visible around the struts of (rhCol III/PDA-PEI)₂-coated stents (Fig. 8h). Moreover, no significant differences in the inflammation scores were observed between the RAPA group and the (rhCol III/PDA-PEI)₂ group. To further investigate the inflammation regulatory effects of these two groups, CD86/CD206 immunohistochemistry was performed (Supplementary Fig. 15a, d, e). The infiltration of CD206-positive cells was dramatically increased in both other groups compared to the PLA control group, especially in the (rhCol III/PDA-PEI)₂ group. Surprisingly, (rhCol III/PDA-PEI)₂ treatment downregulated the

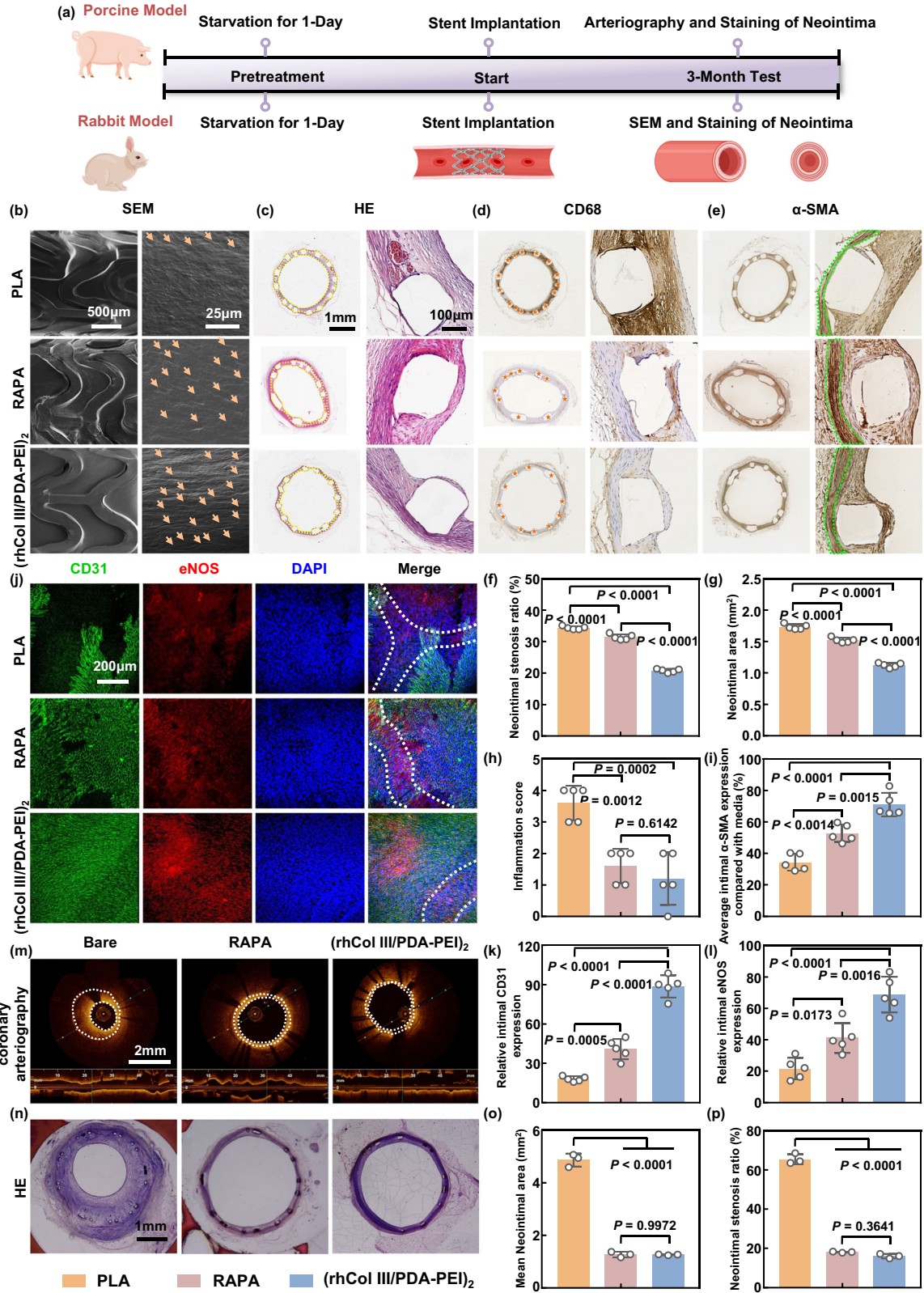

expression of CD86, as demonstrated by the scarcity of CD86-positive cells, in contrast to the numerous CD86-positive cells in the RAPA group. Taking these results together, it was reasonable to conclude that (rhCol III/PDA-PEI)$_2$ treatment favored M2 macrophage polarization and consequently supported endothelial healing.

Some studies have reported that SMCs exhibit a contractile phenotype in healthy mature organisms, which mainly maintains the

elasticity and contraction of blood vessels. However, SMCs with a contractile phenotype shift to a synthetic phenotype in some pathological states[4]. We used α-smooth muscle actin (α-SMA) and matrix metalloproteinase-2 (MMP2), indicators of contractile and synthetic SMCs, respectively, to evaluate the potential of intimal hyperplasia. Unexpectedly, the results of in vivo stent implantation assays demonstrated that the expression level of α-SMA-positive around the

**Fig. 8 | Stent implantation in rabbit and porcine models. a** Schematic representation and timeline of stent implantation in rabbit and porcine models of bare, RAPA- and (rhCol III/PDA-PEI)$_2$-modified stents (PLA and CoCr), and stented arteries were collected at 3 months. **b** Representative luminal surfaces of bare, RAPA- and (rhCol III/PDA-PEI)$_2$-modified PLA stents observed by SEM. Scale bars, 500 μm and 25 μm. The cobblestone-like typical endothelial cells were marked with orange arrows. Representative (**c**) HE staining, (**d**) CD68, and (**e**) α-SMA immunohistochemical staining of stented abdominal aorta 3 months after stent deployment in the rabbit model. Scale bars, 1 mm and 100 μm. The lumen area (inner area, IA) and external elastic lamina (EEL) on HE images were labeled with yellow dotted lines. Orange asterisks on CD68 images indicated stent struts. The media area on the SMA images was circled with green dashed lines. Quantitative analysis of (**f**) neointimal stenosis rate and (**g**) mean neointimal area (*n* = 5 independent samples in independent animals). **h** Inflammation scores in the strut-centered area and (**i**) α-SMA expression in the intima compared with the media (*n* = 5 independent samples in independent animals). **j** Representative immunofluorescence staining of the stented abdominal aorta in the rabbit model surface for CD31 (green), eNOS (red), and DAPI (blue). The struts of the stent are outlined in white dashed lines. Scale bars, 200 μm. Quantitative results of (**k**) the relative intimal CD31 expression and (**l**) the relative intimal eNOS expression of the abdominal aorta implanted with bare, RAPA-, and (rhCol III/PDA-PEI)$_2$-coated PLA stents in the rabbit model (*n* = 5 independent samples in independent animals). **m** Representative optical coherence tomography (OCT) of bare, RAPA- and (rhCol III/PDA-PEI)$_2$-modified CoCr stents after implantation of the coronary artery in the pig model for 3 months. The neointima was delineated in white on the OTC images. Scale bars, 2 mm. **n** Representative HE staining of stented coronary arteries 3 months after stent deployment in the pig model. Quantitative analysis of the (**o**) neointimal area and (**p**) neointimal stenosis rate, respectively (*n* = 3 independent samples in independent animals). One-way ANOVA with Tukey's multiple comparisons was used for the comparisons in (**f**–**i**), (**k**), (**l**), (**o**), and (**p**). The data are presented as the mean ± SD (*p* values < 0.05 were considered statistically significant).

(rhCol III/PDA-PEI)$_2$-coated PLA stent (71.02 ± 7.4 au) was higher than that around the bare PLA (34.39 ± 5.3 au) and RAPA-coated PLA stents (52.76 ± 5.6 au), revealing that (rhCol III/PDA-PEI)$_2$ could promote the contraction SMC phenotype and thus showed higher potential for suppressing intimal hyperplasia (Fig. 8e, i). We also observed more MMP2-positive cells in the control and RAPA groups compared with the (rhCol III/PDA-PEI)$_2$ group, which was strongly associated with continuous intimal hyperplasia (Supplementary Fig. 15c, f). Combined with the results of the SMC phenotype evaluation in vitro and in vivo, we can rationally conclude that the (rhCol III/PDA-PEI)$_2$ group favored the conversion of smooth muscle cells to a contractile phenotype, thereby maintaining vascular patency.

To further investigate the cell types growing on the neointima, immunofluorescence staining of CD31, eNOS, and DAPI was conducted (Fig. 8j). The progress of endothelialization in the bare PLA and RAPA-coated PLA groups was incomplete even after 3 months, implying delayed endothelialization. In stark contrast, (rhCol III/PDA-PEI)$_2$-coated PLA stents promoted the rapid restoration of endothelium, as evidenced by complete intact endothelium coverage and upregulated CD31 expression, and CD31-positive cells were highly elongated and closely aligned (Fig. 8k). The reason for the above phenomenon was that the RAPA-eluted stents inhibited the growth of SMCs and inhibited the adhesion and proliferation of ECs, resulting in immature intimal healing. Quantitative statistics of the expression of eNOS, which is key marker of endothelial cells, can reflect the degree of total endothelialization[59]. As shown in Fig. 8l, the eNOS signal was much lower on PLA (21.6%) and RAPA (41.2%) stents than on (rhCol III/PDA-PEI)$_2$ stents (68.8%). According to DAPI staining, the cell density on the surface of the PLA (1632 cells/mm$^2$) and RAPA (2917 cells/mm$^2$) stents was lower than that of the (rhCol III/PDA-PEI)$_2$ stents (3114 cells/mm$^2$) (Supplementary Fig. 16). To further connect the in vivo and in vitro findings, we performed five-color immunofluorescence staining of stented artery slices to investigate the expression of the abovementioned genes that regulate the growth of HUVECs in vivo. Surprisingly, high expression of XBP1, GATA3, CEACAM6, and CCL5 around struts was indeed observed in the PLA group with severe intimal hyperplasia, whereas such expression was significantly inhibited in the (rhCol III/PDA-PEI)$_2$ group with suppressed neointima hyperplasia (Supplementary Fig. 17). Collectively, these results suggested that the (rhCol III/PDA-PEI)$_n$ coatings favored the normal development of endothelialization in vivo and showed the potential to achieve ideal remodeling of blood vessels.

Our (rhCol III/PDA-PEI)$_2$ stent illustrated advantages over RAPA-eluted stents in rabbit abdominal aorta stenting experiments. Therefore, we hypothesized that the use of (rhCol III/PDA-PEI)$_2$-coated stents could be extended to large animal studies or even clinical trials. Thereafter, the (rhCol III/PDA-PEI)$_2$-coated CoCr stents were implanted in the coronaries of miniature pigs for 3 months to explore the safety

and effectiveness of the (rhCol III/PDA-PEI)$_2$ coating in vivo. According to the longitudinal cross-section images obtained by optical coherence tomography (OCT) (Fig. 8m), it could be concluded that the (rhCol III/PDA-PEI)$_2$ group displayed optimized suppression of intimal hyperplasia, as evidenced by the minimized rates of lumen loss across the segment of stented arteries. In contrast, severe intimal hyperplasia was present throughout the entire segment of stented arteries in the CoCr control group and at the proximal terminals in the RAPA group. The quantitative analysis determined by HE staining revealed that the diameters of CoCr-based stents (3 mm) at 3 months postimplantation matched well with the reference values provided by the manufacturer (Supplementary Fig. 18). The (rhCol III/PDA-PEI)$_2$-coated stent (1.26 ± 0.02 mm$^2$) markedly reduced the neointimal area compared with the bare stent (4.87 ± 0.2 mm$^2$), while there was no significant change compared with the RAPA-coated stents (1.28 ± 0.1 mm$^2$) (Fig. 8o). Taking the HE staining and OTC results into consideration, (rhCol III/PDA-PEI)$_2$-coated stents indicated the clinical feasibility of the (rhCol III/PDA-PEI)$_2$ coating for the healing of the neointima, as evidenced by the sustained patency of the entire stented arteries segment.

In conclusion, in the rabbit model, the (rhCol III/PDA-PEI)$_2$ coating reduced the inflammatory response by facilitating M2 macrophage polarization, accelerating the endothelialization process, and inhibiting the transition of SMCs from a synthetic to a secretory phenotype, thereby reducing restenosis and facilitating long-term stent patency. Hence, the (rhCol III/PDA-PEI)$_2$ coating exhibited a potential surface modification strategy to achieve ideal vascular neointimal healing. Surprisingly, in the porcine animal model, the neointima formed on the surface of the drug-free ((rhCol III/PDA-PEI)$_2$) stents was thinner than that formed on the commercially available RAPA stents from the perspective of the entire segment of stented arteries, which further increased the clinical interest in developing next-generation drug-free stents.

Despite its wide use in stent implantation, drug-eluting stents (DES) exhibit the issues such as poor endothelialization and a high risk of late/very late thrombosis and ISR. Herein, we proposed a one-produces-multi formulation: a drug-free cardiovascular stent functionalized with tailored recombinant humanized collagen type III (rhCol III), which resembles a multifunctional system that modulates vascular remodeling and promotes vascular neointimal healing, with the prospect of addressing the existing challenges in drug-eluting stents. In particular, compared to animal-sourced collagen, customized rhCol III exhibited great anticoagulant properties, anti-inflammatory activity, and cell adhesion activity. Therefore, the individual rhCol III component coating was expected to endow cardiovascular materials with versatility and meet the expected performance in stent applications. The (rhCol III/PDA-PEI)$_n$ coatings were robustly adherent and loaded with high levels of rhCol III to effectively suppress platelet adhesion/

activation and aggregation. Additionally, it could facilitate HUVEC adhesion and proliferation by influencing HUVEC-related signaling, inhibit the growth of HUASMCs by inducing a conversion to a contractile phenotype, and suppress the inflammatory response by promoting the polarization of macrophages toward the M2 phenotype and modulating the expression of inflammation-related proteins. Because of these synergistic effects, the one-produces-multi drug-free coating effectively suppressed neointimal hyperplasia in both rabbit and porcine models, and improved vascular neointimal healing in the rabbit model.

Future studies wound focus on optimized fabrication procedures for the potential translation of such single rhCol III component drug-free coatings from the laboratory to industrial applications. In addition, long-term follow-up studies in the porcine model are needed to fully understand the efficacy and safety of the (rhCol III/PDA-PEI)$_2$ stent compared to the commercially available rapamycin-eluting stents. Overall, the disclosed in vitro and in vivo data could effectively demonstrate the advantages of our one-produces-multi drug-free stent strategy. Given the excellent stability, multifunctionality and easy fabrication procedures, this rhCOL III-coated stent reduced in-stent restenosis in the porcine model, presenting great potential in opening an avenue for the development of next-generation vascular stents.

## Methods

### Ethical issues on animal experiments

All animal experiments were kept to a strict protocol approved by the medical ethics committee of Sichuan Province and conducted with the guidelines for the care and use of laboratory animals of Sichuan University (No. KS2020394).

### Materials

Dopamine hydrochloride (Dopa, CAS: 62-31-7), polyethyleneimine (PEI, $Mw = 25,000$, CAS: 9002-98-6), 2-(N-morpholino) 2-N-morpholino ethane sulfonic acid (MES, CAS: 1132-61-2), N-ethyl-N'-(3-dimethyl aminopropyl) carbodiimide hydrochloride (EDC, CAS: 25952-53-8), and N-hydroxysuccinimide (NHS, CAS: 7646-67-5) were purchased from Sigma–Aldrich (Gillingham, UK). Heparin sodium salt (185 USP units/mg, CAS: 9005-49-6), ampicillin sodium salt (CAS: 69-52-3), and pentobarbital sodium salt (Cat. No.: P0776) were purchased from Macklin Biochemical Co. Ltd. (Shanghai, China), Solarbio Science & Technology Co., Ltd. (Beijing, China), and TCI (Japan). The biological reagents for evaluating blood compatibility and cell compatibility were purchased from professional manufacturers and were mentioned in the "Experimental" section.

### Production of recombinant humanized collagen type III

A total of six peptides derived from human type III collagen (1466 amino acids) associated with platelet and vascular cell adhesion were screened (Supplementary Table S3). In conclusion, T16WTp (sequence: Ac-GERGAPGFRGPAGPNGIPGE-KGPAGERGAP-NH$_2$) was selected for the meticulous tailoring of recombinant humanized collagen type III (rhCol III, average Mw~45 KD) with a stable triple-helix conformation based on the requirements for cardiovascular implant devices, which was acquired by tandemly repeating 16 T16WTp with the use of advanced technologies including peptide synthesis and genetic engineering. The detailed production formulation of recombinant humanized collagen type III was described in the Supplementary Information (Supplementary Table 3).

### Preparation of (rhCol III/PDA-PEI)$_n$ coatings

Given the different variability in data collection methods, polylactic acid (PLA) sheets, PLA stents, silicon wafers, gold-plated quartz crystals, and CoCr stents were selected for the surface modification study. Of note, the construction procedure of the coating was identical on any substrate. In the case of PLA sheets, for example: before fabricating the (rhCol III/PDA-PEI)$_n$ coatings, PLA sheets were ultrasonically cleaned with acetone and ultrapure (UP) water in sequence three times (10 min ×3 times) and then dried with N$_2$ for 2 h before use. The PLA substrates were immersed in Dopa solution (2 mg/ml in a Tris buffer solution of pH 8.5) for 2 h, followed by cleaning with UP water three times (3 min × 3 times). Then, they were immersed in PEI solution (20 mg/ml in Tris buffer at pH 8.5) for 2 h, followed by cleaning with UP water three times (3 min × 3 times). The as-prepared coating was rich in amino groups for subsequent rhCol III grafting and was named PDA-PEI. The (rhCol III/PDA-PEI)$_n$ coatings were grafted onto the PDA-PEI-modified substrates via carbodiimide chemistry and electrostatic interactions. Briefly, the rhCol III (0.5, 1, 2, 4 mg/ml, respectively) was immersed in a solution of EDC (0.05 M) and NHS (0.05 M) in MES buffer (0.05 M, pH 5.40), and then the PDA-PEI-treated samples were immersed in the above solution for 1 h. After that, the pH of the abovementioned solution was adjusted from 5.4 to 7.4, and the reaction was continued for 8 h. The reason for adjusting the pH to 7.4 was to prevent the amino group from protonating and promote the reaction between the carboxyl groups of rhCol III and the amino groups on the surface. When the reaction was completed, the samples were washed with phosphate-buffered saline (PBS) three times (3 min × 3 times) and were denoted as (rhCol III/PDA-PEI)$_n$, where n refers to the concentration of rhCol III.

### Physiochemical characterization of (rhCol III/PDA-PEI)$_n$

The rhCol III loading and the stability of (rhCol III/PDA-PEI)$_n$ coatings prepared with FITC-labeled rhCol III on PLA sheets (1 cm × 1 cm, Sichuan Xingtai Pule Medical Technology Co., Ltd.) were observed by a confocal laser scanning microscope (Leica SP5, Germany). The microstructure, wettability, and surface elemental components of the (rhCol III/PDA-PEI)$_n$ coatings prepared on PLA sheets (1 cm × 1 cm) were investigated by scanning electron microscopy (SEM, FEI Nova NanoSEM 450), attention theta (Biolin Scientific, Sweden), and X-ray photoelectron spectroscopy (XPS, Thermo Scientific ESCALAB 250Xi, USA), respectively. XPS date were then analyzed with the curve-fitting program (CasaXPS, Version 2.3.17PR1.1). A spectroscopic ellipsometer (M-2000 V, J.A. Woollam, USA) measurement was performed to measure the thickness of the coatings prepared on silicon wafers (1 cm × 1 cm, Ningbo Yilin Semiconductor Technology Co., Ltd.). The zeta potentials of rhCol III and the (rhCol III/PDA-PEI)$_n$-coated PLA sheet (2 cm × 1 cm) were measured with a SurPASS 3 (Anton Paar, Austria) instrument.

### Real-time monitoring of the (rhCol III/PDA-PEI)$_n$ construction

The QSense Analyzer instrument (Biolin Scientific AB, Sweden) is a powerful tool to efficiently monitor the real-time change in the grafted molecule mass of the (rhCol III/PDA-PEI)$_n$ coating at room temperature. Briefly, the PDA-PEI film was predeposited on gold-coated quartz crystals (Φ 19.97 mm, QSX 301, Biolin Scientific AB, Sweden) as mentioned earlier before monitoring the construction process of the (rhCol III/PDA-PEI)$_n$ coating, and assumed to be a control group. Then, the reaction solution was passed into the chamber at a flow rate of 50 μl/min, and the concentration of the above solution was consistent with the experimental concentrations[25,60]. The data analysis was conducted with Dfind Smartfit modeling in QSense Dfind software (QSoft401).

### Mechanical stability testing of (rhCol III/PDA-PEI)$_n$

The (rhCol III/PDA-PEI)$_n$-modified PLA stents (Φ2.75 mm × 13 mm, Sichuan Xingtai Pule Medical Technology Co., Ltd) ($n = 0.5$, 1, 2, and 4) were prepared as mentioned above. Next, 2.75 mm diameter balloons were inserted into the inner lumen of the bare and modified stents, and then the stents were press-gripped. The (rhCol III /PDA-PEI)$_n$-modified PLA stents were expanded in serum solution at 37 °C for 40 s with inflation of the balloon (pressure value of 8 atm). Furthermore, the

dilated stents were placed in a commercially bare tube ($\Phi 5.3\,mm \times 130\,mm$, Tuyang Health Products Store) and assembled into an elastic hose of the peristaltic pump (Masterflex® L/S®, Cole-Parmer, USA) to complete the circuit, which was performed at 37 °C. The device provided a stable flow of blood serum of ~20 µl/min, and the blood serum was changed every day. After 1, 2, and 4 weeks, the samples were removed. Then, the retention of the FITC-labeled rhCol III signal after circulation was analyzed by a confocal laser scanning microscope (CLSM) (Leica SP5, Germany) and quantitatively calculated by ImageJ software. Fresh (rhCol III/PDA-PEI)$_2$ coating was used as the control, and the fluorescence intensity was set to 100%. The morphology of the (rhCol III /PDA-PEI)$_n$ coatings before and after the balloon dilatation experiment and after circulation for 4 weeks were investigated by SEM.

## In vitro platelet adhesion test

Blood was drawn from a healthy New Zealand White Rabbit and anticoagulated with sodium citrate. The (rhCol III/PDA-PEI)$_n$-coated ($n$ = 0.5, 1, 2, and 4) PLA sheets ($1\,cm \times 1\,cm$) were placed in 24-well cell culture plates and incubated with 500 µl PRP for 2 h at 37 °C. Subsequently, the samples were carefully washed with PBS to remove non-adherent platelets. After that, the samples were fixed with 2.5% glutaraldehyde overnight. Then, the in vitro blood compatibility of the (rhCol III/PDA-PEI)$_n$ coatings was evaluated by morphological observation of the adhered platelets using SEM.

## Ex vivo anti-thrombogenicity evaluation by arteriovenous shunt assay

Three adult New Zealand white rabbits (2.5–3 kg, ~3 months old) were used in this experiment[62], which were supplied by Chengdu Dossy Experimental Animals co., Ltd. (China). Before the test, the PDA-PEI and (rhCol III/PDA-PEI)$_n$ coatings ($n$ = 0.5, 1, 2, and 4) were prepared on polyvinyl chloride (PVC) tubes (F16, Guilong Medical Equipment Co., Ltd., China), and then connected in series using aspirator cannulas ($\Phi$ 1.2 mm, Guilong Medical Equipment Co., Ltd., China). The rabbits were anesthetized by injecting pentobarbital sodium (25 mg/ml, 0.7 ml/kg) through the ear vein. Then, the right carotid artery and left external jugular vein of the rabbits were carefully separated and pierced with indwelling needles ($\Phi$ 1.3 mm, B. Braun Melsungen AG, Germany). Afterward, the tandem uncoated, PDA-PEI and (rhCol III/PDA-PEI)$_n$-coated PVC tubes were connected with indwelling needles to form a closed-loop circuit with the vessels. After 2 h of blood circulation, the samples were quickly removed and washed carefully with PBS. The thrombus bound to the sample surfaces was collected, weighed, and photographed. The cross-section of the harvested tubes was photographed to analyze the occlusion rates of the samples.

## Cell culture

Primary human umbilical vein endothelial cells (HUVECs, Cat. No.: STCC12103) and primary human umbilical artery smooth muscle cells (HUASMCs, Cat. No.: HTX2180) were both purchased from Service-bio Technology Co. (China), and primary mouse bone marrow-derived macrophages (MBMMCs, Cat. No.: CP-M172) were obtained from Pro-cell Life Science & Technology Co., Ltd. (China). HUVECs were cultured in Roswell Park Memorial Institute 1640 (RPMI 1640, Cat. No.: 31870082, Gibco) medium supplemented with 10% (v/v) fetal bovine serum (FBS, Cat. No.: A3160901, Gibco) and 1% (v/v) penicillin-streptomycin (Cat. No.: 15640055, Gibco). HUASMCs were cultured in smooth muscle cell medium (SMCM, Cat. No.: 1101, ScienCell Research Laboratories) containing 2% (v/v) fetal bovine serum (FBS, Cat. No.: 0010), 1% (v/v) smooth muscle cell growth factor (SMCGS, Cat. No.: 1152), and 1% (v/v) penicillin–streptomycin (P/S, Cat. No.: 0503). MBMMCs were cultured in complete culture media for bone marrow mononuclear cells (Cat. No.: CM-M172, Procell Life Science & Technology Co., Ltd.). The cells were incubated in cell culture flasks at 37 °C in an incubator with 5% carbon dioxide until they grew to ~80%

confluence. The cells were then rinsed with PBS and treated with trypsin (0.05 w/v%)/EDTA (0.53 mM) solution (Cat. No.: 25300054, Gibco). Subsequently, the floating cells were collected and centrifuged, followed by resuspension in their medium for subculture or in vitro vascular cell compatibility experiments.

## Cell proliferation and morphology assay

The bare and (rhCol III/PDA-PEI)$_n$-coated PLA ($n$ = 1, 2, and 4) sheets ($1\,cm \times 1\,cm$) were placed in a 24-well cell culture plate (Cat. No.: 080723BH01, NEST Biotechnology Co., Ltd.) and sterilized with ethylene oxide at 37 °C for 12 h. Then, 1 ml of the suspension at a density of $2 \times 10^4$ cells/ml was seeded onto samples for 1, 3, 5, and 7 days. At the designated time, the morphology of cells adhered to the samples was investigated by fluorescence staining with fluorescein diacetate (FDA, CAS: 596-09-8, Merck), rhodamine-conjugated phalloidin (TRITC-Phalloidin, Cat. No.: CA1610, Solarbio Science & Technology Co., Ltd) and 4′,6-diamidino-2-phenylindole (DAPI, Cat. No.: C0065, Solarbio Science & Technology Co., Ltd). In the case of FDA staining, 500 µl of FDA solution with a concentration of 167 µg ml$^{-1}$ (dissolved in acetone) was added to the 24-well cell culture plate for 3 min. For TRITC-Phalloidin staining, cells were fixed with 4% formaldehyde solution for 15 min and permeabilized with 0.5% Triton X-100 solution for 5 min, followed by staining with phalloidin-TRITC (5 µg/ml in PBS) and DAPI dyes (10 µg/ml) for 30 and 5 min, respectively. After gentle washing with PBS three times, the cell morphology adherent to the samples was visualized by CLSM, and the cell counts were calculated using the ImageJ software (1.52a, NIH, USA). In addition, the proliferation of cells on the substrates was evaluated by the metabolic CCK-8 assay (a fresh basal medium and CCK-8 (9:1, v/v)), and the absorbance value was read at 450 nm[63].

## Antibodies

The primary antibodies used in this study included mouse monoclonal XBP1 (Cat. No.: sc-8015, Clone: F-4, Santa Cruz Biotechnology, USA, 1:50), mouse polyclonal CCL5 (Cat. No.: sc-365826, Clone: A-4, Santa Cruz Biotechnology, USA, 1:50), mouse monoclonal CEACAM6 (Cat. No.: sc-59899, Clone: 9A6, Santa Cruz Biotechnology, USA, 1:50), rabbit monoclonal GATA3 (Cat. No.: ab199428, Clone: EPR16651, Abcam, USA, 1:500), rabbit polyclonal F4/80 (Cat. No.: 29414-1-AP, Proteintech, China, 1:100), rabbit monoclonal CD68 (Cat. No.: ab283654, Clone: EPR23917-164, Abcam, USA, 1:100), rabbit polyclonal CD86 (Cat. No.: bs-1035R, Biosynthesis Biotechnology co., ltd, USA, 1:200), rabbit monoclonal CD206 (Cat. No.: 24595, Clone: E6T5J, Cell Signaling Technology, USA, 1:200), mouse monoclonal α-SMA (Cat. No.: ab7817, Clone: 1A4, Abcam, USA, 1:200), and rabbit monoclonal MMP2 (Cat. No.: 10373-2-AP, Clone: SB13a, Proteintech, USA, 1:200), mouse monoclonal CD31 (Cat. No.: ab9498, Clone: JC/70A, Abcam, USA, 1:200), and rabbit polyclonal eNOS (Cat. No.: ab5589, Abcam, USA, 1:100). The secondary antibodies used in this study included Alexa Fluor 488 goat anti-rabbit IgG (Cat. No.: A-11008, Invitrogen, USA, 1:500), Alexa Fluor 488 goat anti-mouse IgG (Cat. No.: A-11001, Invitrogen, USA, 1:500), and Alexa Fluor647 donkey anti-rabbit IgG (Cat. No.: A-31573, Invitrogen, USA, 1:500).

## Immunocytochemistry

After 3 days of culturing, the cells adhered to the surface of the round coverslips ($\Phi$ 24 mm, Biosharp Life Sciences, China) were first fixed and permeabilized as described in the above cell proliferation and morphology assay. Then, they were blocked with PBS solution containing 5% goat serum at 37 °C for 10 min. After that, primary antibodies were added and incubated with them overnight at 4 °C. After removal and rewarming for 30 min, the samples were washed gently with PBS solution and then incubated with secondary antibodies for 1 h at room temperature in the dark. Subsequently, the nuclei were stained with DAPI dyes for 5 min. All the abovementioned antibodies

and kits were used following the instructions of the manufacturer. Immunofluorescence images were obtained by a whole slide scanner (VS200, Olympus).

## Transcriptome analysis of HUVECs and HUASMCs

HUVECs and HUASMCs were seeded on bare and (rhCol III/PDA-PEI)$_2$-coated PLA sheets (2 cm × 2 cm) at a density of $2 \times 10^4$ cells/ml, as described in the above cell culture procedure. When the fusion rate of HUVECs and HUASMCs seeded on the surface of samples reached ~85%, total RNA was extracted using TRIzol reagent (Thermo Fisher Scientific, USA) according to the manufacturer's instructions. RNA integrity was monitored on agarose gels, followed by concentration and purity measurements using a Nanodrop 1000 (Thermo Fisher Scientific, USA)[64]. The quality of each library was assessed on an Agilent Bioanalyzer 2100 by performing quantitative PCR. RNA sequencing was conducted on an Illumina HiSeq 4000 by using the TruSeq SR Cluster Kit v3-cBot-HS (Illumina, Inc., USA). The whole process of RNA sequencing was carried out by Aksomics Biotech Co. Ltd (Shanghai, China). Solexa pipeline version 1.8 (off-line base caller software, version 1.8) was adopted to perform image processing and base recognition. FastOC software evaluates the sequencing quality of reads after removing adapters; Genome version used in reads mapping is Gen-Code GRCh37. The Ballgown R package was used to calculate expression differences between groups based on FPKM from genetic levels. Differentially expressed mRNAs (DEmRNAs) were determined by the Ballgown package under the conditions of fold change ≥1.2 and $p$ value ≤ 0.05. Volcano maps were generated using the ggscatter R package (1.18.0). Hierarchical bidirectional clustering analysis was performed to compare the expression patterns of different genes and different samples by using the heatmap R package. GO analyses were subjected to statistical calculations and graphing using the topGO software package in the R environment, and pathway analyses were calculated.

## RT–PCR array

After 3 days of cultivation, the RNA extraction of cell samples grown on PLA sheets (2 cm × 2 cm) was performed by using the TRIzol reagent as mentioned above, and the samples were carefully rinsed with RNase-free water (Cat. No.: A57775, Thermo Scientific, USA). Following the detection of RNA quality (concentration and purity) to the desired standard, the RNA was reverse transcribed into cDNA using the iScript cDNA Synthesis Kit (Bio-Rad, USA), which subsequently served as a template for PCR amplification. Levels of gene expression were analyzed using CFX96 instrument (Bio-Rad) with the addition of the AceQ qPCR SYBR green master mix (Vazyme) and specific primers. Relative gene expression was quantified using the $2^{-\Delta\Delta Ct}$ formula using Line Gene 9600 Plus Software (FQD-96A, Bio-Rad, Japan). To examine HUVEC growth behavior, we quantified the expression of *TMED2*, *DAPK3*, *PIP4K2C*, *PIK3R1*, *CXCL8*, *CEACAM6*, *CCL5*, *NFE2L2*, *DUSP6*, *LCN2*, *PEX2*, *HBEGF*, *CXCL3*, *CXCL2*, *XBP1*, *SDC1*, *GATA3*, and *SCHIP1*. For the characterization of different phenotypes of macrophages, we quantified proinflammatory factors such as *CD86*, *iNOS*, *TNF-α*, *IL-6*, and *IL-1β* and anti-inflammatory factors such as *CD206*, *Arginase-1*, *IL-10*, and *TGF-β*. Primers related to HUVEC growth behavior were shown in Supplementary Table 4; Primers related to macrophage polarization were shown in Supplementary Table 5.

## In vivo subcutaneous implantation

For in vivo experiments, all animals were purchased from Chengdu Dossy Experimental Animals Co., Ltd. Male Sprague Dawley rats (SD rats, ~10 weeks old) weighing ~250–300 g were used to assess subcutaneous implantation. Briefly, SD rats ($n = 6$) were anesthetized by intraperitoneal injection with pentobarbital sodium (20 mg/kg), and then the dorsal skin of the SD rats was carefully separated from the underlying muscles with hemostatic forceps. Then, the bare PLA and (rholII/PDA-PEI)$_n$-coated PLA sheets ($n = 1$, 2, and 4, 1 cm × 1 cm) were implanted into both sides of the subcutaneous pockets (two pockets on the back of each rat). After 15 and 30 days of implantation, the fiber capsules wrapped with the samples were harvested, and the SD rats were euthanized with an overdose of pentobarbital sodium salt solution. The wrapped sheets of harvested capsules were stained with hematoxylin and eosin (HE), Masson's trichrome, CD4 immunohistochemistry, and immunofluorescence (CD86 and CD206).

## Preparation of the coatings on the vascular stent

Two base stents were used in this study: PLA stents for the rabbit model and CoCr stents for the porcine model, and two types of bare stents were used as controls. The PLA stents (Φ2.75 mm × 13 mm) were provided by Sichuan Xingtai Pule Medical Technology Co., Ltd. with strut widths and thicknesses of 140 μm and 140 μm, respectively. The strut contact area to the artery was 16 mm². The CoCr stents (Φ3 mm × 17 mm) were supplied by a company with strut widths and thicknesses of 100 μm and 60 μm, respectively. The strut contact area to the artery was 25 mm². Rapamycin (RAPA) was directly attached to two types of bare stents to produce RAPA-coated PLA stents and RAPA-coated CoCr stents, both with a drug loading of 8 μg/mm. The fabrication procedure of the (rhCol III/PDA-PEI)$_2$ coating on the PLA and CrCo stents (rhCol III loading of 12 μg/mm) was consistent with the above mentioned in the "Experimental" section.

## Stent implantation in rabbit model

Eighteen healthy male healthy New Zealand white rabbits (~3 months old) weighing 2.5–3 kg were used in the in vivo vascular stent implantation experiments and were randomly divided into three groups ($n = 6$): bare, RAPA-coated and (rhCol III/PDA-PEI)$_2$-coated PLA, with the bare PLA stent acting as the control group. Before the experiment, uncoated, RAPA-, and (rhCol III/PDA-PEI)$_2$-coated PLA stents were sterilized with ethylene oxide at 37 °C for 12 h. The rabbits were under general anesthesia (25 mg/ml, 0.7 ml/kg) by injecting pentobarbital sodium through the ear vein. Then, the femoral artery was isolated for an adequate length (3 cm) to facilitate the delivery of the stent from the femoral artery to the abdominal aorta using a balloon. The stents were dilated and secured to the abdominal aortic segment under a balloon pressure of 8 atm for 40 s. The animals were injected intramuscularly with ampicillin sodium salt solution (10 mg/ml, 1 ml/kg) after withdrawal of the catheter and ligation of the artery. Additionally, aspirin (100 mg/kg) and clopidogrel (3 mg/kg) were administered for three continuous days after stent implantation. After 3 months of implantation, the stented aorta tissues were explanted and the rabbits were euthanized with an overdose of pentobarbital sodium salt solution. The harvested stented aortas tissues were equally divided into three segments immunofluorescence staining, immunohistochemical staining, and SEM observations.

## Stent implantation in porcine model

The experimental male small fragrant pigs (~35 kg, ~6 months old) were quarantined for 8 days before the trials to confirm that the animals were free of infectious diseases. Pigs were fed and administered aspirin (75 mg) and clopidogrel (100 mg) 12 h before stent implantation. Then, the pigs were injected with 200 IU/kg heparin intravenously in the lateral ear under aseptic conditions and were punctured from the right femoral artery. Subsequently, coronary angiography was conducted to confirm the location of stent placement with contrast agent (200 μg/pig). Then, three types of stents were randomly implanted into the three coronary arteries (right coronary artery, left anterior descending artery, and left circumflex artery) through a 6 F guide catheter under the guidance of a guide wire (0.014 inches). Finally, stents were dilated with a high-pressure balloon at 12 atm, tightening them onto the vessel for 30 s. After implantation of the stent, coronary angiography was performed again

to examine the patency and placement of the stent, for thrombus-free and appropriate placement. A continuous intramuscular injection of penicillin (20,000 u/ml, 1 ml/kg), together with aspirin and clopidogrel orally, was administered for 7 days after the procedure. At implantation for 3 months, the patency of the animal models was observed and recorded by optical coherence tomography (OCT) images, which were recorded by the RadiAnt DICOM Viewer (2020.2.3). The stented arteries ($n = 3$ pigs/stent group) were harvested for histological analysis (HE staining) and then the pigs were euthanized with an overdose of pentobarbital sodium salt solution.

### Histopathological analysis

The collected fiber capsules around PLA sheets and stented aorta (PLA and CoCr) tissues were fixed with 4% paraformaldehyde for 2 days. Then, the tissues were removed and successively dehydrated with a fully automatic dewatering machine (EXCELSIOR AS, Thermo, USA) and permeabilized. The obtained tissues with excess xylene were encapsulated in paraffin using an embedding apparatus (HISTOSTAR, Thermo, USA), and then 4 µm-thick sections were cut from the proximal, middle, and distant ends of the tissues individually by using a hard tissue microtome (HM 340E, Thermo, USA). A portion of the obtained sections was subjected to hematoxylin and eosin (HE) and Masson's trichrome staining. Additional sections were subjected to immunofluorescence staining and immunohistochemical staining via the following procedure: First, antigen retrieval was performed with antigen repair solution (Fuzhou Maixin Biotech Co., Ltd.) under different conditions (pH 9 EDTA in microwave high heat for 8 min for immunofluorescence, Cat. No.: MVS-0099; pH 6 Citric acid monohydrate for 40 min in a 97 °C water bath for immunohistochemistry, Cat. No.: MVS-0101). Then, they were blocked with PBS solution containing 5% BSA for 10 min. Then, they were incubated with primary antibodies overnight at 4 °C and then with reagents from an opal multicolor fluorescence immunohistochemistry kit (Cat. No.: NEL861001KT, Akoya Biosciences, USA) for 10 min. In contrast to immunohistochemistry, DAPI staining for 5 min were performed for immunofluorescence. CCL5, XBP1, CEACAM6 and GATA3 immunostaining images were obtained by PerkinElmer Vectra Polaris™ (Akoya, Vectra Polaris). Remaining immunostaining images were obtained by a whole slide scanner. The rest of the stented aortic tissue was further subjected to CD31 and eNOS immunofluorescence staining and observed with CLSM. With directly measured parameters of the cross-section of the slices such as lumen area (inner area, IA) and external elastic lamina (EEL), determined with ImageJ software, additional histomorphometric data was calculated, such as the neointimal stenosis rate (EEL-IA)/EEL and mean neointimal area (EEL-IA). Furthermore, the expression levels of α-SMA, MMP2, CD31, and eNOS were also analyzed by ImageJ software. Inflammation, proinflammatory, and anti-inflammatory factors were scored semiquantitatively by CD68, CD86, and CD206 staining, respectively. Grading was as follows: 0 = no inflammation, no inflammatory cells surrounding the strut; 1 = minimal inflammation, 0 < inflammatory cells ≤ 5 surrounding the strut; 2 = mild inflammation, 5 < inflammatory cells ≤ 10 surrounding the strut; 3 = moderate inflammation, 10 < inflammatory cells ≤ 20 surrounding the strut; 4 = severe inflammation, inflammatory cells > 20 surrounding the strut.

### Statistical analysis

GraphPad Prism (7. 04) was employed for the statistical analysis. One-way analysis of variance (ANOVA) with Tukey's multiple comparisons test was used for the comparison of more than two groups. Two-way ANOVA with Tukey's multiple comparisons was applied to determine the differences among different groups. The probability value was represented by $p$, and $p$ values < 0.05 were considered statistically significant. Data are expressed as means ± SD. The statistical methods used in each experiment and the corresponding statistical analysis results were listed in the figure legends[65,66].

### Reporting summary

Further information on research design is available in the Nature Portfolio Reporting Summary linked to this article.

### Data availability

The main data supporting the results of this study are available within the paper and its Supplementary Information. The raw HUVECs and HUASMCs transcriptome data have been deposited into the Genome Sequence Archive database. The raw HUVECs transcriptome data are publicly available under project "HRA005037 [https://ngdc.cncb.ac.cn/gsa-human/browse/HRA005037]" and the HUASMCs transcriptome raw data are publicly available under project "HRA005839 [https://ngdc.cncb.ac.cn/gsa-human/browse/HRA005839]". Any additional requests for information can be directed to, and will be fulfilled by, the corresponding author. Source data are provided with this paper.

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

## Acknowledgements

We thank Shanxi Jinbo Bio-Pharmaceutical Co. for assisting us in designing tailored recombinant humanized collagen type III (as-designed rhCol III) together, and Fei Chen, Chunjuan Bao, and Yang Deng from the Institute of Clinical Pathology, West China Hospital of Sichuan University for helping with histological staining processing, together with Hongying Chen from the Core Facilities of West China Hospital of Sichuan University for assisting with polychrome cytometric immunofluorescence staining. This study received funding from the National Natural Science Foundation of China (32101107 and 32371427 to L.Y.), CAMS Innovation Fund for Medical Sciences (2021-I2M-5-013 to Y.W.) and National Key Research and Development Programs, China (2022YFB3807303 to L.Y.).

## Author contributions

Y.W., L.Y., H.W., X.Z., and X.Y. designed and conducted the study; R.L. and L.L. assisted in the preparation and characterization of coatings; T.Z. assisted in analyzing the transcriptome data; K.H. and Y.Q. assisted in the ex vivo and in vivo experiments; H.W., L.Y., and Y.W. wrote and edited the manuscript; All authors contributed to the revision of the manuscript and were in agreement on the final version.

## Competing interests

The authors declare no competing interests.
