## [Peer Review File · Nature Communications]

A Drug-free Cardiovascular Stent Functionalized with Tailored Collagen Supports in-situ Healing of Vascular TissuesREVIEWER COMMENTS

Reviewer #1 (Remarks to the Author):

In this article, the authors reported the development of a drug-free coating (rhCol III/PDA-PEI)² using recombinant humanized collagen type III that can perform multiple functions such as resisting coagulation, reducing inflammatory responses, inhibiting the proliferation of smooth muscle cells, and accelerating neointimal healing via specific cell signaling pathways. The authors also compared the effectiveness of the drug-free coating approach with drug-eluting stent in reducing in-stent restenosis and improving vascular neointimal healing in both rabbit and porcine models. The study provides a new approach for stent development using a "one produces multi" strategy. Although the subject matter is intriguing, the article would benefit from addressing the following questions.

1. The novelty of the paper should be emphasized more. It seems that the whole manuscript is a small modification of the paper Biomaterials. 276, 121055 (2021) published by the author before.
2. Why PDA and PEI were selected for the construction of coating? Including a brief literature review of the materials used in the "Main" section could help readers better understand the development process.
3. The authors mentioned "meticulously tailored rhCol III" several times in the manuscript. A more detailed introduction and methods is needed, what is the difference between this rhCol III and the others? Why do the authors choose this rhCol III sequences?
4. Have there been any studies that report the use of rhCol III, PDA, and PEI as a drug-free coating for stents?
5. What was the control for the immobilization of rhCol III on an amino-rich surface (PDA-PEI coating) experiment (Fig. 2A)?
6. The mechanism of SMC inhibition should be supplemented or at least discussed by RNA-seq.
7. In vitro inflammatory response experiments, why do most of macrophage express M1 marker on the PLA and (rhCol III/PDA-PEI)¹ substrate? Is that indicated these two substrates show the proinflammatory effects?
8. What is the mechanical property of the rhCol III/PDA-PEI coating? The authors should show the SEM images of the morphology of the coating before and after balloon dilation test. The authors should also show SEM images of the coating after the flow experiment, say for 30 days to show the long term stability of the coating.
9. What was the control for the stability of (rhCol III/PDA-PEI)² coating under the flowing system with bovine blood serum experiment (Figs. 2B and 2C)?
10. In Fig. 2B, why (rhCol III/PDA-PEI)² was used for comparing results after one and two weeks, while (rhCol III/PDA-PEI)⁴ was used for the one-month comparison?
11. What was the control for the QCM-D experiment in quantifying the amount of rhCol III bounded to the PDA-PEI coating (Fig. 2M)?
12. In Fig. 3F, the thrombus formed on the surface of PDA-PEI was more severe compared to bare PLA. However, in Figs. 3H and 3I, the occlusion rates and thrombus weights of PDA-PEI were lower than those of bare PLA. Why is this the case?
13. Why were the cell viability experiments only conducted at 1 day and 3 days, and not at longer time points as shown in Fig. 4C?
14. Why was the PDA-PEI group excluded from the experiments in examining various factors, such as the adhesion and proliferation of endothelial cells, the adhesion and proliferation of SMCs, both in vitro and in vivo inflammatory response, in vivo inflammatory response in the rat model, in vivo endothelialization and neointima formation?
15. In the rat model experiment examining the in vivo inflammatory response, why were the samples implanted in different positions in the rats, as shown in Fig. 7C?
16. There are many kinds of stent used in vivo stent implantation. Author need provide more details with stent used in this part. Why do the authors choose the CoCr stents in miniature pig model rather than use the PLA based stents studied before?

17. Are there any limitations or potential issues related to the (rhCol III/PDA-PEI)₂ coating that should be addressed in future studies?

Minor points

1. The figure labelling is not well organized such as Fig 3I. In addition, the aspect ratio of sub-figures should be the same like the Figure 2F, 2M.
2. In Figure 6B caption, authors mentioned the staining of RAW 264.7 cells for 1 and 3 days, which is not consistent with the figure.
3. In many figure captions, the author just used "Fluorescence images" to describe the figure. The authors should revise the figure caption for better reading.
4. There are some wrong or unsuitable label in the figures, such as Figure 2B "(rhCol III/PDA-PEI)₄-One month", Figure 3CD, Figure 5C, etc.

Reviewer #2 (Remarks to the Author):

I co-reviewed this manuscript with one of the reviewers who provided the listed reports as part of the Nature Communications initiative to facilitate training in peer review and appropriate recognition for co-reviewers.

Reviewer #3 (Remarks to the Author):

In this study Wu et al. address whether functionalization of material surface using humanized collagen type III can affect platelets, EC, SMC and macrophages in vitro, and can reduce neointima formation in vivo. The authors demonstrate that PLA sheets can be functionalized with (rhoIII/PDA-PEI)_n, and that this coating reduced platelet adhesion and shape change, promotes EC adhesion and proliferation, but limits SMC adhesion and proliferation. In addition, the (rhoIII/PDA-PEI)_n coating limits macrophage adhesion and promotes their anti-inflammatory phenotype. Macrophage "intensity" and skin thickness was further quantified in the skin after subcutaneous implantation of sheets. Finally, also coated stents were implanted in rabbits and pigs, and the effects on neointima formation were analyzed, showing a reduction in neointima formation, and increased endothelial coverage, and lower inflammation but increased SMC "intensity". These results are interesting. Data presentation, however, needs to be improved, and I have the following comments:

Sheets and stents with different coatings have been generated. Photomicrograph of these should be shown.

The contrast in Figure 2B does not seem to be the same across the different time points. Alternative methods of quantification, including an internal control, would be more convincing.

The full list of all genes regulated should be provided in a Supplemental Table.

Different sources of ECs are stated in the Methods; it is unclear, which cell type was used for which of the experiments in Figure 4.

XBP1, GATA3, CEACAM6, and CCL5 are lower expressed in cells exposed to (rhoIII/PDA-PEI)₂, thus do not seem to be "activated" by the coating. Accordingly, the illustration in Fig. 4H is difficult to understand and does not seem to relate to the data presented in Fig. 4G. What is the relevance of these genes, are they also regulated in vivo or contribute to the phenotype of reduced neointima formation?

The SMC phenotype has not been studied in detail. In the in vivo experiments, the authors claim that the (rhoIII/PDA-PEI)_n coating affects the SMC transition from a synthetic to a secretory phenotype. This needs to be studied in more detail in vitro.

For further analyses of the macrophage phenotype, additional and more commonly used genes indicating macrophage phenotype should be assessed by qPCR analyses. It is surprising to see a strong increase in IL-10 in cell supernatants, despite a dramatic decrease in macrophage numbers.

What are the mechanistic underpinnings of the very different effects on the various cell types?

It is unclear, if truly 1x1 cm sheets were implanted subcutaneously for experiments presented in Figure 7? The skin above the implant was removed? Images in Figure 7A and B do not match (e.g. in the (rhoIII/PDA-PEI)₁-treated groups). Macrophages are located throughout the tissue and do not seem to be in direct contact with the sheet. The fluorescence intensity or contrast seems different in the different groups. How was the "intensity" quantified and what does this actually refer to? The authors could quantify the area populated by macrophages or macrophage cell numbers. In the (rhoIII/PDA-PEI)₁-treated groups there seem to be lymphatic infiltrates, but this was not quantified. The (rhoIII/PDA-PEI)₂-treated sample shows subcutaneous fat, but the area bordering the area where the sheet was implanted is not visible. These issues make it difficult to judge the response to the sheets. In addition, the fibrous encapsulation was not evaluated; fibrosis staining should be performed.

The general idea of the subcutaneous implant is not fully understandable, as the sheet will not be in contact with smooth muscle cells, endothelial cells, but only macrophages if an inflammatory reaction is promoted. Do the authors here primarily investigate if macrophages dictate the response of the sheet-overlying skin?

In vivo, rapamycin-coated stents have been tested. Effects of rapamycin on other readouts have not been included for comparison.

No information on stent fabrication is provided (dimensions, how the coating was performed, quality of the coatings, how rapamycin coating was performed, what is the control for the rapamycin-coated stents? Was rapamycin attached to PLA?). How were stents delivered to the animals?

Luminal surfaces of stents observed by SEM are provided (stents for which species, what the errors points to?)

While intimal areas between struts was chosen for representative images of the CD68 staining in PLA and rapamycin-coated stents, areas close to the strut was chosen for (rhoIII/PDA-PEI)₂-treated groups. No quantification of the CD68 staining was performed. It is unclear what the Inflammation score in Fig. 8H refers to and how it was calculated. The macrophage phenotype in vivo needs to be investigated in order to substantiate the claim that also the macrophage phenotype is important for the readout neointima formation.

Unlike in vitro, the authors also find an increase in α -SMA Intensity? What does the "Intensity" refer to? A quantification of the SMA-stained area would be informative. To further uncover whether SMC have dedifferentiated or retained a contractile phenotype, additional markers would need to be studied. To connect in vivo and in vitro findings, it would also be warranted to analyze SMC proliferation.

Neointima formation is unaltered in rapamycin-treated animals. Endothelialization is equal in rapamycin and (rhoIII/PDA-PEI)₂-treated rabbits. These findings are surprising and go against the hypotheses and clinical data described in the Introduction.

In the pigs, no differences in neointima formation between rapamycin and (rhoIII/PDA-PEI)₂-coated stents are seen. Why are the (rhoIII/PDA-PEI)₂-coated stents then superior to standard treatment? A full characterization of the neointima formation, at least endothelialization, macrophage and SMC content, would be informative.

n-numbers are lacking for some of the Figure panels; for all data presented in plots, the individual data points should be shown. In addition, the authors should clarify in the Figure legend, if "n" refers to animal numbers, technical replicates, biological replicates etc.

In general, the Results section should be shortened, and the Discussion expanded. English language requires revision.

Reviewer #4 (Remarks to the Author):

In this study, Wu et al. showed that a drug-free coating formulation functionalized with meticulously tailored recombinant humanized collagen type III (rhCol III) that performed "one produces multi" behavior in response to injured tissue following stent implantation. "one produces multi" coating has the properties of resisting coagulation, reducing inflammatory responses and inhibiting stent-in restenosis. The work is interesting and the conclusion is mainly supported by the data presented. However, the reviewer has a few comments that need to be addressed.

1. In Fig.6B, the authors claimed that fluorescent staining of the RAW 264.7 cells on uncoated and (rhCol III/PDA-PEI) n-coated PLA after 1 day and 3 days were captured. The reviewer did not observe the images of these two time points, please ask the author to provide an explanation. In addition, immunofluorescence assay is not difficult to quantify. It is desirable to consider the use of flow analysis to verify the pro-inflammatory or anti-inflammatory differentiation of macrophages.

2. It is well known that the inflammatory cells in the stent implanted segment of blood vessels come from monocytes in the blood. Activated macrophages do not have the ability of proliferation, and the Raw264.7 cell line used in this study is immortal. Is inhibition of macrophage proliferation in Raw 264.7 poorly considered? Should the authors consider using bone marrow-derived monocytes to evaluate the pro-inflammatory/anti-inflammatory capacity of the coating, while considering the infiltration capacity of macrophages? For the use of Raw 264.7 in the design of this project, the reviewer is seriously skeptical about the rigor of the work on inflammatory cells.

3. In Fig. 8, it can be seen by SEM or HE staining after section that there is a complete vascular endothelial cell layer at the position of scaffold filament in the PLA group. However, the vascular endothelium of PLA in the Fig. 8J group was incomplete, which was unreasonable. The reviewer hopes the authors can provide confocal images of en face immunofluorescence staining in a larger format to prove that this is not the result of human judgment selection.

4. The authors emphasize that the coating has a significant anti-inflammatory effect, so monochrome immunohistochemistry with CD68 in animals is not enough (Fig. 8D). The reviewer believes that a CD86/CD206 polychromatic immunohistochemistry or immunofluorescence can help authors better elucidate the anti-inflammatory phenotype of the coating.

5. Optical coherence tomography (OCT) of different samples after implantation of the coronary artery in the pig model is a very visual representation of the restenosis of the stent segment. The reviewer also hopes that the authors can provide the longitudinal cut diagram of OCT results of blood vessels in the stent segment, which the reviewer believes it will be more shocking and convincing.

6. In Fig.2B, the reviewer believes the text in the fourth fluorescence image, (rhCol III/PDA-PEI) 4 - one month, is a mislabel, the reviewer wishes the author can verify the original data and correct it. Perhaps in terms of experimental design, four weeks might be more appropriate than one month.

7. Previous literature reports that polylactic acid (PLA) scaffolds start degrading around 3 months and their degradation products, lactic acid, can induce vascular fibrosis, leading to severe complications in late implantation stages. In this study, the in vivo experiments were conducted for 3 months, but the impact of the coating on vascular remodeling during scaffold degradation was not observed. It would be of great research value and clinical significance if the rhCol III coating could effectively reduce restenosis risks associated with PLA scaffold degradation. It needs to take longer to observe in order to obtain reliable results.

8. The full name of the abbreviation "PLA" should be provided when it is first mentioned in the text,

rather than in the Methods section.

9. When conducting stability experiments using bovine serum, it is necessary to specify the environmental temperature in the methodology section.

10. The "Ex vivo blood circulation thrombogenicity test" needs a clear description of how the blood vessels were connected to the PVC tube. The authors should provide more specific details of the experimental procedures for replication by other researchers.

11. Some subsection titles in the Results section need modification to succinctly summarize the research findings. For example, "Hemocompatibility test" seems more appropriate as a title for a method rather than a result subsection.

Recommendation: Major revision.

Thanks a lot for your letter and the reviewers' comments concerning our manuscript entitled "One Produces Multi: A Drug-free Cardiovascular Stent Functionalized with Tailored Collagen Supports in-situ Healing of Vascular Tissues". These comments are very valuable and helpful for revising and improving our manuscript. We have read the comments carefully and made corresponding corrections which we hope to meet with the requirements for acceptance. The main corrections and the responses to the reviewer's comments are as follows:

Reviewer #1 (Remarks to the Author):

In this article, the authors reported the development of a drug-free coating (rhCol III/PDA-PEI)₂ using recombinant humanized collagen type III that can perform multiple functions such as resisting coagulation, reducing inflammatory responses, inhibiting the proliferation of smooth muscle cells, and accelerating neointimal healing via specific cell signaling pathways. The authors also compared the effectiveness of the drug-free coating approach with drug-eluting stent in reducing in-stent restenosis and improving vascular neointimal healing in both rabbit and porcine models. The study provides a new approach for stent development using a "one produces multi" strategy. Although the subject matter is intriguing, the article would benefit from addressing the following questions.

1. The novelty of the paper should be emphasized more. It seems that the whole manuscript is a small modification of the paper *Biomaterials*. 276, 121055 (2021) published by the author before.

Reply: Dear reviewer, thanks a lot for your comments! In our previous study published in the "Biomaterials" journal, an ECM-mimetic multilayer coating composed of hyaluronic acid and rhCol III by LBL assembly was prepared. The related results just initially showed the potential application of multifunctional rhCol III in blood-contacting biomaterials without further in-depth mechanistic investigation. More importantly, the issues such as long-term stability and complicated systems in the previous system discouraged any thought of further clinical investigation. Based on

these existing shortcomings, we prepared a new type of coating preparation system and developed a “one produces multi” drug-free coating by covalent bonding and electrostatic interaction. Collectively, the following are the novelties of the current study as compared to previous work: (1) Translational value: Given the excellent stability, the unitary system, and easy fabrication procedures, our study advanced the translation of rhCol III from the previous laboratory scale to the engineering scale. (2) Validation of the pre-clinical feasibility: Compared with the drug-eluting stent, this new type of rhCOL III-coated stent persistently reduced in-stent restenosis in the porcine model. Altogether, we foresee that this “one produces multi” drug-free strategy has the great potential to serve as a new inspiration for the development of next-generation vascular stent. We have emphasized the novelty of this paper in the “Discussion” part of the revised manuscript (**page 33, lines 765-770**).

2. Why PDA and PEI were selected for the construction of coating? Including a brief literature review of the materials used in the “Main” section could help readers better understand the development process.

Reply: Thank you for your valuable comments. To efficiently immobilize the rhCol III, the PLA substrates required to be pretreated to introduce reactive functional groups. Recently, mussel-inspired dopamine chemistry has been widely applied to surface modification of cardiovascular implant devices owing to universality, simplicity, chemical stability, and robust attachment capability ^{1, 2}. Especially, our group had reported a method of generating a mussel-mimicking amine-rich surface using dopamine and polyethyleneimine (PEI) for further modification, which provided a basis for further immobilization of molecules via the cyclization of the amino and carboxyl groups ³. Thus, the above pretreatment of the substrate with mussel-mimicking amine-rich coatings could contribute to the subsequent robust and efficient immobilization of rhCol III. Meanwhile, we have added a brief literature review on the selection of PDA and PEI for the construction of coating, and the corresponding brief descriptions in the “Main” section of the revised manuscript as suggested (**page 5, lines 121-126; page 6, line 127**).

3. The authors mentioned “meticulously tailored rhCol III” several times in the manuscript. A more detailed introduction and methods is needed, what is the difference between this rhCol III and the others? Why do the authors choose this rhCol III sequences?

Reply: Thanks a lot for your comments. As suggested, a more detailed description of rhCol III production formulation has been added in the **Supplementary Table S3** and

the “Methods” part of the revised manuscript (**page 34, lines 787-797**). Here is the reason for selecting rhCol III sequence in this study: Type III collagen is one of the major components of the extracellular matrix and plays an important role in maintaining the normal structure and function of vascular vessel. Furthermore, type III collagen is positively associated with tissue restoration. Of note, human type III collagen has been proven with the capacity to interact with cells and thus mediate cell adhesion, which could help regulate vascular cells behavior in situations of perturbed tissue injury ⁴. Given that the characteristics of type III collagen correlated closely with the requirements of cardiovascular implantable devices, we chose human type III collagen (1466 amino acids) for the subsequent customized design of rhCol III.

Here is the difference between this rhCol III and the others: Based on the requirements for cardiovascular implant devices, a total of six peptides derived from human type III collagen associated with platelet and vascular cell adhesion were first screened (**Supplementary Table S3**). T16WTp (sequence: Ac-GERGAPGFRGPAGPNGIP-GEKGPAGERGAP-NH₂) associated with no platelet affinity but favoring vascular cell adhesion was then selected for the meticulous tailoring of the recombinant humanized collagen type III (rhCol III, average Mw~45 KD) with a stable triple-helix conformation, which was acquired by tandemly repeating 16 T16WTp with the use of advanced technologies including peptide synthesis and genetic engineering. In sharp contrast to the others, this rhCol III was customized and designed by our group with specific sequences and biological functions based on the requirements, including enhanced cell adhesion activity, anticoagulant properties, good water solubility, and low inflammatory response.

4. Have there been any studies that report the use of rhCol III, PDA, and PEI as a drug-free coating for stents?

Reply: To the best of our knowledge, there were no studies reporting that rhCol III, PDA, and PEI were used as a drug-free coating for stents. The strong supporting evidence is as follows, i.e., the rhCol III was first meticulously tailored by our group with specific sequences and biological functions to satisfy the specific requirements of cardiovascular implants including excellent anticoagulation, enhanced cell adhesion activity, good water solubility, and low inflammatory response. Moreover, we prepared a single rhCol III component drug-free coating using mussel-mimicking amine-rich method ², independently of drugs, that could efficiently reduce in-stent restenosis and enhance vascular neointimal healing.

5. What was the control for the immobilization of rhCol III on an amino-rich surface

(PDA-PEI coating) experiment (Fig. 2A)?

Reply: Excuse for the confusion caused by our previous description of the controls. For

Fig. 2A, the bare PLA sheet was set as the control group. As shown in **Supplementary Fig. 1**, no fluorescence signals were observed in the bare PLA control and PDA-PEI groups (**page 6, lines 130-131**). Corresponding descriptions and results about the control group have been added in the “Results” part of the revised manuscript.

Fig. R1 | (i.e. Supplementary Fig. 1) Representative FITC fluorescence signals of bare, and PDA-PEI-coated PLA sheets. Scale bars, 500 μm .

6. The mechanism of SMC inhibition should be supplemented or at least discussed by RNA-seq.

Reply: Thank you for your great suggestion. We have supplemented the RNA-seq assay to further uncover the underlying mechanism of suppression effects of the drug-free coating on HUASMCs. As shown in **Fig. 5D**, 182 and 197 genes significantly up-regulated and down-regulated were observed in (rhCol III/PDA-PEI)₂ compared to the control group after 3 days of incubation with HUASMCs under the threshold treatments of $|\log_2\text{FC}| > 1.5$ and $P < 0.05$, respectively. Moreover, genomic enrichment analysis using the GO database revealed that there were mainly seven signaling pathways involved in the (rhCol III/PDA-PEI)₂ group that influenced the growth behavior of HUASMCs compared to the control PLA group (**Fig. 5E**), as follows: cell proliferation, cell adhesion, collagen-containing extracellular matrix, serine-type endopeptidase activity, immune response, cytoskeleton development, and regulation of nitric-oxide synthase activity. Heat map of differentially expressed genes (**Fig. 5F**) showing the up-regulation of MAPT⁵, MGP⁶, MMP7⁷, and CHI3L1⁸ in the PLA control group favored atherosclerosis. Furthermore, the PLA group further increased the expression of the pro-proliferative genes such as PI3⁹, IGFBP2¹⁰, and MMP7 and reduced the expression of the apoptosis gene DUSP15¹¹, which contributed to the proliferation of HUASMCs. Encouragingly, (rhCol III/PDA-PEI)₂ treatment induced the down-regulation of synthetic phenotypic HUASMCs markers MMP2 and MMP9¹², and the up-regulation of contractile phenotype HUASMCs markers VIM¹³ and

EGFR¹⁴, as compared to the control PLA group. The (rhCol III/PDA-PEI)₂ group also facilitated the relaxation of smooth muscle cells and thus prevented the abnormal proliferation of the stented artery, as indicated by the up-regulation of the GUCY1A2¹⁵ and LRRC26¹⁶ (page 18, lines 425-435; page 19, lines 436-446). In summary, the PLA group induced the proliferation and adverse development of HUASMCs. However, the (rhCol III/PDA-PEI)₂ group enhanced anti-proliferative and pro-apoptotic functions, which was beneficial for maintaining the blood vessel patency rate and provided evidence for the blood-contacting devices to achieve neointimal healing. The discussion has been added to the “Results” part of the revised manuscript (page 19, lines 450-454).

Fig. R2 | (i.e. Figs. 5(D)- 5(F)) (rhCol III/PDA-PEI)_n inhibits the adhesion and proliferation of HUASMCs via mediating the phenotypic switch. (D) Volcano plot showing differentially expressed genes in (rhCol III/PDA-PEI)₂ group as compared to the PLA control group. Down-regulated and up-regulated genes were colored in blue and red, respectively, at significantly differentially expressed thresholds $|\log_2FC| > 1.5$ and $P < 0.05$. (E) GO analysis of differentially expressed genes in HUASMCs. (F) Heat map of the differentially expressed genes in the PLA and (rhCol III/PDA-PEI)₂ groups with the thresholds $|\log_2FC| > 1.5$ and $P < 0.05$.

7. In vitro inflammatory response experiments, why do most of macrophage express M1 marker on the PLA and (rhCol III/PDA-PEI)₁ substrate? Is that indicated these two substrates show the proinflammatory effects?

Reply: The pro-inflammatory effects of PLA have been reported in our previous work

17 and other published works 18, 19, and Riemann et al. have reviewed the mechanism of the pro-inflammatory of PLA substrate²⁰. Degradation of PLA can release acidic by-products. Accordingly, the acidization of the microenvironment around implants induces local inflammation, evidenced by the massive adhesion of macrophages, higher expression of pro-inflammatory cytokines TNF- α (M1 marker), and lower expression of anti-inflammatory cytokine IL-10 (M2 marker). In addition, PEI layers have been reported to present pro-inflammatory effects, which may be attributed to the moderate wettability and the positive surface potential mediated by the high amounts of amino groups²¹. In the current study, the reason for (rhCol III/PDA-PEI)₁ also displaying pro-inflammatory properties might be attributable to incomplete coverage of the rhCol III on the surface of PDA-PEI, thus leading to the predominance of a strong inflammatory response induced by the exposed amines. To make our results more accessible, we have added explanations and references concerning the pro-inflammatory properties of PLA and (rhCol III/PDA-PEI)₁ in the “Results” part of the revised manuscript (**page 21, lines 495-496; page 22, line 497**).

8. What is the mechanical property of the rhCol III/PDA-PEI coating? The authors should show the SEM images of the morphology of the coating before and after balloon dilation test. The authors should also show SEM images of the coating after the flow experiment, say for 30 days to show the long term stability of the coating.

Reply: Thank you for your valuable suggestions. As suggested, mechanical property tests and long-term stability experiments have been conducted. The (rhCol III/PDA-PEI)_n coatings (n=0.5, 1, 2, and 4) exhibited favorable mechanical properties, as evidenced by remaining connected without cracks after dilatation, together with still visible particles with no clear difference versus before dilation (**Fig. 2E, Supplementary Fig. 3, and Supplementary Fig. 4**). Then, the dilated (rhCol III/PDA-PEI)_n-modified stents (n=0.5, 1, 2, and 4) were placed in a flowing system with bovine blood serum at 37 °C, simulating a complex and changeable physiological environment. After 4 weeks of circulation, relatively few particles were shed with no significant differences as compared to the correspondent fresh coatings (**Supplementary Figs. 5**). More details and discussions have been added in the “Results” and “Methods” parts of the revised manuscript (**Results: page 7, lines 155-166**) (**Methods: page 36, lines 851863; page 37, lines 864-868**).

Fig. R3 | (i.e. Supplementary Fig. 3) | Representative SEM images of the bare and (rhCol III /PDA-PEI)_n-coated PLA stents (n=0.5, 1, 2, and 4). Scale bars, 20 μm and 200 μm.

Fig. R4 | (i.e. Supplementary Fig. 4) | Representative SEM images of bare and (rhCol III /PDA-PEI)_n-coated PLA stents (n=0.5, 1, 2, and 4) after balloon dilation in PBS at 37°C. Scale bars, 20 μm and 200 μm.

Fig. R5 | (i.e. Supplementary Fig. 5) Representative SEM images showing the strut surfaces of the dilated stents coated with (rhCol III /PDA-PEI)_n (n=0.5, 1, 2, and 4) coatings after 4 weeks of circulation under the flowing system with bovine blood serum at 37°C. Scale bars, 20 μm and 200 μm.

9. What was the control for the stability of (rhCol III/PDA-PEI)₂ coating under the flowing system with bovine blood serum experiment (Figs. 2B and 2C)?

Reply: The control for the stability test was the fresh (rhCol III/PDA-PEI)₂ coating (**now page 7, lines 169-170**). We would like to separately clarify that we re-evaluated the stability of the (rhCol III/PDA-PEI)₂ coating after stent expansion with the consolidation of your suggestion in the above question 8 (**now Figs. 2F and Supplementary Figs. 6**).

Fig. R6 | (i.e. Figs. 2(F)) (F) Under the flowing system with bovine blood serum at 37°C, the representative fluorescence signal of the (rhCol III/PDA-PEI)₂ coating after balloon dilation, measured at one week, two weeks, and 4 weeks. Fresh (rhCol III/PDA-PEI)₂ coating was set as control. Scale bars, 500 μm.

Fig. R7 | (i.e. Supplementary Fig. 6) Quantification of fluorescence intensity after balloon dilation in PBS at 37°C, calculated by Image J software. The fluorescence intensity of fresh (rhCol III/PDA-PEI)₂ coating was set as 100% (n=5). One-way ANOVA was used for the comparisons. All error bars are mean ± s.d.

10. In Fig. 2B, why (rhCol III/PDA-PEI)₂ was used for comparing results after one and two weeks, while (rhCol III/PDA-PEI)₄ was used for the one-month comparison? **Reply:** Thank you very much for pointing this out. Following careful verification of the original data, “(rhCol III/PDA-PEI)₄ - one month” was mislabeled, which was generated from (rhCol III/PDA-PEI)₂ circulating for one month. We have corrected this error in the revised manuscript.

11. What was the control for the QCM-D experiment in quantifying the amount of rhCol III bounded to the PDA-PEI coating (Fig. 2M)?

Reply: Thank you very much for pointing this out. Considering the reliability of data collection, the gold-coated quartz crystal (Φ 19.97mm, Biolin Scientific AB, Sweden) was used as the substrate in the QCM-D experiment. Furthermore, the PDA-PEI film was pre-deposited on gold-coated quartz crystals (AT-cut/5 MHz) earlier before monitoring the construction process of the (rhCol III/PDA-PEI)_n coating and assumed as a control group. We have added information about the control group for the QCM-D experiment in the “Methods” part of the revised manuscript (**page 36, lines 843-846**).

12. In Fig. 3F, the thrombus formed on the surface of PDA-PEI was more severe compared to bare PLA. However, in Figs. 3H and 3I, the occlusion rates and thrombus weights of PDA-PEI were lower than those of bare PLA. Why is this the case?

Reply: In Fig. 3F, the poor resolution of the images may lead to a distorted judgment of the severity of thrombus formation on the PLA and PDA-PEI surfaces. To present our results more clearly, we have re-uploaded images with higher resolution. As shown in Fig 3F, a stable, growing thrombus was observed in the PLA group, which comprised a platelet-rich core, held together by very tight red blood cells-platelet contacts and by

a very dense network of thin fibrin fibers, with little porosity. Such thrombus may have sufficient stability to resist high-shear and arterial flow conditions and grow to occlude the vessel lumen or embolize whole or in large parts. In contrast, unstable thrombi comprising sparse red blood cells-platelet contacts and less dense fibrin mesh (composed of thicker, more porous fibers) were found on the PDA-PEI surface, which was more susceptible to the permeability of fibrinolytic proteins and dislodgement by the arterial flow. This process resulted in a shower of embolization of small fragments and failure of occlusive thrombus formation ²². Thus, the thrombus formed on the surface of bare PLA was more severe compared to PDA-PEI (**Fig. 3F**). Again, it was consistent with our results of higher occlusion rate (**Fig. 3H**) and weight of the thrombus (**Fig. 3I**) in the PLA group. Meanwhile, we have added a brief description and references in the “Results” part of the revised manuscript to exhibit our results more convincingly (**page 11, lines 263-266**).

Fig. R8 | (i.e. Fig. 3F) Representative SEM images of the thrombus formed on different samples after 2 h of blood circulation. Scale bars, 5 μm and 20 μm.

13. Why were the cell viability experiments only conducted at 1 day and 3 days, and not at longer time points as shown in Fig. 4C?

Reply: Thank you very much for pointing this out. 1 and 3 days are usually designed as time points to evaluate the early proliferative tendency of vascular cells on different samples, as reported in other studies ^{23, 24}. After reviewing more literature, we realized that cell culture assays with a longer time point were indeed of great significance for further assessing the potential of our “one produces multi” drug-free coating in promoting endothelial regeneration ^{25, 15}. Accordingly, we have supplemented the growth behavior results of HUVECs and HUASMCs with prolonged time points (5 and 7 days) as suggested, including fluorescence images and cell viability. The number of adherent HUVECs on the surface of (rhCol III/PDA-PEI)_n (n=1, 2, 4) persistently increased with the prolongation of the incubation time to 5 and 7 days, and even formed confluent monolayers, but not in the PLA group (**Figs. 4A-4C**). More detailed

discussions have been added in the “Results” part of the revised manuscript (**Results: page 13, lines 302-322**). In contrast to HUVECs, the counts of HUASMCs adhered to the surface of various groups did not show a continued increase with the prolongation of the incubation time to 7 days, but rather gradually decreased showing a signal of apoptosis (**Supplementary Figs. 9A and 9B**). The results of the CCK-8 assay also confirmed that the cell proliferation of the PLA group was significantly enhanced compared with those of the (rhCol III/PDA-PEI)_n groups after 5 and 7 days of culturing (**Supplementary Fig. 9C**) (**page 17, lines 401-406; page 18, lines 407-414**). The diverse growth trend of (rhCol III/PDA-PEI)_n (n=1, 2, and 4) coatings on HUVECs versus HUASMCs might be due to the different sensitivity and responsiveness that interact between the material surface/interface and cells (**page 18, lines 418-421**). More details and discussions have been added in the “Results” part of the revised manuscript.

Fig. R9 | (i.e. Left: Figs. 4 ((A), 4 (C), and 4 (E)); Right: Supplementary Figs. 9 (A)-9(C)) Summary of Representative fluorescence images, cell number, and cell viability of HUVECs and HUASMCs on the bare and (rhCol III/PDA-PEI)_n-coated PLA sheets after 5 and 7 days of culture. Scale bars, 200 μm (n=5). Two-way ANOVA was used for the comparisons in 4(C), 4(E), Supplementary Fig. 9 (B), and Supplementary Fig. 9 (C). All error bars are mean ± s.d.

14. Why was the PDA-PEI group excluded from the experiments in examining various factors, such as the adhesion and proliferation of endothelial cells, the adhesion and proliferation of SMCs, both in vitro and in vivo inflammatory response, in vivo

inflammatory response in the rat model, in vivo endothelialization and neointima formation?

Reply: As stated in the study, as a typical blood-contacting cardiovascular implant, the vascular stents are required to exhibit antithrombotic properties. Given that high thrombosis risk affects the endothelial process, exacerbates the inflammatory response, induces late thrombosis, and ultimately results in cardiovascular material failure. Taken together with the in vitro platelet adhesion and ex vivo arteriovenous shunt assays, it was concluded that PDA-PEI could induce severe thrombosis. Therefore, the subsequent characterization focused on (rhCol III/PDA-PEI)_n (n=1, 2, 4) with the exclusion of the PDA-PEI group. A description of the reasons for the exclusion of the PDA-PEI group has been added to the “Results” part of the revised manuscript (**page 11, lines 275-279**).

15. In the rat model experiment examining the in vivo inflammatory response, why were the samples implanted in different positions in the rats, as shown in Fig. 7C?

Reply: Excuse for the confusion caused by the insufficient description of subcutaneous implantation experiments. Subcutaneous implantation is a typical model for investigating inflammatory responses around the implant/tissue interface, where the location of implantation is the subcutaneous tissue on both sides of the back²⁶. To present the operation procedures more clearly, we re-provided the schematic diagram of subcutaneous implantation (**Supplementary Fig. 11**) and enriched the experimental procedures (**page 42, lines 1021-1034**). In this experiment, the procedure was as follows: Briefly, SD rats (n=6) were anesthetized by intraperitoneal injection with 10% chloral hydrate (0.003 mL/g), and then the dorsal skin of the SD rats was carefully separated from the underlying muscles with hemostatic forceps. After that, the bare PLA and (rhColIII/PDA-PEI)_n-coated PLA sheets (n =1, 2, and 4, 1 cm × 1 cm) were implanted into both sides of the subcutaneous pockets (two pockets on the back of each rat). After 15 and 30 days of implantation, the fiber capsules wrapped with the samples were harvested and fixed with 4% paraformaldehyde for 2 days for subsequent studies.

Fig. R10 | (i.e. Supplementary Fig. 11) Schematic diagram of the SD rat subcutaneous implantation model.

16. There are many kinds of stent used in vivo stent implantation. Author need provide more details with stent used in this part. Why do the authors choose the CoCr stents in miniature pig model rather than use the PLA based stents studied before?

Reply: As suggested. We have added detailed information on all the stents used in vivo stent implantation, including diameter, strut widths, thicknesses, and strut contact area to the artery to the “Methods” section of the revised manuscript (**page 43, lines 1037-1048**). Moreover, the responses for selecting CoCr stents in miniature pig model rather than the PLA-based stents were shown in the following points: (1) Due to the robust attachment capability of mussel-mimicking amine-rich coating (PDA-PEI) to a wide range of substrate materials, it allowed us to focus on exploring the rhCOL III-based “one produces multi” drug-free coating ignoring the influence from the substrate material (PLA or CoCr). (2) The predominantly marketed products from the company that we collaborated with were CoCr alloy stents. Based on the encouraged in vitro and in vivo results of our “one produces multi” formulation, they were very confident to compare it with rapamycin-eluting stents, which represent the gold standard for coronary stents, to validate the feasibility of the clinical application. To sum up, we believed that it was also meaningful to investigate the potential industrialization and clinical application of “one produces multi” drug-free coating even with the CoCr stents.

17. Are there any limitations or potential issues related to the (rhCol III/PDA-PEI)₂ coating that should be addressed in future studies?

Reply: Thanks for your comments. For future study, there are some concerns that should be addressed. Firstly, we will focus on optimized fabrication procedure for the potential translation of such single rhCol III component drug-free coating from laboratory to industrial applications. Secondly, long-term follow-up studies in the porcine model are required to fully understand the efficacy and safety of (rhCol III/PDA-PEI)₂ stent compared to commercially available rapamycin-eluting stents. We have also added a discussion of limitations or potential issues related to the (rhCol III/PDA-PEI)₂ coating in the “Discussion” section of the revised manuscript (**page 33, lines 761-765**).

Minor points

1. The figure labelling is not well organized such as Fig 3I. In addition, the aspect ratio of sub-figures should be the same like the Figure 2F, 2M.

Reply: Thank you for your advice. We have reorganized the figure labeling of Fig 3I and modified the aspect ratio of Figs. 2F and 2M sub-figures.

2. In Figure 6B caption, authors mentioned the staining of RAW 264.7 cells for 1 and 3 days, which is not consistent with the figure.

Reply: Sorry, there was a mistake during the previous submission. We have provided the results of RAW 264.7 cell staining for 1 and 3 days in the Response Material this time. We would like to separately clarify that we have used mouse bone marrow-derived monocytes alternatively to the RAW 264.7 cells to re-evaluate the in vitro inflammatory response after combining comments from other reviewers.

Fig. R11 |(rhCol III/PDA-PEI)_n inhibited the adhesion and activation of RAW 264.7 cells. (A) Representative FDA fluorescent staining of the RAW 264.7 cells on uncoated- and (rhCol III /PDA-PEI)_n coated-PLA sheets (n=1, 2, and 4) after incubation for 1 day and 3 days at 37 °C. Scale bars 500 μm. Quantification of (B) Cell viability and (C) Rate of increase of RAW 264.7 cells cultured on different samples (n=5). One-way ANOVA was used for the comparisons in (C). Two-way ANOVA was used for the comparisons in (B). All error bars are mean ± s.d.

3. In many figure captions, the author just used “Fluorescence images” to describe the figure. The authors should revise the figure caption for better reading.

Reply: We have revised the figure captions of all the “fluorescence images” mentioned in the manuscript with a detailed description.

4. There are some wrong or unsuitable label in the figures, such as Figure 2B “(rhCol III/PDA-PEI)₄-One month”, Figure 3CD, Figure 5C, etc.

Reply: Excuse for these mistakes. All of them have been corrected.

Reviewer #2 (Remarks to the Author):

I co-reviewed this manuscript with one of the reviewers who provided the listed reports as part of the Nature Communications initiative to facilitate training in peer review and appropriate recognition for co-reviewers.

Reviewer #3 (Remarks to the Author):

In this study Wu et al. address whether functionalization of material surface using humanized collagen type III can affect platelets, EC, SMC and macrophages in vitro, and can reduce neointima formation in vivo. The authors demonstrate that PLA sheets can be functionalized with (rhColIII/PDA-PEI)_n, and that this coating reduced platelet adhesion and shape change, promotes EC adhesion and proliferation, but limits SMC adhesion and proliferation. In addition, the (rhColIII/PDA-PEI)_n coating limits macrophage adhesion and promotes their anti-inflammatory phenotype. Macrophage “intensity” and skin thickness was further quantified in the skin after subcutaneous implantation of sheets. Finally, also coated stents were implanted in rabbits and pigs, and the effects on neointima formation were analyzed, showing a reduction in neointima formation, and increased endothelial coverage, and lower inflammation but increased SMC “intensity”. These results are interesting. Data presentation, however, needs to be improved, and I have the following comments:

1. Sheets and stents with different coatings have been generated. Photomicrographs of these should be shown.

Reply: Thanks a lot for your suggestion. Photomicrographs of PLA sheets with different coatings have been provided in the previously submitted Supplementary Information. As suggested, we have supplemented the photomicrograph of the PLA stents with different coatings this time. Corresponding results and discussion have been added in **Supplementary Figs. 2 and 3** and the “Results” part of the revised manuscript (**page 6, lines 134-142**).

Fig. R12 | (i.e. Supplementary Fig. 2) Representative scanning electron microscopy (SEM) images of bare and (rhCol III /PDA-PEI)_n-coated Poly (l-lactic acid) (PLA) sheets (n=0.5, 1, 2, and 4). Scale bars, 20 µm.

Fig. R13 | (i.e. Supplementary Fig. 3) Representative SEM images of the bare and (rhCol III /PDA-PEI)_n-coated PLA stents (n=0.5, 1, 2, and 4). Scale bars, 20 µm and 200 µm.

2. The contrast in Figure 2B does not seem to be same across the different time points. Alternative methods of quantification, including an internal control, would be more convincing.

Reply: Thank you so much for pointing this out. The fluorescence images were collected in different batches, which might lead to inconsistent contrast. Accordingly, we re-evaluated the stability of the (rhCol III/PDA-PEI)₂ coating under unified photograph parameters. Meanwhile, we should state that this experiment was evaluated after stent expansion after incorporating comments from other reviewers (**original Fig. 2B, now Fig. 2F**). Additionally, we also quantitatively calculated the amount of rhCol

III in different coatings by QCM test, a powerful tool to efficiently monitor the real-time change in the grafted molecules' mass. The loading amounts of rhCol III in (rhCol III/PDA-PEI)_n coatings (n=0.5, 1, 2 and 4) were 241.77 ng/cm², 324.36 ng/cm², 1117.06 ng/cm², and 1964.06 ng/cm², respectively (Fig. 2N), analyzed by Dfind Smartfit modeling in QSense Dfind software. We have included the above-mentioned results and discussions in the “Results” part of the revised manuscript (page 8, lines 200-203).

Fig. R14 | (i.e. Figs. 2(F)) (F) Under the flowing system with bovine blood serum at 37°C, the representative fluorescence signal of the (rhCol III/PDA-PEI)₂ coating after balloon dilation, measured at one week, two weeks, and 4 weeks. Fresh (rhCol III/PDA-PEI)₂ coating was set as control. Scale bars, 500 μm.

Fig. R15 | (i.e. Supplementary Fig. 6) Quantification of fluorescence intensity after balloon dilation in PBS at 37°, calculated by Image J software. The fluorescence intensity of fresh (rhCol III/PDA-PEI)₂ coating was set as 100% (n=5). One-way ANOVA was used for the comparisons. All error bars are mean ± s.d.

3. The full list of all genes regulated should be provided in a Supplemental Table. Reply: Thank you for your valuable suggestions. We have added the full list of regulated genes in the Supplementary Table S1 of the revised Supplementary Information.

Supplementary Table S1. Abbreviations and full names of genes labeled in the volcano plot

Abbreviation	Full name
DAPK3	Death-associated protein kinase 3

PIK3R1	Phosphoinositide-3-kinase regulatory
XBP1	X-box binding protein 1
CXCL8	C-X-C motif chemokine ligand 8
GATA3	GATA binding protein 3
CEACAM6	CEA cell adhesion molecule 6
CCL5	C-C chemokine ligand 5
NFE2L2	Nuclear factor, erythroid 2 like 2
JADE1	Jade family PHD finger 1
DUSP6	Dual specificity phosphatase 6
LCN2	Lipocalin-2
PEX2	Phosphatidylinositol-3,4,5-trisphosphate dependent Rac exchange
CSF2	colony-stimulating factor 2
IL11	Adipogenesis inhibitory factor
ATF3	Activating transcription factor 3
CXCL3	C-X-C motif chemokine ligand 3
CXCL2	C-X-C motif chemokine ligand 8
SDC1	Syndecan-1
AVL9	AVL9 homolog (S. cerevisiae)

4. Different sources of ECs are stated in the Methods; it is unclear, which cell type was used for which of the experiments in Figure 4.

Reply: We carefully checked the descriptions of the “sources of ECs” in the “Methods” part and did not observe the statement “different sources”. All primary human umbilical vein endothelial cells (HUVECs, Cat. No.: STCC12103) used in this study were of the same source and purchased from Service-bio Technology Co. Wuhan, China. Potentially, the two abbreviations (including “ECs” and “HUVECs”) for human umbilical vein endothelial cells might have caused some confusion. To present the results more rigorously, we have standardized the abbreviation of human umbilical vein endothelial cells to HUVECs in Fig. 4 of the revised manuscript.

5. XBP1, GATA3, CEACAM6, and CCL5 are lower expressed in cells exposed to (rhoIII/PDA-PEI)₂, thus do not seem to be “activated” by the coating. Accordingly, the illustration in Fig. 4H is difficult to understand and does not seem to relate to the data presented in Fig. 4G. What is the relevance of these genes, are they also regulated in

vivo or contribute to the phenotype of reduced neointima formation? Reply: With respect to your concerns, we would like to respond to one by one as follows: (1) Yes, XBP1, GATA3, CEACAM6, and CCL5 were not activated, as evidenced by low expression in the (rhCol III/PDA-PEI)₂ group. However, a further examination of the manuscript failed to discover a description that XBP1, GATA3, CEACAM6, and CCL5 were activated by the (rhCol III/PDA-PEI)₂ coating. Perhaps the statement “the (rhCol III/PDA-PEI)₂ coating might activate the related signaling of ECs to induce intima formation” in the original manuscript caused confusion. To avoid misleading, we have revised it to “Among them, CCL5, GATA3, XBP1, and CEACAM6 were involved in all the above-mentioned signaling pathways, indicating that they might be related to the regulation of HUVECs growth” (**page 14, lines 343-345**). Furthermore, relevant literature also confirmed the down-regulation of these genes associated with adverse cell growth behavior favored the normal development of HUVECs adhesion, migration, and proliferation²⁷⁻³⁰. More detailed discussions have been added in the “Results” part of the revised manuscript (**page 14, lines 335-343**). (2) Validation of genes involved in regulation: We supplemented PCR assay (**Fig. 4J**) and immunofluorescent staining (**Figs. 4K and Supplementary Fig. 8**) of HUVECs in vitro. The results showed that the expression of the above-listed four genes was significantly suppressed in the (rhCol III/PDA-PEI)₂ group compared with the PLA group, consistent with the above-mentioned results. More detailed discussions have been added to the “Results” part of the revised manuscript (**page 14, lines 345-352**). (3) To more clearly illustrate **Fig. 4H**, we refined the three potentially involved signaling pathways based on our transcriptome sequencing results, including PI3KR1/AKT, mTOR, and MAPK. More details and discussions have been added in the “Results” and part of the revised manuscript (**page 14, lines 353-354; page 15, lines 355-371**) (4) Validation of whether these genes are also regulated in vivo: We performed five-color immunofluorescence staining of slices of stented vessels to validate that these genes could also be regulated in vivo, including XBP1, GATA3, CEACAM6, CCL5, and DAPI. Surprisingly, high expression of XBP1, GATA3, CEACAM6, and CCL5 around struts was indeed observed in the PLA group with severe intimal hyperplasia, whereas such expressions were significantly inhibited in the (rhCol III/PDA-PEI)₂ group with suppressed neointima hyperplasia. Corresponding results and discussion have been added in **Supplementary Fig. 17** and the “Results” part of the revised manuscript (**page 29, lines 672-679**).

Fig. R16 | (i.e. Figs. 4(J)- 4(L)) (rhCol III/PDA-PEI)_n coatings (n=0.5, 1, 2, and 4) regulate HUVECs behaviors. (J) Quantification of the expression of representative genes in PLA versus (rhCol III/PDA-PEI)₂, validated by qRT-PCR arrays. Value in the PLA group was normalized to that in the (rhCol III/PDA-PEI)₂ group (n = 3). **(K)** Representative immunofluorescence staining of CCL5, GATA3, XBP1, and CEACAM6 of HUVECs on 3 days of culture. Scale bar, 50 μ m. **(L)** Schematic diagram of the potential three signaling pathways involved in the regulation of HUVECs behavior induced by the (rhCol III/PDA-PEI)₂ coating, including PI3KR1/AKT, mTOR, and MAPK. Two-way ANOVA was used for the comparisons in (J). All error bars are mean \pm s.d.

Fig. R17 | (i.e. Supplementary Fig. 17) Representative immunofluorescence staining of the stented abdominal aorta in the rabbit model surface for XBP1, GATA3, CEACAM6, and CCL5. Scale bars, 500 μm and 100 μm .

6. The SMC phenotype has not been studied in detail. In the *in vivo* experiments, the authors claim that the $(\text{rhCol III/PDA-PEI})_n$ coating affects the SMC transition from a synthetic to a secretory phenotype. This needs to be studied in more detail *in vitro*.

Reply: Thank you for your valuable suggestions. As suggested, the Quantitative real-time polymerase chain reaction (qPCR) array was conducted to study the phenotypic transformation of SMC *in vitro*. In brief, analysis of HUASMCs phenotype-related PCR arrays (**Figs 5G and 5H**) supported that the $(\text{rhCol III/PDA-PEI})_2$ treatment reduced the expression of synthetic markers MMP2 and MMP9, and increased contractile markers VIM and EGFR compared to the control PLA group. More details and discussions have been added in the “Results” and part of the revised manuscript (**page 19, lines 446-449**). To connect *in vivo* and *in vitro* findings, we also investigated the SMC phenotypic *in vivo*. In addition to the indicator of contractile SMC ($\alpha\text{-SMA}$) (**Fig. 8(E)**), we added Matrix metalloproteinase-2 (MMP2) as a synthetic SMC indicator to evaluate the potential of intimal hyperplasia. Unexpectedly, the results of *in vivo* stent implantation

assays demonstrated that the expression level of α -SMA-positive around the (rhCol III/PDA-PEI)₂-coated PLA stent (71.02 ± 7.4 au) was higher compared with the bare PLA (34.39 ± 5.3 au) and RAPA-coated PLA stents (52.76 ± 5.6 au), revealing that (rhCol III/PDA-PEI)₂ could promote the contraction SMC phenotype and thus showed higher potential for suppressing intimal hyperplasia (Figs. 8E and 8I). Compared with the (rhCol III/PDA-PEI)₂ group, we also observed more MMP2-positive cells in the control and RAPA groups, which was strongly associated with subsequent continuous intimal hyperplasia (Supplementary Figs. 15C and 15F). Combined with the results of the evaluation of the SMC phenotype in vitro and in vivo, we can rationally conclude that the (rhCol III/PDA-PEI)₂ group favored the conversion of smooth muscle cells to a contractile phenotype, thereby maintaining vascular patency. Corresponding results and discussion have been added in the Supplementary Fig. 15 and the “Results” part of the revised manuscript (page 28, lines 643-656).

Fig. R18 | (i.e. Figs. 5(G) and 5(H)) (rhCol III/PDA-PEI)_n inhibited the adhesion and proliferation of HUASMCs via mediating the phenotypic switch. PCR arrays displaying the expression of genes related to the (G) Synthetic phenotype and (H) Contractile phenotype of HUASMCs in the PLA and (rhCol III/PDA-PEI)₂ groups. Value in the (rhCol III/PDA-PEI)₂ group was normalized to that in the PLA group. Two-way ANOVA was used for the comparisons in (G) and (H). All error bars are mean \pm s.d.

Fig. R19 | (i.e. Supplementary Fig. 15(C) and 15(F)) (C) Representative MMP2 immunohistochemical staining of stented abdominal aorta 3 months after stent deployment in the rabbit model. Scale bars, 1 mm and 100 μm. (F) Quantitative results of MMP2 expression in the intima compared with the media (n=5). One-way ANOVA was used for the comparisons in (F). All error bars are mean ± s.d.

7. For further analyses of the macrophage phenotype, additional and more commonly used genes indicating macrophage phenotype should be assessed by qPCR analyses. Reply: Thank you for your suggestion. Combining the advice of other reviewers, we have replaced RAW264.7 with primary mouse bone marrow mononuclear cells (MBMNCs) to evaluate the inflammatory response in vitro. At the same time, we also supplemented the qPCR assay as suggested to further analyze the macrophage phenotypes by evaluating the gene expression of some commonly used representative cytokine panels (i.e., anti-inflammatory cytokine CD206, Arginase-1, IL-10, and TGF-β and pro-inflammatory cytokines CD86, iNOS, TNF-α, IL-6, and IL-1β). As expected, the expression of such factors was nicely correlated with CD86/CD206 expression in the immunofluorescence results, as indicated by markedly decreased expression of pro-inflammatory genes and significantly increased expression of anti-inflammatory genes in the (rhCol III/PDA-PEI)_n (n= 2, and 4) groups (**Fig. 6E**). More details and discussions have been added in the “Methods” and “Results” manuscript (**Methods: page 41, lines 1001-1008; page 42, lines 1009-1019**) (**Results: page 22, lines 506514**).

Fig. R20 | (i.e. Fig. 6. (E)) (rhCol III/PDA-PEI)_n reduced inflammatory response and modulated the phenotype of MBMNCs. (E) Quantification of the expression of a list of representative cytokines (i.e., anti-inflammatory cytokine CD206, Arginase-1, IL-10, and TGF-β and pro-inflammatory cytokines CD86, iNOS, TNF-α, IL-6, and IL-1β), validated by qRT-PCR arrays. Value in the PLA, (rhCol III/PDA-PEI)₂, and (rhCol III/PDA-PEI)₄ groups were normalized to that in the (rhCol III/PDA-PEI)₁ group (n=3). Two-way ANOVA was used for the comparisons in (E). All error bars are mean ± s.d.

8. It is surprising to see a strong increases in IL-10 in cell supernatants, despite a dramatic decrease in macrophage numbers. What are the mechanistic underpinnings of the very different effects on the various cell types?

Reply: Here are our responses to your concerns one by one as follows: (1) To more accurately show the number of cells, we quantified the density of macrophages present in the original manuscript. We discovered that the density of RAW 264.7 cells on the PLA surface was the highest (1540 ± 134 cells/mm²), which was approximately 3 folds higher than that of the (rhCol III/PDA-PEI)₄ group (517 ± 87 cells/mm²). Correspondingly, the IL-10 concentration in the supernatant increased from 200 ± 10.8 pg/mL to 263 ± 5.1 pg/mL with a ratio of 1.315, not a strong increase. Owing to the superior anti-inflammatory properties of the rhCol III, it was reasonable that fewer cells but more IL-10 were secreted in the (rhCol III/PDA-PEI)₄ group. (2) In an attempt to more rigorously demonstrate the results, we have replaced RAW264.7 with primary mouse bone marrow mononuclear cells (MBMNCs) after combining comments from other reviewers and performed a range of more comprehensive assays to evaluate inflammation response in vitro. More detailed descriptions and discussions have been added to the “Methods” and “Results” parts of the revised manuscript (**Methods: page**

41, lines 990-1008; page 42, lines 1009-1019) (Results: page 21, lines 483-496; page 22, lines 497-514). (3) The underlying mechanism for the distinct effects on different cell types was related to the tailored design of rhCol III. With respect to the inflammatory response, rhCol III presented a mild inflammatory response as the immune-terminal peptides at the two ends of the collagen molecule - the N-terminus and the C-terminus ^{31, 32} were removed, which was dramatically different from that of animal-derived collagen. Furthermore, rhCol III exhibited a high affinity for cell adhesion but not platelets, which was attributed to the retention of highly adhesive fragments (Gly-Glu-Arg (GER) and Gly-Glu-Lys (GEK)) and the bypassing of the hydroxyproline (O) sequence that may induce platelet adhesion and activation. To clarify our tailored rhCol III more clearly, we have added a detailed description of rhCol III production formulation in **Supplementary Table S3** and the “Methods” part of the revised manuscript (**page 34, lines 787-797**).

9. It is unclear, if truly 1x1 cm sheets were implanted subcutaneously for experiments presented in Figure 7? The skin above the implant was removed? Images in Figure 7A and B do not match (e.g. in the (rhCol III/PDA-PEI)₁-treated groups). Macrophages are located throughout the tissue and do not seem to be in direct contact with the sheet. The fluorescence intensity or contrast seems different in the different groups. How was the “intensity” quantified and what does this actually refer to? The authors could quantify the area populated by macrophages or macrophage cell numbers. In the (rhColIII/PDA-PEI)₁-treated groups there seem to be lymphatic infiltrates, but this was not quantified. The (rhCol III/PDA-PEI)₂-treated sample shows subcutaneous fat, but the area bordering the area where the sheet was implanted is not visible. These issues make it difficult to judge the response to the sheets. In addition, the fibrous encapsulation was not evaluated; fibrosis staining should be performed. The general idea of the subcutaneous implant is not fully understandable, as the sheet will not be contact with smooth muscle cells, endothelial cells, but only macrophages if an inflammatory reaction is promoted. Do the authors here primarily investigate if macrophages dictate the response of the sheet-overlying skin?

Reply: To more clearly present the general idea of subcutaneous implantation, we have summarized the following points: (1) Subcutaneous implantation model: Exactly as you stated, subcutaneous implantation is a classic model for investigating inflammatory responses around the implant/tissue interface ²⁵, whereas smooth muscle and endothelial cells were evaluated with alternative methods. Corresponding descriptions have been added in the “Results” part of the revised manuscript (**page 24, lines 541542**). (2) Procedure: Briefly, SD rats (n=6) were anesthetized by intraperitoneal

injection with 10% chloral hydrate (0.003 mL/g), and then the dorsal skin of the SD rats was carefully separated from the underlying muscles with hemostatic forceps. After that, the bare PLA and (rhCol III/PDA-PEI)_n-coated PLA sheets (n =1, 2, and 4, 1 cm × 1 cm) were implanted into both sides of the subcutaneous pockets (two pockets on the back of each rat). After 15 and 30 days of implantation, the fiber capsules wrapped with the samples were harvested. Corresponding descriptions have been added in the “Methods” part of the revised manuscript (**page 42, lines 1021-1034**). (3) Target subjects: Harvested fiber capsule tissue. (4) Evaluation of the inflammatory response: The inflammatory response of the implants was reflected by the thickness of the fibrous encapsulation and the number of infiltrating inflammatory cells (primarily including macrophages and lymphocytes).

Then, we would like to respond to your concerns one by one as follows: (1) As shown in **Fig. R21**, the size of the subcutaneous implant sheets was indeed 1 cm x 1 cm. The harvested tissues were fibrous capsules surrounding the samples, excluding the skin above. (2) The mismatch in Figs. 7A and B and the inconsistent fluorescence intensity were caused by the discontinuity of the slices (slices were cut from the proximal, middle section, and distant end of the fibrous capsule individually) and the fluorescence photos not collected with the same batch, respectively. To avoid the non-uniformity caused by the above-mentioned case, the remained fiber capsule tissues were continuously sliced and stained. After that, the fluorescence photographs were obtained under the unified parameters. (3) As you note, macrophages are indeed spread located throughout the tissue. However, the target subjects were fiber capsules in direct contact with the sheets, excluding the native tissue under the guidelines of evaluation of subcutaneous implantation. And, the observed subcutaneous fat was the result of some uncleared tissue outside the capsule. To show the target subjects more clearly, we have traced the capsules with lines and marked the positions of PLA sheets with different coatings with star patterns in Fig. 7A. (4) The mentioned “intensity” referred to the ratio of the sum of fluorescence intensity of the area of the fibrous capsules to its area, which was calculated by Image J software. (5) As suggested, we have quantified the macrophage cell numbers in the fibrous capsule. In this study, in addition to hemoglobin and eosin (HE) staining and immunofluorescence staining (CD86 and CD206), we also supplemented Masson's trichrome staining and CD4 immunohistochemical staining as suggested to evaluate the fibrous encapsulation and lymphocytic infiltration. Corresponding descriptions, results, and discussion have been added in Supplementary Figs. 11-13, “Methods”, and “Results” parts of the revised manuscript (**Methods: page 42, lines 1021-1034**) (**Results: page 24, lines 550-563**).

Fig. R21 | Photograph of the bare and (rhCol III /PDA-PEI)_n-coated PLA sheets (n=0.5, 1, 2, and 4) used for subcutaneous implantation.

Fig. R22 | (i.e. Fig. 7) **In vivo (rhCol III/PDA-PEI)_n reduce inflammatory response in the rat model.** (A) Representative HE staining showing a thicker fibrous capsule in the PLA group than in the (rhCol III/PDA-PEI)_n groups (n=1, 2, and 4) after subcutaneous implantation for 15 and 30 days. Scale bars, 200 μ m. The fibrous capsule was labeled with pink dotted lines, and the PLA implants were marked by orange stars. (B) Quantification of the thickness of the fibrous capsules of the different samples

depending on the H&E staining images (n= 5). (C) Representative immunofluorescence images of macrophages stained for CD206 (M2, red) and CD68 (M1, purple) from different groups. Scale bars, 100 μm . (D) Corresponding quantification of the numbers of macrophages and (E) ration of CD206 cells/CD68 cells (n=5). Two-way ANOVA was used for the comparisons in (B), (D), and (E). All error bars are mean \pm s.d.

Fig. R23 | (i.e. Supplementary Fig. 12) | Representative Masson's trichrome staining showing severe fibrosis in the PLA and (rhCol III/PDA-PEI)₁ groups compared to the (rhCol III/PDA-PEI)_n groups (n=2 and 4) after subcutaneous implantation for 15 and 30 days. Scale bars, 200 μm . The PLA implants marked by orange stars.

Fig. R24 | (i.e. Supplementary Fig. 13) | (A) Representative CD4 staining showing more lymphocytes in the PLA and (rhCol III/PDA-PEI)₁ groups compared to the (rhCol III/PDA-PEI)_n groups (n=2 and 4) after subcutaneous implantation for 15 and 30 days. Scale bars, 200 μm. The PLA implants marked by orange stars. (B) Corresponding quantification of the numbers of lymphocytes (n=5). Two-way ANOVA was used for the comparisons. All error bars are mean ± s.d.

10. In vivo, rapamycin-coated stents have been tested. Effects of rapamycin on other readouts have not been included for comparison.

Reply: Thank you so much for pointing this out. We have added the missing information on the effects of rapamycin-coated stents compared to bare or (rhCol III/PDA-PEI)₂-coated stents as presented below. With regard to the readouts in the rabbit model: (1) The cells adhering to the (rhCol III/PDA-PEI)₂ group were denser and accompanied by a more mature phenotype with elongated morphology and a high degree of orientation as compared to the RAPA group (**Fig. 8B**) (**page 26, lines 607-609**). (2) The bare PLA stents exhibited the most severe intimal hyperplasia in all groups, as evidenced by both higher the lumen stenosis rate ($34.4 \pm 0.51\%$) and the area of the neointima ($1.73 \pm 0.041 \text{ mm}^2$). Compared with the RAPA group, the restenosis rate and the area of the neointima of (rhCol III/PDA-PEI)₂ decreased from $31.5 \pm 0.9\%$ to $20.8 \pm 0.56\%$ and

from $1.52 \pm 0.040 \text{ mm}^2$ to $1.12 \pm 0.037 \text{ mm}^2$, respectively, indicating stronger suppression performance of intimal hyperplasia (**Figs. 8F and 8G**) (**page 27, lines 615-621**). (3) Nevertheless, no significant difference in inflammation scores was observed

between the RAPA group and the (rhCol III/PDA-PEI)₂ group (**Figs. 8D and 8H**). To further reveal the difference between these two groups in the inflammation-regulatory effects, CD86/CD206 immunohistochemistry was performed (**Supplementary Figs. 15A, 15D, and 15E**). Compared to the PLA control group, the infiltration of CD206-positive cells was dramatically increased in both other groups, especially for the (rhCol III/PDA-PEI)₂ treatment. Surprisingly, (rhCol III/PDA-PEI)₂ treatment adversely favored the expression of CD86, as demonstrated by the scarcely any CD86-positive cells, in contrast to the numerous CD86-positive cells in the RAPA group (**page 27, lines 628-637**). (4) Compared with the (rhCol III/PDA-PEI)₂ group, we also observed more MMP2-positive cells in the control and RAPA groups, which was strongly associated with subsequent continuous intimal hyperplasia (**Supplementary Figs. 15C and 15F**) (**page 28, lines 651-653**). With regard to the readings in the porcine model: (1) According to the longitudinal cut diagram of the optical coherence tomography (OCT) results (**Fig. 8M**), it can be concluded that the (rhCol III/PDA-PEI)₂ group displayed optimized suppression of intimal hyperplasia, as evidenced by the minimized rates of lumen loss across the segment of stented arteries. In contrast, severe intimal hyperplasia was present throughout the entire segment of stented arteries in the CoCr control group and at the proximal terminals in the RAPA group (**page 29, lines 687-693**).

11. No information on stent fabrication is provided (dimensions, how the coating was performed, quality of the coatings, how rapamycin coating was performed, what is the control for the rapamycin-coated stents? Was rapamycin attached to PLA?). How were stents delivered to the animals?

Reply: Thank you for your valuable suggestions. Two base stents were used in this study: PLA stents for the rabbit model and CoCr stents for the porcine model, and two types of bare stents were set as controls. For the PLA stents ($\Phi 2.75 \text{ mm} \times 13 \text{ mm}$), they were provided by Sichuan Xingtai Pule Medical Technology Co., Ltd. with strut widths and thicknesses at $140 \mu\text{m}$ and $140 \mu\text{m}$, respectively. The strut contact area to the artery was 16 mm^2 . For the CoCr stents ($\Phi 3 \text{ mm} \times 17 \text{ mm}$), they were supplied by a company with the strut width and thickness at $100 \mu\text{m}$ and $60 \mu\text{m}$, respectively. The strut contact area to the artery was 25 mm^2 . Rapamycin (RAPA) was directly attached to two types of bare stents to produce RAPA-coated PLA stents and RAPA-coated CoCr stents, both with drug loading of $8 \mu\text{g}/\text{mm}$. The fabrication procedure of the (rhCol III /PDA-PEI)₂

coating on the PLA and CrCo stents (rhCol III loading of 12 $\mu\text{g}/\text{mm}$) was consistent with the above-mentioned in the experimental section. We have included these in the “Methods” part (**page 43, lines 1037-1048**). Meanwhile, more detailed descriptions of the stent implantation procedures including rabbit and porcine models have been added in the “Methods” part of the revised manuscript (**page 43, lines 1050-1065; page 44, lines 1066-1089**).

12. Luminal surfaces of stents observed by SEM are provided (stents for which species, what the errors points to?) While intimal areas between struts was chosen for representative images of the CD68 staining in PLA and rapamycin-coated stents, areas close to the strut was chosen for (rhColIII/PDA-PEI)₂-treated groups. No quantification of the CD68 staining was performed. It is unclear what the Inflammation score in Fig. 8H refers to and how it was calculated.

Reply: Thank you for your valuable suggestion. Following your concerns, we would like to respond one by one as follows: (1) As stated in the text, SEM results (**Fig. 8B**) exhibited the neointima formed in bare-, RAPA- and (rhCol III/PDA-PEI)₂-modified PLA-stented arteries (**page 26, lines 596-598**). (2) Error of the stents: Quantitative analysis (**Supplementary Fig. 14 and Supplementary Fig. 18**) indicated that the diameters of both types of stents at 3 months post-implantation matched well with the reference values provided by the manufacturer (2.75 mm for PLA stents; 3 mm for CoCr stents). Corresponding descriptions and results about the stent error have been added in the “Results” part of the revised manuscript (**page 27, lines 613-615; page 29, line 693-695; page 30, line 696**). (2) After further review of the references, we realized that the quantification of immunohistochemistry results should be evaluated in the strut-centered area. Correspondingly, we modified the representative images of the CD68 staining in PLA and RAPA groups and re-calculated the inflammation score (**Fig. 8(D)**). (3) The “inflammation score” you referred to was the quantification of CD68 staining (**Figs. 8(D) and 8(H)**). The inflammation was scored semi-quantitatively by CD68 staining: grading was as follows: 0 = No inflammation, no inflammatory cells surrounding the strut; 1 = Minimal inflammation, $0 < \text{Inflammatory cells} \leq 5$ surrounding the strut; 2 = Mild inflammation, $5 < \text{Inflammatory cells} \leq 10$ surrounding the strut; 3 = Moderate inflammation, $10 < \text{Inflammatory cells} \leq 20$ surrounding the strut; 4= Severe inflammation, Inflammatory cells > 20 surrounding the strut. The above corresponding information has been added to the “Methods” and “Results” parts of the revised manuscript¹⁵ (**page 45, lines 1118-1122; page 46, lines 1123-1124**).

13. The macrophage phenotype in vivo needs to be investigated in order to substantiate

the claim that also the macrophage phenotype is important for the readout neointima formation.

Reply: Thank you for your valuable suggestions. The macrophage phenotype in vivo was discussed in question 10 above. In summary, no significant difference in total inflammation scores was observed between the RAPA group and the (rhCol III/PDA-PEI)₂ group. To further reveal the difference between these two groups in the inflammation-regulatory effects, CD86/CD206 immunohistochemistry was performed (Supplementary Figs. 15A, 15D, and 15E). Compared to the PLA control group, the infiltration of CD206-positive cells was dramatically increased in both other groups, especially for the (rhCol III/PDA-PEI)₂ treatment. Surprisingly, (rhCol III/PDA-PEI)₂ treatment adversely favored the expression of CD86, as demonstrated by the scarcely any CD86-positive cells, in contrast to the numerous CD86-positive cells in the RAPA group. Taking together CD68, CD86, and CD206 immunohistochemistry results, it is reasonable to conclude that the (rhCol III/PDA-PEI)₂ treatment favored M2 macrophage polarization and supported the consequently endothelial healing (page 27, lines 628-637).

Fig. R25 | (i.e. Supplementary Fig. 15) | Representative (A) CD86 and (B) CD206 immunohistochemical staining of stented abdominal aorta 3 months after stent deployment in the rabbit model. Scale bars, 1 mm and 100 μm. Quantitative results of

(D) Pro-Inflammation scores and (E) Anti-Inflammation scores as determined by CD86 and CD206 immunohistochemical images (n=5). One-way ANOVA was used for the comparisons in (D)-(E). All error bars are mean \pm s.d.

14. Unlike *in vitro*, the authors also find an increase in α -SMA Intensity? What does the “Intensity” refer to? A quantification of the SMA-stained area would be informative.

Reply: Excuse for the confusion caused by the absence of detailed annotation Y-axis of **Fig. 8I**. The “ α -SMA Intensity” mentioned in the stent implantation indicated the quantification of α -SMA expression in the intima (around struts) compared with the media, calculated by Image J software. We have labeled struts and media in Figs. 8(D) and 8(E), respectively. Quantification of the stained area by a uniform standard would be difficult as the area of the stent struts of each slice varied considerably. Additional pieces of literature have confirmed that the ratio of α -SMA expression “in the intima (around struts)” to “medium” as a quantification method was convincing³³. To make the quantification data of the immunostaining images more readable and not limited to the α -SMA, we have also enriched the annotation Y-axis of other images universally in the revised manuscript (**Fig. 8**). Such as: “ α -SMA Intensity” were revised to “Average α -SMA expression in the intima compared with the media”, “CD31 expression” was revised to “Relative intimal CD31 expression, and “eNOS expression” were revised to “Relative intimal eNOS expression”.

15. To further uncover whether SMC have dedifferentiated or retained a contractile phenotype, additional markers would need to be studied. To connect *in vivo* and *in vitro* findings, it would also be warranted to analyze SMC proliferation.

Reply: Thank you very much for your valuable comments. With respect to your concerns, we have discussed “whether SMCs have dedifferentiated or retained a contractile phenotype” in detail in question 6 above, which mainly involved the investigation of the SMC phenotype *in vitro* and *in vivo*.

16. Neointima formation is unaltered in rapamycin-treated animals. Endothelialization is equal in rapamycin and (rhoIII/PDA-PEI)₂-treated rabbits. These findings are surprising and go against the hypotheses and clinical data described in the Introduction. Reply: As stated in question 12 above, the strut-centered area should be evaluated in the investigation of the endothelialization process. Nevertheless, it is difficult to judge

whether the stent struts are included or not owing to the smaller format of the confocal images of CD31 immunofluorescence staining in the original manuscript. To more rigorously evaluate the endothelialization process, we have re-provided immunofluorescence images in a larger format containing the stent struts. Correspondingly, we re-quantified the expression of CD31 and eNOS based on the immunofluorescent images. As shown in **Fig. 8K**, the progress of endothelialization in bare PLA and RAPA-coated PLA groups was incomplete even after 3 months, implying delayed endothelialization. In stark contrast, (rhCol III/PDA-PEI)₂-coated PLA stents promoted rapid restoration of endothelium, as evidenced by complete intact endothelium coverage, increased CD31 expression, and CD31-positive cells were highly elongated and closely aligned (**Figs. 8K and 8L**). Meanwhile, we have revised the discussion in the corresponding part of the main text (**page 28, lines 658-663**).

Fig. R26 | (i.e. Figs. 8J, 8K, and 8L) | Stent implantation in rabbit and porcine models. (J) Representative immunofluorescence staining of the stented abdominal aorta in the rabbit model surface for CD31, eNOS, and DAPI. The struts of the stent are outlined in white dashed lines. Scale bars, 200 μm . Quantitative results of (K) the relative intimal CD31 expression and (L) the relative intimal eNOS expression of the abdominal aorta implanted with bare, RAPA-, and (rhCol III/PDA-PEI)₂-coated PLA stents in the rabbit model (n= 5). One-way ANOVA was used for the comparisons in (K) and (L). All error bars are mean \pm s.d.

17. In the pigs, no differences in neointima formation between rapamycin and (rh Col III/PDA-PEI)₂-coated stents are seen. Why are the (rhCol III /PDA-PEI)₂-coated stents then superior to standard treatment? A full characterization of the neointima formation,

at least endothelialization, macrophage and SMC content, would be informative. Reply: Thanks so much for your great comments. With respect to your concerns, we tried to respond as follows:

(1) Based on the collected data in the rabbit model, it can be concluded that the (rhCol III/PDA-PEI)₂ coating reduced the inflammatory response by facilitating the macrophage phenotypic transformation to M2, accelerated the endothelialization process, and inhibited the transition of SMCs from a synthetic to a secretory phenotype, thereby reducing restenosis and facilitating long-term stent patency. These findings provide supporting evidence for the successful development of drug-free coatings in the porcine model.

(2) Encouraged by our preliminary in vitro and in vivo results in the rabbit model, a company was interested in our “one produces multi” drug-free formulation and provided us with some funding for further validation of pre-clinical feasibility in the porcine model. Since they were more interested in-stent restenosis, merely OTC and hard tissue slice data were collected. Unfortunately, limited by the current deplasticization technique in hard tissue staining, we could not complement immunohistochemical and immunofluorescence staining of CoCr stented arteries such as CD31, CD68, and α -SMA. Subsequently, when we advance our drug-free coatings toward industrial application, a high-volume stent implantation in porcine models would be conducted. During this time, we would focus on a full characterization of the neointima formation, including endothelialization, macrophage, and SMC content as suggested.

(3) To demonstrate more powerfully the advancement of our (rhCol III/PDA-PEI)₂ coating in suppression of intimal hyperplasia, we have supplemented the longitudinal cut diagram of the OCT results 3 months after implantation (**Fig. 8M**), which is a very visual representation of the restenosis of the entire segment of stented arteries. Encouragingly, the (rhCol III/PDA-PEI)₂ group displayed optimized suppression of intimal hyperplasia, as evidenced by the minimized rates of lumen loss across the segment of stented arteries. In contrast, severe intimal hyperplasia was present throughout the entire segment of stented arteries in the CoCr control group and at the proximal terminals in the RAPA group (**page 29, lines 687-693**).

(4) From your statement, we have realized that the statement “Compared with the drug-eluting stent, the rhCOL III-coated stent reduced in-stent restenosis and improved vascular neointimal healing in both rabbit and porcine models” in the “Abstract” section was not rigorous, and it has been revised to “Compared with the drug-eluting stent, the rhCOL III-coated stent reduced in-stent restenosis in both rabbit and porcine models, and also improved vascular neointimal healing in rabbit model” (**page 2, lines 40-42**).

18. n-numbers are lacking for some of the Figure panels; for all data presented in plots, the individual data points should be shown. In addition, the authors should clarify in the Figure legend, if "n" refers to animal numbers, technical replicates, biological replicates etc.

Reply: Thanks for your advice. We have added the missing n numbers and clarified the indication of “n” in each figure legend. Additionally, we have re-uploaded the images with all data points presented.

19. In general, the Results section should be shortened, and the Discussion expanded. English language requires revision.

Reply: Thank you so much for your suggestions. We have concise the “Results” part and expanded the “Discussion” part with key conclusions, together with the flashpoints and limitations of the drug-free coating formulation. Additionally, the manuscript has been revised for language editing by Wiley Editing Service.

Reviewer #4 (Remarks to the Author):

In this study, Wu et al. showed that a drug-free coating formulation functionalized with meticulously tailored recombinant humanized collagen type III (rhCol III) that performed “one produces multi” behavior in response to injured tissue following stent implantation. “one produces multi” coating has the properties of resisting coagulation, reducing inflammatory responses and inhibiting stent-in restenosis. The work is interesting and the conclusion is mainly supported by the data presented. However, the reviewer has a few comments that need to be addressed.

1. In Fig.6B, the authors claimed that fluorescent staining of the RAW 264.7 cells on uncoated and (rhCol III/PDA-PEI)_n-coated PLA after 1 day and 3 days were captured. The reviewer did not observe the images of these two time points, please ask the author to provide an explanation. In addition, immunofluorescence assay is not difficult to quantify. It is desirable to consider the use of flow analysis to verify the pro-inflammatory or anti-inflammatory differentiation of macrophages.

Reply: Dear reviewer, thanks a lot for your comments! We have provided the results of RAW 264.7 cell staining for 1 and 3 days in the Response Material this time (**Fig. R27**). Integrated with your comments in question 2 below, we used mouse bone marrow-derived monocytes alternatively to the RAW 264.7 cells to quantify the pro-inflammatory polarization of macrophages with flow analysis, and the results were

shown in **Fig. 6D**. CD86-positive cells accounted for 40.58% and 38.08% in the PLA and (rhCol III/PDA-PEI)₁ groups (with slight amount rhCol III content), respectively, whereas only 17.65% and 19.77% of CD86-positive cells were observed in the (rhCol III/PDA-PEI)_n (n= 2, and 4) groups, respectively, which contained significant amount of rhCol III. We have included the relevant discussion in our revised manuscript (**page 22, lines 502-506**).

Fig. R27 |(rhCol III/PDA-PEI)_n inhibited the adhesion and activation of RAW 264.7 cells. (A) Representative FDA fluorescent staining of the RAW 264.7 cells on uncoated- and (rhCol III /PDA-PEI)_n coated-PLA sheets (n=1, 2, and 4) after incubation for 1 day and 3 days at 37 °C. Scale bars 500 μm. Quantification of (B) Cell viability and (C) Rate of increase of RAW 264.7 cells cultured on different samples (n=5). One-way ANOVA was used for the comparisons in (C). Two-way ANOVA was used for the comparisons in (B). All error bars are mean ± s.d.

Fig. R28 | (i.e. Fig. 6D) | (rhCol III/PDA-PEI)_n reduced inflammatory response and modulated the phenotype of MBMMCs. (D) Representative flow cytometry analysis of CD86-positive cells in different groups.

2. It is well known that the inflammatory cells in the stent implanted segment of blood vessels come from monocytes in the blood. Activated macrophages do not have the ability of proliferation, and the Raw264.7 cell line used in this study is immortal. Is inhibition of macrophage proliferation in Raw 264.7 poorly considered? Should the authors consider using bone marrow-derived monocytes to evaluate the pro-inflammatory/anti-inflammatory capacity of the coating, while considering the infiltration capacity of macrophages? For the use of Raw 264.7 in the design of this project, the reviewer is seriously skeptical about the rigor of the work on inflammatory cells.

Reply: Thanks a lot for this valuable suggestion. As suggested, the primary bone marrow mononuclear cells (MBMMCs) were used in this study to re-evaluate the *in vitro* inflammatory response by rhodamine-conjugated phalloidin (TRITC-Phalloidin) and (DAPI) fluorescence staining, immunofluorescence staining (F4/80, CD206, and CD68), flow cytometry analysis, and PCR arrays. As shown in **Supplementary Fig. 10**, MBMMCs adhered to the surfaces of PLA, and (rhCol III/PDA-PEI)₁ presented elongated and stretched shapes. Nevertheless, more rounded MBMMCs were observed in the (rhCol III/PDA-PEI)₂ and (rhCol III/PDA-PEI)₄ groups as the rhCol III loading increased. The morphology and polarization of macrophages are closely correlated as has been previously reported¹⁵. Thus, we first further investigated the polarization of MBMMCs via immunofluorescence staining, with markers F4/80, CD86, and CD206 characterizing all, M1 and M2 phenotypes of macrophages, respectively. In the (rhCol

III/PDA-PEI)_n (n= 2, and 4) groups, the expression of CD86 was significantly down-regulated, while the expression of CD206 was markedly up-regulated compared to that of the PLA and (rhCol III/PDA-PEI)₁ groups (**Fig. 6A**). In particular, both degradation products of PLA and exposed amines on the surface of (rhCol III/PDA-PEI)₁ (incomplete coverage by rhCol III) could lead to a strong inflammatory response ^{20, 21}. Correspondingly, quantitative statistical results also revealed that the proportion of M1 polarization decreased and the proportion of M2 polarization of macrophages increased in the (rhCol III/PDA-PEI)_n (n= 2, and 4) groups compared to the control PLA and (rhCol III/PDA-PEI)₁ groups (**Figs. 6B and 6C**). Then, flow cytometry analysis was employed to quantify the polarization of macrophages, and the results were shown in **Fig. 6D**. CD86-positive cells accounted for 40.58% and 38.08% in the PLA and (rhCol III/PDA-PEI)₁ groups (with slight amount rhCol III content), respectively, whereas only 17.65% and 19.77% of CD86-positive cells were observed in the (rhCol III/PDA-PEI)_n (n= 2, and 4) groups, respectively, which contained significant amount of rhCol III. At the same time, the gene expression of a list of representative cytokines (i.e., anti-inflammatory cytokine CD206, Arginase-1, IL-10, and TGF- β 3 and pro-inflammatory cytokines CD86, iNOS, TNF- α , IL-6, and IL-1 β) was identified by PCR arrays, to prove that differences in macrophage phenotypes mediated by the (rhCol III/PDA-PEI)_n coatings. As expected, the expression of such factors was nicely correlated with CD86/CD206 expression in the immunofluorescence results, as indicated by markedly decreased expression of pro-inflammatory genes and significantly increased expression of anti-inflammatory genes in the (rhCol III/PDA-PEI)_n (n= 2, and 4) groups (**Fig. 6E**). Overall, (rhCol III/PDA-PEI)_n (n=2 and 4) coatings with large amounts of rhCol III favored the transformation in macrophages to the M2-phenotype and the secretion of anti-inflammatory cytokines, generating a positive anti-inflammatory effect. Corresponding results discussions have been added in the “Results” and “Methods” parts of the revised manuscript (**Methods: page 41, lines 990-1008; page 42, lines 1009-1019**) (**Results: page 21, lines 483-496; page 22, lines 497-514**).

Fig. R29 | (i.e. Fig. 6) | (rhCol III/PDA-PEI)_n coatings (n=0.5, 1, 2, and 4) reduce inflammatory response and modulated the phenotype of MBMMCs. (A) Representative immunofluorescence images of MBMMCs stained for F4/80 (All, green), CD206 (M2, red) and CD68 (M1, rose) on the surface of uncoated and (rhCol III/PDA-PEI)_n-coated PLA sheets (n=1, 2, and 4) after 3 days of incubation at 37 °C. Scale bars, 100 μm. (B) Quantification of the expression ratio of the proinflammatory marker CD86/F4/80 and (C) Anti-inflammatory marker CD206/ F4/80 after 3 days of incubation (n=5). (D) Representative flow cytometry analysis of CD86-positive cells in different groups. (E) Quantification of the expression of a list of representative cytokines (i.e., anti-inflammatory cytokine CD206, Arginase-1, IL-10, and TGF-β and pro-inflammatory cytokines CD86, iNOS, TNF-α, IL-6, and IL-1β), validated by qRT-PCR arrays. Value in the PLA, (rhCol III/PDA-PEI)₂, and (rhCol III/PDA-PEI)₄ groups were normalized to that in the (rhCol III/PDA-PEI)₁ group (n=3). One-way ANOVA was used for the comparisons in (B) and (C). Two-way ANOVA was used for the comparisons in (E). All error bars are mean ± s.d.

3. In Fig. 8, it can be seen by SEM or HE staining after section that there is a complete vascular endothelial cell layer at the position of scaffold filament in the PLA group. However, the vascular endothelium of PLA in the Fig. 8J group was incomplete, which was unreasonable. The reviewer hopes the authors can provide confocal images of en face immunofluorescence staining in a larger format to prove that this is not the result of human judgment selection.

Reply: The complete “intimal layer” you observed in the SEM or HE staining at the position of scaffold filament in the PLA group is not indicative of an “endothelial cell layer”. The SEM images (**Fig. 8B**) displayed that although the PLA stents were completely covered by the neointima, the cell morphology on the inner surface of the neointima was randomly distributed, reportedly due to other types of cells or extracellular matrix secretions³⁴. whereas (rhCol III/PDA-PEI)₂-modified PLA stents were completely covered by a layer of regularly arranged cells with a cobblestone-like shape, typical of endothelial cells. The immunofluorescence staining of CD31 and eNOS was commonly employed to assess the extent of endothelialization. As suggested, the original image has been replaced by a larger format of the confocal images of en face immunofluorescence staining. As shown in **Fig. 8K**, the progress of endothelialization in bare PLA and RAPA-coated PLA groups was incomplete even after 3 months, implying delayed endothelialization. In stark contrast, (rhCol III/PDA-PEI)₂-coated PLA stents promoted rapid restoration of endothelium, as evidenced by complete intact endothelium coverage, increased CD31 expression, and CD31-positive cells were highly elongated and closely aligned (**Figs. 8K and 8L**). Thus, it is reasonable that complete coverage of the “intimal” layer was observed in the SEM or HE staining results, whereas incomplete “endothelial cell layer” in the immunofluorescence staining of CD31 and eNOS.

Fig. R30 | (i.e. Figs. 8J, 8K, and 8L) | Stent implantation in rabbit and porcine models. (J) Representative immunofluorescence staining of the stented abdominal aorta in the rabbit model surface for CD31, eNOS, and DAPI. The struts of the stent are outlined in white dashed lines. Scale bars, 200 μm. Quantitative results of (K) the relative intimal CD31 expression and (L) the relative intimal eNOS expression of the abdominal aorta implanted with bare, RAPA-, and (rhCol III/PDA-PEI)₂-coated PLA stents in the rabbit model (n= 5). One-way ANOVA was used for the comparisons in (K) and (L). All error bars are mean ± s.d.

4. The authors emphasize that the coating has a significant anti-inflammatory effect, so monochrome immunohistochemistry with CD68 in animals is not enough (Fig. 8D). The reviewer believes that a CD86/CD206 polychromatic immunohistochemistry or immunofluorescence can help authors better elucidate the anti-inflammatory phenotype of the coating.

Reply: Thank you for your valuable comments. We have added the CD86/CD206 immunohistochemistry to further reveal the difference between different groups in the inflammation-regulatory effects (Supplementary Figs. 15A, 15D, and 15E). Compared to the PLA control group, the infiltration of CD206-positive cells was dramatically increased in both other groups, especially for the (rhCol III/PDA-PEI)₂ treatment. Surprisingly, (rhCol III/PDA-PEI)₂ treatment adversely favored the expression of CD86, as demonstrated by the scarcely any CD86-positive cells, in contrast to the numerous CD86-positive cells in the RAPA group. Taking together these results, it was reasonable to conclude that the (rhCol III/PDA-PEI)₂ treatment favored M2 macrophage polarization and supported consequently endothelial healing.

Corresponding discussions have been added in the “Results” part of the revised manuscript (page 27, lines 630-637; page 28, lines 638-639).

Fig. R31 | (i.e. Supplementary Fig. 15) | Representative (A) CD86 and (B) CD206 immunohistochemical staining of stented abdominal aorta 3 months after stent deployment in the rabbit model. Scale bars, 1 mm and 100 µm. Quantitative results of (D) Pro-Inflammation scores and (E) Anti-Inflammation scores as determined by CD86 and CD206 immunohistochemical images (n=5). One-way ANOVA was used for the comparisons in (D)-(E). All error bars are mean ± s.d.

5. Optical coherence tomography (OCT) of different samples after implantation of the coronary artery in the pig model is a very visual representation of the restenosis of the stent segment. The reviewer also hopes that the authors can provide the longitudinal cut diagram of OCT results of blood vessels in the stent segment, which the reviewer believes it will be more shocking and convincing.

Reply: Thanks for pointing this out. Yes, as you stated, the extent of restenosis in the overall stented arteries segment can be visualized clearly from the longitudinal cut

diagram of OCT results, also contributing to a comprehensive evaluation of the ability of the different samples to reduce intimal hyperplasia. As suggested, we added the longitudinal cut diagram of OCT results in the revised manuscript. According to the longitudinal cut diagram of the optical coherence tomography (OCT) results (**Fig. 8M**), it can be concluded that the (rhCol III/PDA-PEI)₂ group displayed optimized suppression of intimal hyperplasia, as evidenced by the minimized rates of lumen loss across the segment of stented arteries. In contrast, severe intimal hyperplasia was present throughout the entire segment of stented arteries in the CoCr control group and at the proximal terminals in the RAPA group. Corresponding discussions have been added in the “Results” part of the revised manuscript (**page 29, lines 687-693**).

Fig. R32 | (i.e. Fig. 8M) | Stent implantation in rabbit and porcine models. (M) Representative optical coherence tomography (OCT) of bare, RAPA- and (rhCol III/PDA-PEI)₂-modified CoCr stents after implantation of the coronary artery in the pig model for 3 months. The neointima was delineated in white on the OTC images. Scale bars, 2 mm.

6. In Fig.2B, the reviewer believes the text in the fourth fluorescence image, (rhCol III/PDA-PEI)₄ - one month, is a mislabel, the reviewer wishes the author can verify the original data and correct it. Perhaps in terms of experimental design, four weeks might be more appropriate than one month.

Reply: Following careful verification of the original data, “(rhCol III/PDA-PEI)₄ - one month” was indeed mislabeled, which was generated from (rhCol III/PDA-PEI)₂ circulating for 30 days. Furthermore, we modified the experimental design with respect to the circulation time from 1 month to 4 weeks and re-executed the stability of the (rhCol III/PDA-PEI)₂ coating after stent expansion with the consolidation of other reviewer comments (**original Fig. 2B, now Fig. 2F**). Corresponding results and discussion have been added in the “Results” part of the revised manuscript (**page 7, lines 162-174**).

Fig. R33 | (i.e. Figs. 2(F)) (F) Under the flowing system with bovine blood serum at 37°C, the representative fluorescence signal of the (rhCol III/PDA-PEI)₂ coating after balloon dilation, measured at one week, two weeks, and four weeks. Fresh (rhCol III/PDA-PEI)₂ coating was set as control. Scale bars, 500 μm.

Fig. R34 | (i.e. Supplementary Fig. 6) Quantification of fluorescence intensity after balloon dilation in PBS at 37°, calculated by Image J software. The fluorescence intensity of fresh (rhCol III/PDA-PEI)₂ coating was set as 100% (n=5). One-way ANOVA was used for the comparisons. All error bars are mean ± s.d.

7. Previous literature reports that polylactic acid (PLA) scaffolds start degrading around 3 months and their degradation products, lactic acid, can induce vascular fibrosis, leading to severe complications in late implantation stages. In this study, the in vivo experiments were conducted for 3 months, but the impact of the coating on vascular remodeling during scaffold degradation was not observed. It would be of great research value and clinical significance if the rhCol III coating could effectively reduce restenosis risks associated with PLA scaffold degradation. It needs to take longer to observe in order to obtain reliable results.

Reply: Thank you for your valuable comments. The core of this study is focused on investigating the properties of drug-free coatings in suppressing the ISR and modulating vascular neointimal healing. We highly agree with you that longer observation would be needed to investigate the impact of our drug-free-coated stents on vascular remodeling during PLA stents degradation. PLA stents could provide mechanical support properties up to 9 months after implantation, while completely lose their support properties after 2 years and subsequently continue to degrade until they are

completely absorbed ^{36, 37}. Currently, we are in the process of discussing with the corresponding company to advance a batch of minipig stent implantation experiment up to 3 years to validate the feasibility of our drug-free coatings in clinical applications. If it can be implemented, we would incorporate your suggestions by establishing a series of data collection time points.

8. The full name of the abbreviation "PLA" should be provided when it is first mentioned in the text, rather than in the Methods section.

Reply: Thanks for this comment. We have added the full name of "PLA" to the first mentioned "Results" part of the revised manuscript (**page 5, lines 122**).

9. When conducting stability experiments using bovine serum, it is necessary to specify the environmental temperature in the methodology section.

Reply: Thanks for this suggestion. The stability experiments using bovine serum were performed at 37°C as already added in the "Methods" section of the revised manuscript (**page 36, lines 861**).

10. The "Ex vivo blood circulation thrombogenicity test" needs a clear description of how the blood vessels were connected to the PVC tube. The authors should provide more specific details of the experimental procedures for replication by other researchers. Reply: Thank you for pointing this out. The blood vessels were connected to the PVC tubes (F16, Guilong Medical Equipment Co., Ltd., China) using aspirator cannulas (CD 1.2 mm, Guilong Medical Equipment Co., Ltd., China) and indwelling needles (CD 1.3 mm, B. Braun Melsungen AG, Germany). To more clearly illustrate the "Ex vivo blood circulation thrombogenicity test", we have made a detailed annotation on the schematic diagram (**Fig. 3(E)**) and enriched the experimental procedure the experimental procedures as follows: Before the test, the PDA-PEI and (rhCol III/PDA-PEI)_n coatings (n = 0.5, 1, 2, and 4) were prepared on PVC tubes, and then connected in series by aspirator cannulas. The rabbits were under general anesthesia by injecting pentobarbital sodium (25 mg/mL, 0.7 mL/kg) through the ear vein. Then the right carotid artery and left external jugular vein of the rabbits were carefully separated and pierced with indwelling needles. Afterward, the tandem uncoated, PDA-PEI and (rhCol III /PDA-PEI)_n-coated PVC tubes were connected with indwelling needles to form a closed-loop circuit with the vessels. After 2 h of blood circulation, the samples were quickly removed and washed carefully with PBS. The thrombus bound to the sample surfaces was collected, weighed, and photographed. The cross-section of the harvested tubes was photographed to analyze the occlusion rates of the samples (**page 37, lines 879-**

892; page 38, lines 893-894).

Fig. R35 | (i.e. Fig. 3(E)) (E) Ex vivo schematic diagram of the arteriovenous shunt model in New Zealand white rabbits.

11. Some subsection titles in the Results section need modification to succinctly summarize the research findings. For example, "Hemocompatibility test" seems more appropriate as a title for a method rather than a result subsection.

Reply: We thank the reviewer for this valuable suggestion. As suggested, we modified some subsection titles in the "Results" section as follows: (1) "Hemocompatibility test" was modified to "Drug-free (rhCol III/PDA-PEI)_n coatings improve blood compatibility". (2) "Cell proliferation and morphology of vascular cells" was separated into "Drug-free (rhCol III/PDA-PEI)_n coatings regulate the growth behavior of HUVECs" and "Drug-free (rhCol III/PDA-PEI)_n coatings induce conversion of HUASMCs to contractile phenotype". (3) "In vitro and in vivo inflammatory response" was separated into "In vitro drug-free (rhCol III/PDA-PEI)_n coatings suppress inflammatory responses" and "Tissue compatibility of drug-free (rhCol III/PDA-PEI)_n coatings in rat model". (4) "In vivo endothelialization and neointima formation" was modified to "Stent implantation in rabbit and porcine models".

References

1. Yang, C. et al. Brush-Like Polycarbonates Containing Dopamine, Cations, and PEG Providing a Broad-Spectrum, Antibacterial, and Antifouling Surface via One-Step Coating. *Adv. Mater.* 26, 7346-7351 (2014).
2. Luo, R. et al. Dopamine-assisted deposition of poly (ethylene imine) for efficient heparinization. *Colloids and Surfaces B: Biointerfaces.* 144, 90-98 (2016).
3. Yang, L. et al. A robust mussel-inspired zwitterionic coating on biodegradable poly(L-lactide) stent with enhanced anticoagulant, anti-inflammatory, and anti-

- hyperplasia properties. *Chemical Engineering Journal*. 427, 130910 (2022).
4. Wang, J. et al. The biological effect of recombinant humanized collagen on damaged skin induced by UV-photoaging: An in vivo study. *Bioactive Materials*. 11, 154-165 (2022).
 5. Gao, D. et al. A deep learning approach to identify gene targets of a therapeutic for human splicing disorders. *Nature communications*. 12, 3332 (2021).
 6. Hao, H. et al. Expression of matrix Gla protein and osteonectin mRNA by human aortic smooth muscle cells. *Cardiovascular Pathology*. 13, 195-202 (2004).
 7. Stene, C. et al. MMP7 modulation by short-and long-term radiotherapy in patients with rectal cancer. *in vivo*. 32, 133-138 (2018).
 8. Tsantilas, P. et al. Chitinase 3 like 1 is a regulator of smooth muscle cell physiology and atherosclerotic lesion stability. *Cardiovascular Research*. 117, 2767-2780 (2021).
 20. Liu, Y., & Fanburg, B. L. Phospholipase D signaling in serotonin-induced mitogenesis of pulmonary artery smooth muscle cells. *American Journal of Physiology-Lung Cellular and Molecular Physiology*. 295, L471-L478 (2008).
 21. Pedroza, A. J. et al. Smooth Muscle Cell Embryologic Origin Does Not Define Propensity for Phenotype Modulation in Murine Marfan Syndrome Aortic Root Aneurysm. *Circulation*. 144, A12006-A12006 (2021).
 22. Schwertassek, U. et al. Myristoylation of the dual-specificity phosphatase c-JUN N-terminal kinase (JNK) stimulatory phosphatase 1 is necessary for its activation of JNK signaling and apoptosis. *The FEBS Journal*. 277, 2463-2473 (2010).
 23. Yuan, Y., Wang, C., Xu, J., Tao, J., Xu, Z., & Huang, S. BRG1 overexpression in smooth muscle cells promotes the development of thoracic aortic dissection. *BMC Cardiovascular Disorders*. 14, 1-9 (2014).
 24. Zhou, Z. et al. Dressing blood-contacting devices by platelet membrane enables large-scale multifunctional biointerfacing. *Matter*. 5, 1-18 (2022).
 25. Zhang, B. et al. A Polyphenol-Network-Mediated Coating Modulates Inflammation and Vascular Healing on Vascular Stents. *ACS Nano*. 16, 6585–6597 (2022).
 26. Chen, Y. et al. A tough nitric oxide-eluting hydrogel coating suppresses neointimal hyperplasia on vascular stent. *Nature communications*. 12, 7079 (2021).
 27. Noda, S., Suzuki, Y., Yamamura, H., Giles, W. R., & Imaizumi, Y. Roles of LRRC26 as an auxiliary γ 1-subunit of large-conductance Ca^{2+} -activated K^{+} channels in bronchial smooth muscle cells. *American Journal of Physiology-Lung Cellular and Molecular Physiology*. 318, L366-L375 (2020).
 28. Yang, L. et al. A robust mussel-inspired zwitterionic coating on biodegradable poly(L-lactide) stent with enhanced anticoagulant, anti-inflammatory, and anti-hyperplasia properties. *Chemical Engineering Journal*. 427, 130910 (2022).

29. Kang, E. Y. et al. Enhanced mechanical and biological characteristics of PLLA composites through surface grafting of oligolactide on magnesium hydroxide nanoparticles. *Biomaterials science*. 8, 2018-2030 (2020).
30. Baek, S. W. et al. Enhanced Mechanical Properties and Anti-Inflammation of Poly (L-Lactic Acid) by Stereocomplexes of PLLA/PDLA and Surface-Modified Magnesium Hydroxide Nanoparticles. *Polymers*. 14, 3790 (2022).
31. Riemann, A. et al. Acidic environment activates inflammatory programs in fibroblasts via a cAMP-MAPK pathway. *Biochim. Biophys. Acta*. 1853, 299-307 (2015).
32. Zhou, G., Niepel, M. S., Saretia, S., & Groth, T. Reducing the inflammatory responses of biomaterials by surface modification with glycosaminoglycan multilayers. *Journal of Biomedical Materials Research Part A*. 104(2), 493-502 (2016).
33. Gorog, D., Fayad, Z., Fuster, V. et al. Arterial Thrombus Stability: Does It Matter and Can We Detect It? *J Am Coll Cardiol*. 70, 2036-2047 (2017).
34. Yao, Y. et al. Fucoidan and topography modification improved in situ endothelialization on acellular synthetic vascular grafts. *Bioactive Materials*. 22, 535-550 (2023).
35. Zhang, B., Yao, R., Li, L., Wang, Y. Green tea polyphenol induced Mg²⁺-rich multilayer conversion coating: toward enhanced corrosion resistance and promoted in situ endothelialization of AZ31 for potential cardiovascular applications. *ACS Appl. Mater. Interfaces*. 11, 41165-41177 (2019).
36. Wang, Y. et al. A thrombin-triggered self-regulating anticoagulant strategy combined with anti-inflammatory capacity for blood-contacting implants. *Science advances*. 8, eabm3378 (2022).
37. Zhao, Q. et al. Programmed shape-morphing scaffolds enabling facile 3D endothelialization. *Advanced Functional Materials*. 28, 1801027 (2018).
38. Margariti, A., Li, H., Chen, T. et al. XBP1 mRNA splicing triggers an autophagic response in endothelial cells through BECLIN-1 transcriptional activation. *Journal of Biological Chemistry*. 288, 859-872 (2013).
39. Zhang, W., Xu, J., Fang, H. et al. Endothelial cells promote triple-negative breast cancer cell metastasis via PAI-1 and CCL5 signaling. *The FASEB Journal*. 32, 276-288 (2018).
40. Xu, Y. et al. GATA3-induced vWF upregulation in the lung adenocarcinoma vasculature. *Oncotarget*. 8, 110517 (2017).
41. Ieta, K. et al. CEACAM6 gene expression in intrahepatic cholangiocarcinoma. *British journal of cancer*. 95, 532-540 (2006).
42. Liu, Z. et al. Structures of the gasdermin D C-terminal domains reveal mechanisms of autoinhibition. *Structure*. 26, 778-784 (2018).
9. Chao, K. L., Kulakova, L., & Herzberg, O. Gene polymorphism linked to increased asthma and IBD risk alters gasdermin-B structure, a sulfatide

and phosphoinositide binding protein. *Proceedings of the National Academy of Sciences*. 114, E1128-E1137 (2017).

10. Hu, S. et al. Exosome-eluting stents for vascular healing after ischaemic injury. *Nature Biomedical Engineering*. 5, 1174-1188 (2021).

11. Liu, Y., Lu, J., Li, H., Wei, J., Li, X. Engineering blood vessels through micropatterned co-culture of vascular endothelial and smooth muscle cells on bilayered electrospun fibrous mats with pDNA inoculation. *Acta Biomater*. 11, 114-125. (2015).

12. Yang, L. et al. A tailored extracellular matrix (ECM) - Mimetic coating for cardiovascular stents by stepwise assembly of hyaluronic acid and recombinant human type III collagen. *Biomaterials*. 276, 121055 (2021).

13. Hua, R. et al. The effect of intrinsic characteristics on mechanical properties of poly (l-lactic acid) bioresorbable vascular stents. *Medical Engineering & Physics*. 81, 118124 (2020).

14. Bergsma, J. E. et al. Biocompatibility and degradation mechanisms of predegraded and non-predegraded poly (lactide) implants: an animal study. *Journal of Materials*

Reviewers' Comments:

Reviewer #1:

Remarks to the Author:

The revised manuscript can be accepted.

The approaches used are appropriate and sufficient to support the conclusions, and the concerns of Reviewer 4 have been addressed.

Reviewer #2:

Remarks to the Author:

I co-reviewed this manuscript with one of the reviewers who provided the listed reports. This is part of the Nature Communications initiative to facilitate training in peer review and to provide appropriate recognition for Early Career Researchers who co-review manuscripts

Reviewer #3:

Remarks to the Author:

The authors have fully responded to all prior concerns and addressed critical points by new data.

Flow cytometric plots do not convincingly demonstrate differences in CD86 staining (Fig. 6D). This data should be removed.

REVIEWERS' COMMENTS

Reviewer #1 (Remarks to the Author):

The revised manuscript can be accepted.

The approaches used are appropriate and sufficient to support the conclusions, and the concerns of Reviewer 4 have been addressed.

Thank you for your comments and we appreciate your effort in helping us with editing our manuscript.

Reviewer #2 (Remarks to the Author):

I co-reviewed this manuscript with one of the reviewers who provided the listed reports. This is part of the Nature Communications initiative to facilitate training in peer review and to provide appropriate recognition for Early Career Researchers who co-review manuscripts

Thank you for your comments and we appreciate your effort in helping us with editing our manuscript.

Reviewer #3 (Remarks to the Author):

The authors have fully responded to all prior concerns and addressed critical points by new data. Flow cytometric plots do not convincingly demonstrate differences in CD86 staining (Fig. 6D). This data should be removed.

Thank you for your comments and we appreciate your effort in helping us with editing our manuscript. As suggested, we have removed the flow cytometry data in Fig. 6(d) entirely from the revised manuscript.